# Skywork-Reward-V2: Scaling Preference Data Curation via Human-AI Synergy

**Chris Yuhao Liu, Liang Zeng✉, Yuzhen Xiao, Jujie He, Jiacai Liu, Chaojie Wang, Rui Yan, Wei Shen, Fuxiang Zhang, Jiacheng Xu, Yang Liu✉**

2050 Research, Skywork AI

## Abstract

Despite the critical role of reward models (RMs) in Reinforcement Learning from Human Feedback (RLHF), current state-of-the-art open RMs perform poorly on most existing evaluation benchmarks, failing to capture the spectrum of nuanced and sophisticated human preferences. Even approaches incorporating advanced training techniques have failed to yield meaningful performance improvements. We hypothesize that this brittleness stems primarily from limitations in preference datasets, which are often narrowly scoped, synthetically labeled, or lack rigorous quality control. To address these challenges, we present a large-scale preference dataset comprising 40 million preference pairs, named SynPref-40M. To enable data curation at scale, we design a human-AI synergistic two-stage pipeline that leverages the complementary strengths of human annotation quality and AI scalability. In this pipeline, humans provide verified annotations, while Large Language Models (LLMs) perform automatic curation based on human guidance. Training on this preference mixture, we introduce Skywork-Reward-V2, a suite of eight reward models ranging from 0.6B to 8B parameters, trained on a carefully curated subset of 26 million preference pairs from SynPref-40M. We demonstrate that Skywork-Reward-V2 is versatile across a wide range of capabilities, including alignment with human preferences, objective correctness, safety, resistance to stylistic biases, and best-of-N scaling. These reward models achieve state-of-the-art performance across seven major reward model benchmarks, outperform the latest paradigm of generative reward models, and demonstrate strong downstream performance. Ablation studies confirm that the effectiveness of our approach stems not only from data scale but also from high-quality curation. The Skywork-Reward-V2 series represents substantial progress in open reward models, highlighting the untapped potential of existing preference datasets and demonstrating how human-AI curation synergy can unlock significantly higher data quality.

## 1 Introduction

Reward models (RMs) have become critical components in Reinforcement Learning from Human Feedback (RLHF) pipelines (Christiano et al., 2017; Stiennon et al., 2020; Ouyang et al., 2022; Dong et al., 2024a; Lambert, 2025; Schulman et al., 2017), now standard in Large Language Model (LLM) post-training (Tie et al., 2025). Recent advancements in LLM reasoning capabilities (Jaech et al., 2024; Guo et al., 2025; Xu et al., 2025; Chen et al., 2025a) and Reinforcement Learning with Verifiable Rewards (RLVR) (Lambert et al., 2024) have sparked interest in policy optimization via rule-based rewards (Luo et al., 2025c; Wen et al., 2025; Team, 2025b;a; Luo et al., 2025b; He et al., 2025b). These reward functions typically verify whether answers match ground truth for math problems or pass unit tests for coding tasks, and can include fine-grained rules for verifiable outputs (Bercovich et al., 2025; Ma et al., 2025). However, complex human preferences often cannot be captured through simple rules, limiting the effectiveness of rule-based approaches in advancing general preference learning. Thus, the challenge of modeling nuanced, sophisticated, and sometimes conflicting human preferences through effective reward models remains largely unresolved.

---

✉ Corresponding authors.

To model human preferences, previous works have curated various datasets (Cui et al., 2023; Wang et al., 2025d; Dong et al., 2024a; Xu et al., 2024; Park et al., 2024; Lambert et al., 2024; OLMo et al., 2024) with prompts drawn from diverse sources. These efforts employ automatic methods (Cui et al., 2023; Xu et al., 2024) or human annotators (Wang et al., 2024e; 2025d) to generate preference pairs, enabling preference learning in a pairwise contrastive manner (Bradley & Terry, 1952; Ouyang et al., 2022). Beyond dataset construction, some works aim to improve reward modeling via inductive biases in enhanced loss functions (Liu et al., 2024b; Cai et al., 2024; Yang et al., 2024b; Wang et al., 2024e; Zhang et al., 2024b) or modified model architectures (Wang et al., 2024a; Chen et al., 2025b; Dorka, 2024). To evaluate progress in reward modeling, RewardBench (Lambert et al., 2025) was released as the first benchmark for RMs. As reward models evolve, scores on RewardBench have begun to saturate (Wang et al., 2024a; Park et al., 2024; Wang et al., 2024b; Liu et al., 2024b; Shiwen et al., 2024; Wang et al., 2025b; 2024d), but multiple studies (Frick et al., 2024; Zhou et al., 2024; Song et al., 2025; Wen et al., 2024) have argued that such saturated scores are weak indicators of real progress. These studies highlight weak (or even inverse) correlations between RewardBench scores and downstream task performance (e.g., best-of-N or policy training).

In this work, **we focus exclusively on the dual goal of both enhancing the quality and scaling the quantity of preference data**, to advance the development of open reward models. We introduce SynPref-40M, a large-scale preference dataset comprising 40 million preference pairs. We design a two-stage preference data curation pipeline (Figure 2) that (1) combines human verification under a stringent protocol for quality assurance (Section 3.2), and (2) employs human-preference-guided LLM judges for scalability (Section 3.3). The pipeline also involves iterative training of a reward model, which continuously incorporates feedback from human labels and retrieves preference data where the RM itself performs poorly, to enable further learning. Our pipeline yields 26 million carefully curated preference pairs, which we use to develop and train Skywork-Reward-V2, our second-generation reward model series, consisting of eight high-performing reward models, ranging from 0.6B to 8B parameters.

Through comprehensive evaluations on seven major RM benchmarks (Lambert et al., 2025; Frick et al., 2024; Zhou et al., 2024; Liu et al., 2024c; Tan et al., 2024; Malik et al., 2025), we demonstrate that the Skywork-Reward-V2 series, trained using only the Bradley-Terry objective (Bradley & Terry, 1952), achieves state-of-the-art performance, with our 8B reward model **outperforming all existing open reward models across all seven benchmarks by a significant margin**. We also demonstrate Skywork-Reward-V2's superior performance across multiple critical dimensions, including general human preferences, objective correctness, resistance to stylistic biases, safety, and best-of-N scaling (Section 4.2). Through data ablations, we show that the success of SynPref-40M is driven not only by its scale but also by its high quality (Section 4.3). Our method-wise ablations confirm the importance of human annotation, LLM annotation guided by human preferences, and our carefully designed and rigorously implemented annotation protocols (Section 4.4).

We outline our main contributions as follows:

- We collect and curate SynPref-40M, which, to the best of our knowledge, is the largest curated preference mixture to date.
- We release Skywork-Reward-V2, a series of eight state-of-the-art reward models ranging from 0.6B to 8B parameters, which achieve top rankings on seven major reward model benchmarks, demonstrating strong performance across diverse evaluation dimensions.
- We propose a preference data curation pipeline that combines human verification for quality with LLM-as-a-Judge, guided by human preferences for scalability.

## 2 THE BRITTLENESS OF CURRENT OPEN REWARD MODELS

**Single-benchmark evaluation has limitations.** RewardBench (Lambert et al., 2025) is a dataset for pairwise preference evaluation in chat, safety, and reasoning, and has become the standard benchmark for assessing reward models. However, several subsequent studies (Frick et al., 2024; Zhou et al., 2024; Wen et al., 2024) argue that scores on RewardBench do not directly correlate with downstream performance and, in some cases, exhibit an inverse relationship. Our evaluation results in Figure 1 corroborate this concern: while RewardBench shows positive correlation with other benchmarks overall, improvements on RewardBench from ∼80 to 90+ do not consistently translate to gains on other benchmarks. We advocate for benchmarks that either (1) involve more challeng-

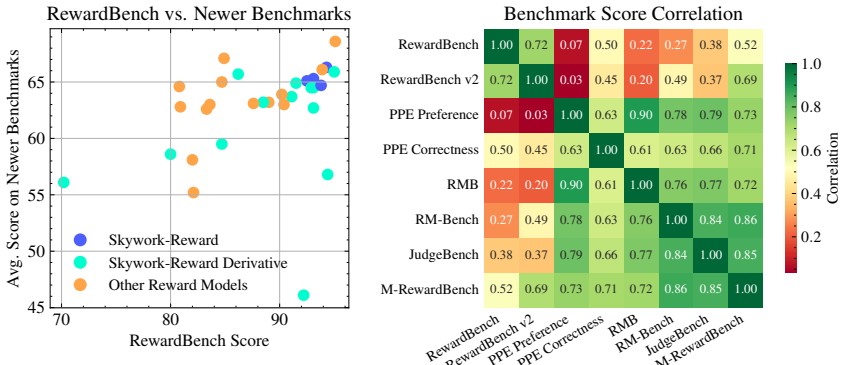

Figure 1: **Left:** Comparison of the performance of 31 top open reward models on RewardBench (Lambert et al., 2025) and their average scores across seven newer benchmarks (Frick et al., 2024; Zhou et al., 2024; Liu et al., 2024c; Tan et al., 2024; Gureja et al., 2024). **Right:** Pearson correlation scores across seven reward model benchmarks.

ing evaluation methods (e.g., best-of-N) or (2) demonstrate stronger correlations with downstream performance.

**A comprehensive evaluation suite reveals inconsistent improvements.** Based on the above criteria, in addition to RewardBench, we select several other benchmarks that span multiple evaluation dimensions. Specifically, we include PPE Preference and Correctness (Frick et al., 2024) to assess both real human preferences and unambiguous correctness; RMB (Zhou et al., 2024) for its challenging best-of-N evaluation; RM-Bench (Liu et al., 2024c) to evaluate robustness to content variation and style bias; and JudgeBench (Tan et al., 2024), which evaluates preference pairs drawn from difficult, real-world LLM evaluation datasets, such as LiveCodeBench (Jain et al., 2024). Finally, we include the newly released RewardBench v2 (Malik et al., 2025), which enforces global best-of-N evaluation and extremely difficult capability assessments (e.g., distinguishing highly similar responses and reward margin requirements). A detailed description of these benchmarks is provided in Section C.1. We present the main results in Figure 1, comparing RewardBench scores with average scores across the seven newer benchmarks, and report Pearson correlations among all benchmarks. Our findings are as follows:

- **Improvements on RewardBench do not guarantee broader gains.** As model scores on RewardBench increase from ∼80 to 90+, performance on other benchmarks does not consistently improve — it may get better, worse, or remain approximately the same. This inconsistency, combined with the weak correlations shown in the right plot of Figure 1, suggests that researchers and practitioners should avoid interpreting reward model quality based on a single benchmark.
- **Alternative loss functions or model modifications fail to yield consistent gains for the Gemma-2-27B variants** (Yang et al., 2024b; Dorka, 2024; Lou et al., 2024; Zhang et al., 2024b; Liu et al., 2025). When examining the 27B models derived from Gemma in our experiments (see Table 7 in the appendix), the original Skywork-Reward-Gemma-2-27B-v0.2 remains the best RM in terms of average performance, while all variants with loss modifications fall behind. However, we acknowledge this claim is limited to this specific model series and does not generalize to all model modifications.
- Among the top 20 models on RewardBench, 16 directly or indirectly use the same base model (Liu et al., 2024b) or are fine-tuned on highly similar training data, indicating **stagnant progress in both open preference datasets and reward models** since September 2024.

## 3 SCALING PREFERENCE DATA CURATION VIA HUMAN-GUIDED AI FEEDBACK

### 3.1 PIPELINE OVERVIEW

Our pipeline (Figure 2) has two stages. In **Stage 1**, human annotators label *gold* data under a strict protocol, while LLMs produce *silver* labels conditioned on human preferences. A reward model trained on *silver* data is evaluated against *gold* data, and an adaptive retrieval mechanism selects new samples where the RM performs poorly for re-annotation. This iterates over multiple rounds.

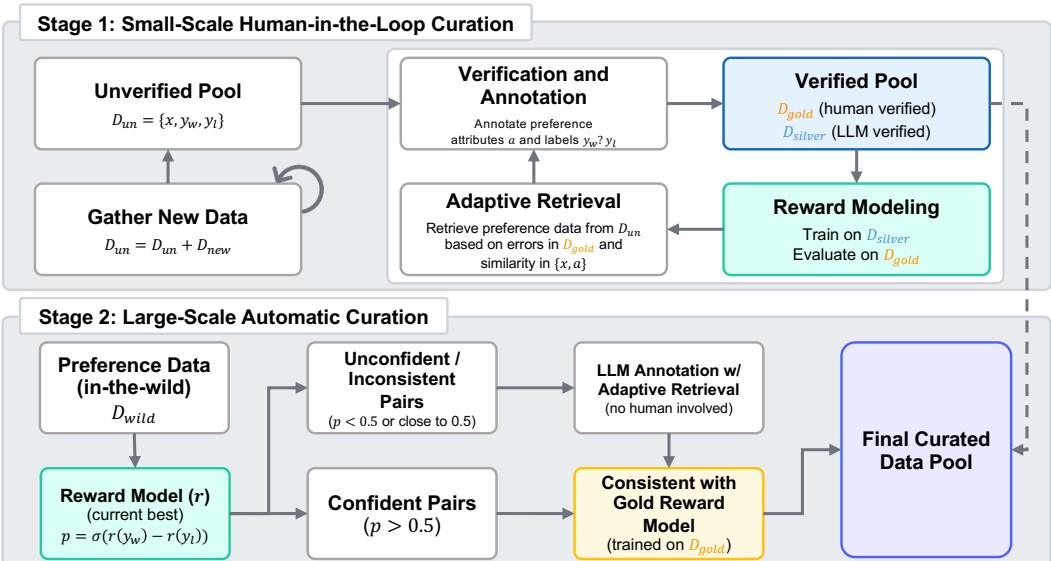

Figure 2: A two-stage preference data curation pipeline. **Stage 1 (top)** involves human-AI synergistic curation and runs iteratively. **Stage 2 (bottom)** scales data curation automatically using reward model consistency checks, eliminating the need for further human supervision.

In **Stage 2**, the Stage 1 reward model and a gold reward model (trained on verified human data only) jointly guide consistency-based data selection at scale, requiring no further human supervision.

## 3.2 STAGE 1: SMALL-SCALE HUMAN-IN-THE-LOOP CURATION

**Overview.** Stage 1 is an iterative procedure consisting of 8 iterations. In each iteration, we focus on two things: (1) collecting more unverified preference pairs $\mathcal{D}_{un}$, and (2) producing a silver set $\mathcal{D}_{silver}$ via LLM annotation and a gold set $\mathcal{D}_{gold}$ via human annotation to train and identify flaws in the current reward model. Throughout Stage 1, we accumulate roughly 1M preference pairs in total.

**Seed preference data initialization.** We begin by collecting available preference data to form an unverified pool, $\mathcal{D}_{un}$. During preference data collection, we source our data from publicly available preference pairs, primarily collected from a wide range of sources (over 40) on Hugging Face, provided that the licenses and terms of use permit it (see Section D.2 for detailed composition and licensing information). For each pair in this pool, given the 3-tuple $(x, y_w, y_l)$ — comprising the conversation $x$, the chosen (winning) response $y_w$, and the rejected (losing) response $y_l$ — we collect LLM-generated preference attributes $a$. Each attribute set is a 5-tuple consisting of: (1) task category, (2) preference objectivity, (3) controversiality, (4) desired attributes, and (5) annotation guideline. Task category, objectivity, and controversiality serve as metadata to ensure annotation diversity across scenarios. The desired attributes describe the qualities users seek in good responses, while the annotation guideline provides instance-specific, context-dependent criteria for determining the preference label. We provide examples of these attributes and quality analysis in Section E.

**Human verification and annotation protocol.** We initialize with a small, high-quality, and diverse set of preference pairs[1] as the *seed data*. Using the generated preference attributes, human annotators perform strict verification following a predefined protocol (detailed in Section E.2). At a high level, the protocol outlines core principles and practices, as well as specific guidelines tailored to each task category, objectivity type, and controversiality level. For example, it permits the use of external tools — such as search engines, frontier LLM assistants, and domain-specialized LLMs (e.g., for math or code) — to aid in labeling. However, full reliance on LLMs for labeling is strictly prohibited. Specifically, annotators provide a final judgment based on the attributes and the instance-specific guideline. For tasks involving fact-checking, annotators are required to use a search engine; for code correctness, annotators instruct an LLM to execute the code and verify correctness. Even when LLMs and external tools are used, annotators remain responsible for the final judgment. (Note that we still consider this process "human annotation.") Preference pairs produced during this process

---

[1]In our case, we use the Skywork-Reward-Preference-80K-v0.2 dataset.

are collected as $\mathcal{D}_{\text{gold}}$. This rigorous process yields the seed dataset $\mathcal{D}_{\text{seed}}$, where the human-verified portion is denoted as $\mathcal{D}_{\text{gold}}$ (for validation), and the LLM-verified portion as $\mathcal{D}_{\text{silver}}$ (for training). Importantly, the preference attributes in $\mathcal{D}_{\text{gold}}$ are also edited and verified by human annotators, ensuring higher quality.

**Step 1: Reward model training and evaluation.** We initialize a pointwise Bradley-Terry reward model (Bradley & Terry, 1952; Ouyang et al., 2022) and train it on $\mathcal{D}_{\text{silver}}$. We select the best current reward model checkpoint $\theta$ based on validation accuracy on $\mathcal{D}_{\text{gold}}$. This checkpoint is referred to as the "best reward model" throughout the paper, including in Stage 2 filtering. For each $(x, y_w, y_l)$, we collect its prediction $p = \sigma(r_\theta(x, y_w) - r_\theta(x, y_l))$.

**Step 2: Error-driven adaptive preference retrieval.** Instead of relying solely on human-annotated data to increase data volume, we leverage LLM annotators via an adaptive retrieval mechanism (Ram et al., 2023) to collect representative samples aligned with human preferences. This step involves two separate retrieval processes. First, we identify pairs where the current RM performs poorly: we evaluate the current reward model on $\mathcal{D}_{\text{gold}}$ to identify pairs it misclassifies, then sample new pairs from $\mathcal{D}_{\text{un}}$ that are similar to these misclassified pairs. This first retrieval selects samples at the current RM's weak spots for subsequent LLM annotation. This mechanism selects new examples from the unverified pool based on both the preference attributes $a$ and the reward model's predictions. For each pairwise instance, we compute the embedding (Sturua et al., 2024) of $(x, a)$ and retrieve the top-$k$ similar items. Intuitively, we prioritize preference data that resemble instances where the reward model errs or shows low confidence. We set the retrieval upper bound $k_{\max} = 8$ and use a dynamic rule to determine $k$:

$$k = \begin{cases} k_{\max}, & \text{if } p \le 0.5 \quad \text{(incorrect prediction)} \\ \lceil k_{\max} \cdot (1 - p) \rceil, & \text{if } p > 0.5 \quad \text{(correct prediction)} \end{cases}$$

**Step 3: Preference-aware labeling.** To augment LLM annotation with gold human labels, we perform a second retrieval step: for each pair selected in Step 2, we retrieve similar pairs from $\mathcal{D}_{\text{gold}}$ (which are human-labeled) and insert them as few-shot examples to guide the LLM in making the final judgment. This ensures that LLM annotation is conditioned on human-verified examples throughout the process. Using the retrieved examples with human labels, we employ a group of strong LLMs to aggregate final judgments using self-consistency (Wang et al., 2022). First, we perform intra-model aggregation via self-consistency, then merge results across models to mitigate potential bias from any single model. The list of LLMs used and the annotation prompts are provided in Section E. For all LLM annotations, responses are labeled as "Candidate 1" and "Candidate 2," with their order randomized in the prompt. While pointwise scoring (He et al., 2025a; Liu et al., 2025) has shown greater effectiveness, it is not applicable here due to our reliance on both human and LLM annotators, making it impractical to enforce a shared standard. Finally, human-labeled samples are added to $\mathcal{D}_{\text{gold}}$, and LLM-labeled samples to $\mathcal{D}_{\text{silver}}$. Throughout Stage 1, we iteratively perform Steps 1, 2, and 3. After each iteration, we use an internal human-labeled validation set for sanity checking. However, scores from this sanity check serve only as a reference; pipeline execution does not depend on them.

### 3.3 STAGE 2: LARGE-SCALE AUTOMATIC CURATION OF PREFERENCE DATA IN THE WILD

We now scale up to tens of millions of in-the-wild preference data pairs. We denote this set as $\mathcal{D}_{\text{wild}}$, which contains the remaining publicly available preference pairs not allocated to $\mathcal{D}_{\text{un}}$ in Stage 1. Like $\mathcal{D}_{\text{un}}$, $\mathcal{D}_{\text{wild}}$ is sourced from publicly available preference pairs collected from over 40 diverse sources on Hugging Face, provided that licenses and terms of use permit it. We allocate all originally human-labeled pairs to $\mathcal{D}_{\text{un}}$ (used in Stage 1) and leave the rest to $\mathcal{D}_{\text{wild}}$ (used in Stage 2). See Section D.2 for full details. However, annotating the entire dataset — even automatically — can be prohibitively costly and unnecessary. Below, we describe two consistency-based filtering strategies to determine which data points warrant further verification. Importantly, Stage 2 involves no human annotation; all annotation and filtering are performed by LLMs and the best reward model checkpoint from Stage 1.

**Preference consistency with the best reward model.** Inspired by Kim et al. (2024) and Liu et al. (2024b), we skip (i.e., directly retain without re-annotation) all pairs where the reward model's confidence exceeds 0.5 under the current best reward model. For the remaining pairs where a mismatch is detected (i.e., the preference pair does not agree with the current best reward model), we fall back to LLM annotation to make the final judgment. This LLM annotation step uses exactly the same

| Model | RewardBench | RewardBench v2 | PPE Pref | PPE Corr | RMB | RM-Bench | JudgeBench | Avg. |
|---|---|---|---|---|---|---|---|---|
| *Open Reward Models* | | | | | | | | |
| Llama-3-OffsetBias-RM-8B (Park et al., 2024) | 89.0 | 64.8 | 59.2 | 64.1 | 57.8 | 71.3 | 63.5 | 67.1 |
| ArmoRM-Llama3-8B-v0.1 (Wang et al., 2024a) | 90.4 | 66.5 | 60.6 | 60.6 | 64.6 | 69.2 | 59.7 | 67.4 |
| Internlm2-20b-reward (Cai et al., 2024) | 90.2 | 56.3 | 61.0 | 63.0 | 62.9 | 72.1 | 64.3 | 67.1 |
| Skywork-Reward-Llama-3.1-8B-v0.2 (Liu et al., 2024b) | 93.1 | 71.8 | 62.2 | 62.5 | 66.6 | 64.7 | 62.9 | 69.1 |
| LDL-Reward-Gemma-2-27B-v0.1 | 95.0 | 72.5 | 62.4 | 63.9 | 67.9 | 71.0 | 64.2 | 71.0 |
| Skywork-Reward-Gemma-2-27B-v0.2 (Liu et al., 2024b) | 94.3 | 75.3 | 63.6 | 61.9 | 69.4 | 67.6 | 66.5 | 71.2 |
| Llama-3.1-Nemotron-70B (Wang et al., 2024d) | 93.9 | 76.7 | 64.2 | 63.2 | 64.9 | 72.2 | 65.8 | 71.6 |
| INF-ORM-Llama3.1-70B (Yang et al., 2024b) | 95.1 | 76.5 | 64.2 | 64.4 | 70.5 | 75.4 | 70.2 | 73.8 |
| *LLM-as-a-Judge & Generative Reward Models* | | | | | | | | |
| GPT-4o (Hurst et al., 2024) | 86.7 | 64.9 | 67.7 | 67.1 | 73.8 | 73.1 | 59.8 | 70.4 |
| Claude-3.5-Sonnet (Anthropic, 2024) | 84.2 | 64.7 | 67.3 | 69.2 | 70.6 | 74.5 | 64.8 | 70.8 |
| DeepSeek-GRM-27B (Liu et al., 2025) | 88.5 | - | 65.3 | 60.4 | 69.0 | - | - | - |
| DeepSeek-GRM-27B (w/ MetaRM) (Liu et al., 2025) | 90.4 | - | 67.2 | 63.2 | 70.3 | - | - | - |
| RM-R1-Qwen-Instruct-32B (Chen et al., 2025c) | 92.9 | - | - | - | 73.0 | 79.1 | - | - |
| RM-R1-DeepSeek-Distill-Qwen-32B (Chen et al., 2025c) | 90.9 | - | - | - | 69.8 | 83.9 | - | - |
| EvalPlanner (Llama-3.1-70B) (Saha et al., 2025) | 93.9 | - | - | - | - | 80.0 | 50.9 | - |
| EvalPlanner (Llama-3.3-70B) (Saha et al., 2025) | 93.8 | - | - | - | - | 82.1 | 56.6 | - |
| J1-Llama-8B (Whitehouse et al., 2025) | 85.7 | - | 60.3 | 59.2 | - | 73.4 | 42.0 | - |
| J1-Llama-8B (Maj@32) (Whitehouse et al., 2025) | - | - | 60.6 | 61.9 | - | - | - | - |
| J1-Llama-70B (Whitehouse et al., 2025) | 93.3 | - | 66.3 | 72.9 | - | 82.7 | 60.0 | - |
| J1-Llama-70B (Maj@32) (Whitehouse et al., 2025) | - | - | 67.0 | 73.7 | - | - | - | - |
| *Our Reward Models* | | | | | | | | |
| Skywork-Reward-V2-Qwen3-0.6B | 85.2 | 61.3 | 65.3 | 68.3 | 74.5 | 74.4 | 67.6 | 70.9 |
| Skywork-Reward-V2-Qwen3-1.7B | 90.3 | 68.3 | 67.6 | 70.5 | 78.1 | 78.7 | 72.9 | 75.2 |
| Skywork-Reward-V2-Qwen3-4B | 93.4 | 75.5 | 69.5 | 74.7 | 80.6 | 81.6 | 69.5 | 77.8 |
| Skywork-Reward-V2-Qwen3-8B | 93.7 | 78.2 | 70.6 | 75.1 | 81.2 | 82.6 | 73.4 | 79.3 |
| Skywork-Reward-V2-Llama-3.2-1B | 89.9 | 64.3 | 66.6 | 67.4 | 76.7 | 76.3 | 65.0 | 72.3 |
| Skywork-Reward-V2-Llama-3.2-3B | 93.0 | 74.7 | 69.1 | 72.1 | 80.5 | 81.1 | 69.2 | 77.1 |
| Skywork-Reward-V2-Llama-3.1-8B | 96.4 | 84.1 | 77.3 | 83.4 | 86.4 | 92.8 | 80.0 | 85.8 |
| Skywork-Reward-V2-Llama-3.1-8B-40M | **97.8** | **86.5** | **79.8** | **87.2** | **89.3** | **96.0** | **83.4** | **88.6** |

Table 1: Reward model performance assessed on seven benchmarks. **Bold** numbers indicate the best performance among all models, while underlined numbers represent the second best. Entries marked with "-" indicate that a model is unreleased. A complete evaluation is provided in Table 7.

procedure as in Stage 1: we retrieve similar pairs from $\mathcal{D}_{\text{gold}}$ and insert them as few-shot examples to help the LLM make the final judgment. We apply the same adaptive preference retrieval and human-preference-guided LLM annotation from Section 3.2 without involving human verifiers.

**Preference consistency with the gold reward model.** We train a separate gold reward model using all cumulative human-verified samples to approximate the "true" human preference distribution. From the unverified pool, we retain only those pairs whose original chosen-rejected labels are consistent with (1) the gold reward model and (2) either the LLM judges or the current best reward model. Approximately 5 million preference pairs passed through this consistency mechanism without requiring attribute generation or additional labeling. To leverage the discarded pool, we also experiment with "recycling" the discarded data by simply flipping the chosen-rejected order, which incurs no additional annotation or computational overhead.

## 4 EXPERIMENTAL RESULTS

### 4.1 REWARD MODEL TRAINING

We train all models in the Skywork-Reward-V2 series using the Llama 3.1 and 3.2 series (Grattafiori et al., 2024) and the Qwen3 (Yang et al., 2025) collection as backbones, with no more than 8B parameters. From the Llama 3 series, we employ Llama-3.1-8B-Instruct, Llama-3.2-3B-Instruct, and Llama-3.2-1B-Instruct. For Qwen3, we consider sizes of 0.6B, 1.7B, 4B, and 8B. Although larger backbones (e.g., 70B) yield greater gains (Malik et al., 2025), we do not consider them due to training cost and deployment practicality.

All reward models are trained with a maximum context length of 16K tokens, which encompasses the majority of the samples in our data mixture to avoid truncation. We conduct all early-stage experiments with varying learning rates based on model size, using a linear decay schedule and a small batch size of 256, for 1 epoch, following the hyperparameters in Lambert et al. (2024). For all final model training runs, we adopt the hyperparameters from Wang et al. (2025a), with a large global batch size of 10,240 and a constant learning rate schedule. We observe no change in final performance; however, using a large batch size significantly reduces convergence time, saving approximately 35% of total training compute. We train all reward models in the Skywork-Reward-V2 series exclusively on the 26 million curated subset. As an experimental release, Skywork-Reward-V2-Llama-3.1-8B-40M is trained using 26 million curated pairs, along with additional pairs that have a flipped chosen-rejected order (i.e., those that agree with humans) from the discarded 14 million pairs.

| Model | Knowledge | Reasoning | Math | Coding | Avg. |
|---|---|---|---|---|---|
| GPT-4o | 50.6 | 54.1 | 75.0 | 59.5 | 59.8 |
| Claude-3.5-Sonnet | 62.3 | 66.3 | 66.1 | 64.3 | 64.8 |
| DeepSeek-R1 | 59.1 | 82.7 | 80.4 | 92.9 | 78.8 |
| o1-preview | 66.2 | 79.6 | 85.7 | 85.7 | 79.3 |
| o3-mini | 58.4 | 62.2 | 82.1 | 78.6 | 70.3 |
| o3-mini (low) | 63.0 | 69.4 | 83.4 | 83.3 | 74.8 |
| o3-mini (medium) | 62.3 | 86.7 | 85.7 | 92.9 | 81.9 |
| o3-mini (high) | 67.5 | 89.8 | 87.5 | 100 | 86.2 |
| Skywork-Reward-V2-Qwen3-0.6B | 62.3 | 66.3 | 82.1 | 59.5 | 67.5 |
| Skywork-Reward-V2-Qwen3-1.7B | 66.9 | 69.4 | 83.9 | 71.4 | 72.9 |
| Skywork-Reward-V2-Qwen3-4B | 66.9 | 64.3 | 80.4 | 66.7 | 69.5 |
| Skywork-Reward-V2-Qwen3-8B | 70.1 | 67.3 | 82.1 | 73.8 | 73.3 |
| Skywork-Reward-V2-Llama-3.2-1B | 61.0 | 66.3 | 73.2 | 59.5 | 65.0 |
| Skywork-Reward-V2-Llama-3.2-3B | 64.3 | 65.3 | 87.5 | 59.5 | 69.2 |
| Skywork-Reward-V2-Llama-3.1-8B | 76.6 | 75.5 | 89.3 | 78.6 | 80.0 |
| Skywork-Reward-V2-Llama-3.1-8B-40M | 79.9 | 78.6 | 89.3 | 85.7 | 83.4 |

Table 2: Performance comparison of RMs with state-of-the-art LLM-as-a-Judge and reasoning models on JudgeBench (Tan et al., 2024).

| Model | Helpfulness (BoN) | Harmlessness (BoN) | Avg. |
|---|---|---|---|
| Skywork-Reward-Llama-3.1-8B-v0.2 | 60.5 | 56.8 | 58.7 |
| Skywork-Reward-Gemma-2-27B-v0.2 | 63.1 | 59.9 | 61.5 |
| DeepSeek-GRM-27B | 63.9 | 58.0 | 61.0 |
| DeepSeek-GRM-27B + MetaRM | 64.2 | 58.0 | 61.1 |
| RM-R1-DeepSeek-Distill-Qwen-32B | 62.0 | 61.8 | 61.9 |
| RM-R1-Qwen-Instruct-32B | 63.6 | 68.2 | 65.9 |
| Qwen2-72B-Instruct | 64.5 | 64.9 | 64.7 |
| GPT-4o-2024-05-03 | 63.9 | 68.2 | 66.1 |
| Skywork-Reward-V2-Qwen3-0.6B | 68.4 | 69.1 | 68.8 |
| Skywork-Reward-V2-Qwen3-1.7B | 72.0 | 72.2 | 72.1 |
| Skywork-Reward-V2-Qwen3-4B | 74.7 | 75.1 | 74.9 |
| Skywork-Reward-V2-Qwen3-8B | 76.5 | 75.8 | 76.2 |
| Skywork-Reward-V2-Llama-3.2-1B | 68.0 | 73.2 | 70.6 |
| Skywork-Reward-V2-Llama-3.2-3B | 74.4 | 76.2 | 75.3 |
| Skywork-Reward-V2-Llama-3.1-8B | 82.3 | 82.8 | 82.5 |
| Skywork-Reward-V2-Llama-3.1-8B-40M | 86.2 | 86.6 | 86.4 |

Table 3: Reward model pairwise accuracy on the Best-of-N split for Helpfulness and Harmlessness in RMB (Zhou et al., 2024).

## 4.2 A COMPREHENSIVE EVALUATION OF THE SKYWORK-REWARD-V2 SERIES

We evaluate on seven major benchmarks (details in Section C.1).

**General preferences.** We report full benchmark results for the current top-performing reward models, LLM-as-a-Judge models, and generative reward models in Table 1. Across all seven benchmarks, the Skywork-Reward-V2 series of models outperform not only much larger ones (i.e., 70B) but also the emerging class of generative reward models (Liu et al., 2025; Chen et al., 2025c). We interpret this as strong evidence that SynPref-40M captures a wide range of preferences, enabling more robust preference learning across multiple dimensions simultaneously. Meanwhile, Skywork-Reward-V2 highlights the importance of data quality relative to the strength of the base models. Even at a scale of 1.7B parameters, a reward model can outperform a 70B model on all benchmarks except for RewardBench and RewardBench v2, effectively bridging the model size gap.

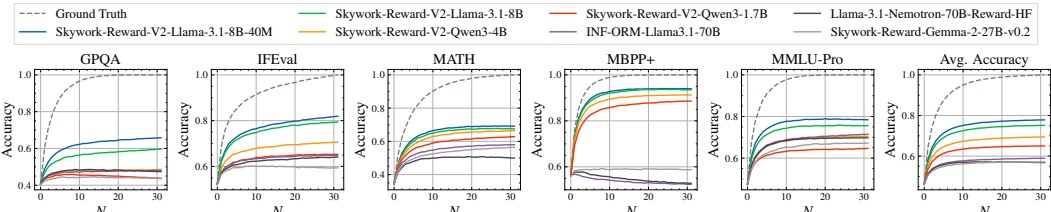

Figure 3: Best-of-N scaling curves of RMs across five tasks on PPE Correctness (Frick et al., 2024).

**Correctness preferences.** We compare our RMs with leading LLMs and reasoning models on JudgeBench (Tan et al., 2024) (Table 2) and PPE Correctness (Frick et al., 2024). While our RMs underperform state-of-the-art reasoning models on average, they outperform all leading models on knowledge tasks by a significant margin. Notably, Skywork-Reward-V2-Llama-3.2-3B achieves math performance equivalent to o3-mini (high), while Skywork-Reward-V2-Llama-3.1-8B outperforms o3-mini (high) in this category.

**Best-of-N accuracy and scaling.** We evaluate our RMs on the BoN splits from RMB (Zhou et al., 2024) and PPE Correctness Preference (Frick et al., 2024). As shown in Table 3, our RMs demonstrate strong Best-of-N (BoN) capability in both helpfulness and harmlessness. All eight RMs outperform GPT-4o, the previous state-of-the-art, by a margin of up to 20 points. We further present BoN curves for five challenging tasks in PPE Correctness in Figure 3. Skywork-Reward-V2-Llama-3.1-8B and Skywork-Reward-V2-Llama-3.1-8B-40M show superior scaling, outperforming all other models evaluated. Among all BoN scaling curves, all Skywork-Reward-V2 variants exhibit positive scaling (i.e., performance continues to improve as $N$ increases), except for our 1.7B variant in GPQA. We further confirm the strengths of Skywork-Reward-V2 in BoN capability on RewardBench v2 (Malik et al., 2025) (Table 5), which requires precise best-of-N selection globally across the dataset.

**Resistance against style biases.** Using RM-Bench (Liu et al., 2024c), we assess the ability of reward models to judge substance under varying stylistic differences between chosen and rejected responses. As shown in Table 4, most baseline models exhibit significant performance gaps across the three stylistic conditions, indicating high sensitivity to such biases. This is particularly evident for INF-ORM-Llama3.1-70B, with a gap of 36 points between Normal and Hard accuracy. In contrast, Skywork-Reward-V2 models outperform all baselines — not only in absolute scores across all

| Model | Easy | Normal | Hard | Avg. |
|---|---|---|---|---|
| Skywork-Reward-Llama-3.1-8B-v0.2 | 70.5 | 74.2 | 49.3 | 64.7 |
| Skywork-Reward-Gemma-2-27B-v0.2 | 88.9 | 71.9 | 42.1 | 67.6 |
| ArmoRM-Llama3-8B-v0.1 | 80.4 | 71.5 | 55.8 | 69.2 |
| Nemotron-340B-Reward | 81.0 | 71.4 | 56.1 | 69.5 |
| LDL-Reward-Gemma-2-27B-v0.1 | 92.4 | 75.2 | 45.5 | 71.0 |
| Llama-3-OffsetBias-RM-8B | 83.9 | 73.2 | 56.9 | 71.3 |
| Internlm2-20b-reward | 79.4 | 74.2 | 62.8 | 72.1 |
| Llama-3.1-Nemotron-70B | 92.2 | 76.5 | 47.8 | 72.2 |
| INF-ORM-Llama3.1-70B | 92.1 | 80.0 | 54.0 | 75.4 |
| Skywork-Reward-V2-Qwen3-0.6B | 90.3 | 78.0 | 54.8 | 74.4 |
| Skywork-Reward-V2-Qwen3-1.7B | 93.0 | 83.4 | 59.7 | 78.7 |
| Skywork-Reward-V2-Qwen3-4B | 92.1 | 84.7 | 67.9 | 81.6 |
| Skywork-Reward-V2-Qwen3-8B | 91.9 | 85.7 | 70.1 | 82.6 |
| Skywork-Reward-V2-Llama-3.2-1B | 91.3 | 79.9 | 57.8 | 76.3 |
| Skywork-Reward-V2-Llama-3.2-3B | 91.5 | 84.1 | 67.8 | 81.1 |
| Skywork-Reward-V2-Llama-3.1-8B | 97.0 | 95.0 | 86.5 | 92.8 |
| Skywork-Reward-V2-Llama-3.1-8B-40M | 97.6 | 96.9 | 93.5 | 96.0 |

Table 4: Fine-grained difficulty-level scores on RM-Bench (Liu et al., 2024c).

| Model | Factuality | Precise IF | Math | Safety | Focus | Ties | Avg. |
|---|---|---|---|---|---|---|---|
| Skywork-Reward-Llama-3.1-8B | 69.9 | 42.5 | 62.8 | 93.3 | 96.2 | 74.1 | 73.1 |
| URM-LLama-3.1-8B | 68.8 | 45.0 | 63.9 | 91.8 | 97.6 | 76.5 | 73.9 |
| Skywork-Reward-Gemma-2-27B-v0.2 | 76.7 | 37.5 | 67.2 | 96.9 | 91.7 | 81.8 | 75.3 |
| claude-3-7-sonnet-20250219 | 73.3 | 54.4 | 75.0 | 90.3 | 92.1 | 67.2 | 75.4 |
| Skywork-Reward-Gemma-2-27B | 73.7 | 40.3 | 70.5 | 94.2 | 93.2 | 82.6 | 75.8 |
| llama-3.1-70B-Instruct-RM-RB2 | 81.3 | 41.9 | 69.9 | 88.4 | 86.5 | 88.3 | 76.0 |
| INF-ORM-Llama3.1-70B | 74.1 | 41.9 | 69.9 | 96.4 | 90.3 | 86.2 | 76.5 |
| claude-opus-4-20250514 | 82.7 | 41.9 | 74.9 | 89.5 | 86.2 | 83.7 | 76.5 |
| QRM-Gemma-2-27B | 78.5 | 37.2 | 69.9 | 95.8 | 95.4 | 83.2 | 76.7 |
| gemini-2.5-flash-preview-04-17 | 65.7 | 55.3 | 81.1 | 90.9 | 86.7 | 83.4 | 77.2 |
| LMUnit-llama3.1-70b | 84.6 | 48.8 | 71.6 | 90.7 | 97.0 | 90.6 | 80.5 |
| LMUnit-qwen2.5-72b | 87.2 | 54.4 | 72.7 | 91.3 | 96.8 | 90.1 | 82.1 |
| Skywork-Reward-V2-Qwen3-0.6B | 58.2 | 40.0 | 71.6 | 84.4 | 79.4 | 34.0 | 61.3 |
| Skywork-Reward-V2-Qwen3-1.7B | 65.8 | 45.0 | 72.7 | 89.1 | 88.5 | 48.7 | 68.3 |
| Skywork-Reward-V2-Qwen3-4B | 77.3 | 46.2 | 73.2 | 92.2 | 96.6 | 67.4 | 75.5 |
| Skywork-Reward-V2-Qwen3-8B | 79.8 | 49.1 | 77.0 | 94.0 | 96.4 | 72.9 | 78.2 |
| Skywork-Reward-V2-Llama-3.2-1B | 60.9 | 45.6 | 59.6 | 87.3 | 89.3 | 43.1 | 64.3 |
| Skywork-Reward-V2-Llama-3.2-3B | 76.2 | 45.6 | 69.4 | 93.1 | 96.0 | 67.7 | 74.7 |
| Skywork-Reward-V2-Llama-3.1-8B | 84.6 | 66.2 | 77.6 | 96.7 | 98.4 | 81.2 | 84.1 |
| Skywork-Reward-V2-Llama-3.1-8B-40M | 87.9 | 67.8 | 83.1 | 97.3 | 99.2 | 83.9 | 86.5 |

Table 5: Comparison of our RMs with the top 12 RMs on RewardBench v2 (Malik et al., 2025).

three categories but also in maintaining much smaller performance differences. We also observe a rapidly shrinking gap as model size increases. These results suggest that training on SynPref-40M leads to more debiased representations of preferences.

**Superiority in advanced capabilities.** On RewardBench v2, Skywork-Reward-V2 further demonstrates superior capability in precise instruction following, including assessing whether a model's response adheres to specific instructions in the prompt. Notably, all existing reward models score below 50 in this category. In contrast, Skywork-Reward-V2-Llama-3.1-8B and Skywork-Reward-V2-Llama-3.1-8B-40M outperform strong proprietary models like Claude-3.7-Sonnet and Gemini-2.5-Flash-Preview-04-17, and generative reward models that utilize rubrics (Saad-Falcon et al., 2024), through learning a pure representation of preferences. We also observe a significant increase in the Factuality score, likely due to the volume of SynPref-40M and the richness of the information it contains.

## 4.3 ABLATION STUDIES ON DATA QUANTITY AND QUALITY

We further examine the effect of data quantity and quality through performance trends across our pipeline, based on an early version of SynPref-40M with only 16 million preference pairs.

**Preference data scaling does not hold for uncurated data. (quality and quantity)** In the left plot of Figure 4, we show that increasing the amount of uncurated data results in minimal performance gains. During Stage 2, training on an additional 12 million preference pairs fails to surpass the performance of the initial seed model. In contrast, with curated data, we observe consistent performance improvements as more data is added, with the most significant gains occurring in Stage 2 — where the largest volume of curated data is introduced. The "Filtered" curve represents preference pairs that pass both human labeling and LLM annotation in terms of agreement — from the Stage 1 perspective, this is simply the concatenation of $\mathcal{D}_{\text{gold}}$ and $\mathcal{D}_{\text{silver}}$ at each specific iteration. The "Corrected" curve includes the "Filtered" subset plus the subset of preference pairs that pass neither human labeling nor LLM annotation, but with their preference labels flipped (i.e., chosen-rejected swapped). This latter subset corresponds to data where either humans or LLMs consider the rejected response to be better. Each point on the curves represents a reward model trained with all curated preference pairs accumulated up to that iteration (cumulative from iterations 1 through N). Notably, this result partially aligns with findings in concurrent work (Wang et al., 2025a), which specifically demonstrates that subjective preference learning does not exhibit scaling behavior, whereas objective preferences do.

**Data curation enables preference "correction." (quality)** We further demonstrate that our data curation process not only selects high-quality data for training but also identifies low-quality or "incorrect" preferences, which are placed in a discarded pool during training. By "recycling" this discarded data — simply flipping the chosen and rejected responses — we achieve consistent performance gains across all stages and iterations, as illustrated by the orange curve in Figure 4. As a result, Skywork-Reward-V2-Llama-3.1-8B-40M benefits from the inclusion of preference data even with flipped chosen-rejected responses.

**Training on 1.8% of a 16M mixture outperforms previous SOTA open RM (70B) at the 8B scale. (quality and quantity)** In the right plot of Figure 4, we report the average RM score across six benchmarks (excluding RewardBench v2 (Malik et al., 2025), which had not been released at the

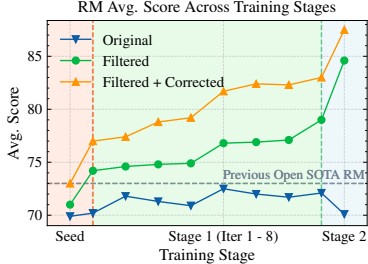 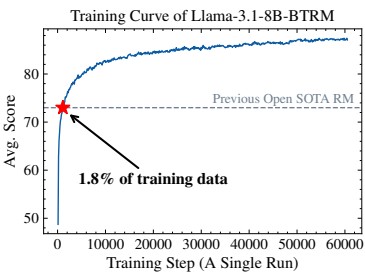

Figure 4: **Left:** Reward model score progress throughout the entire curation pipeline, including three data ablations: original data, filtered data, and filtered data with corrected preference pairs, based on an early version of SynPref-40M. **Right:** The average score of the final training run of a preliminary version of Skywork-Reward-V2-Llama-3.1-8B. The Avg. Score indicates the averaged RM score across all benchmarks considered except RewardBench v2.

time) during training. Using only 1.8% (roughly 290K samples) of the full training set surpasses the previous SOTA. This underscores that our data mixture excels not only in scale but also in quality.

## 4.4 ABLATION STUDIES ON ANNOTATION METHOD

We ablate key pipeline components, focusing on iteration 1 of Stage 1 for controlled experiments (full-pipeline ablation is infeasible due to its recursive nature and long annotation intervals).

### 4.4.1 PIPELINE-LEVEL ABLATIONS

**Setup.** We begin with the filtered seed dataset and examine five settings: (1) direct training on unverified data (i.e., no curation), (2) simple LLM curation only, (3) both human and LLM curation, and (4) incorporating adaptively retrieved examples into LLM curation. These components collectively represent one iteration of Stage 1 in Figure 2. Note that the text labels in Figure 5 describe the *change* between two consecutive settings rather than the settings themselves. Bar 1 represents training on seed data only (the baseline RM we start with). Bar 2 corresponds to seed data plus randomly sampled unverified preference pairs from $\mathcal{D}'_{un}$ (a randomly sampled subset from $\mathcal{D}_{un}$), representing no curation. Bar 3 uses seed data plus $\mathcal{D}'_{un}$ filtered by pure LLM annotation using ensemble

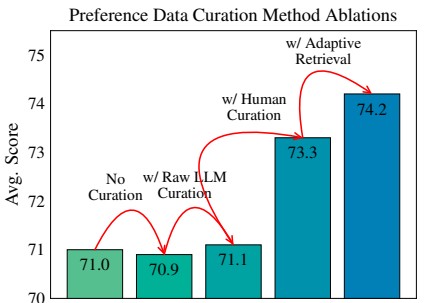

Figure 5: Ablations over different curation variants.

aggregation with self-consistency. Bar 4 employs seed data plus $\mathcal{D}'_{un}$ filtered by human annotation. Finally, Bar 5 uses seed data plus $\mathcal{D}'_{un}$ curated with our full recipe (human-guided LLM annotation with adaptive retrieval). Bar 2 has more data than Bar 1 because it includes randomly sampled pairs, while Bars 3, 4, and 5 have less data than Bar 2 because they are filtered by LLM and/or human annotation.

**Finding 1: Simple LLM curation barely improves RM quality.** As shown in Figure 5, simple LLM curation increases the final RM score by only 0.1 — potentially within the error margin of optimization randomness. Given that much in-the-wild preference data is synthetically labeled (Cui et al., 2023; Dong et al., 2024a; Lambert et al., 2024) by LLMs, this result aligns with our findings in Figure 4, where scaling uncurated preference data yields negligible gains. A potential factor may be the limited capabilities or annotation quality of the LLM judges used in our study (Ye et al., 2024; Chen et al., 2024).

**Finding 2: Human curation is crucial to data quality.** From Figure 5, we observe that the largest improvement comes from human curation, with a relative gain of 2.3 points over the seed RM baseline. This highlights the need for scalable methods of collecting human preference data and showcases the strength of our approach, which requires only a modest amount of human annotation. In Section 4.4.2, we further analyze which component of the human curation pipeline contributes most significantly.

**Finding 3: Adaptive retrieval boosts LLM curation quality.** Given access to human-curated gold data, adding similar gold examples to the LLM annotation prompt improves RM quality. This technique results in a 0.9-point gain compared to raw LLM annotation in the human curation variant. While the improvement is smaller than with direct human curation, this method is simple, scalable, and incurs minimal overhead, making it an attractive tool for enhancing LLM annotation.

### 4.4.2 HUMAN ANNOTATION ABLATIONS

**Setup.** We now focus specifically on the most impactful component: human curation. We evaluate three variants: (1) raw human curation, where annotators are shown only the conversation history and two responses, (2) human curation with LLM-generated preference attributes, and (3) human curation following our full annotation protocol (i.e., with external tools such as search engines and frontier LLMs). To control for memorization, the same

| Method | Avg. Score |
|---|---|
| **Seed RM** | 71.0 |
| w/ Raw Curation | 71.4 (+0.4) |
| w/ Pref Attributes | 72.1 (+1.1) |
| w/ Verification Protocol | **74.2 (+3.2)** |

Table 6: RM scores on three human annotation setups.

annotators label three distinct subsets of preference data sampled with similar distributions over task category, objectivity, and controversiality. Before running the ablation, we train reward models on each of the three subsets and confirm they yield similar final performance-within a maximum 0.6-point difference. This reduces the influence of intrinsic data quality as a confounding factor, ensuring controlled experiments. All other components remain unchanged from our final method. As in Section 4.4.1, we begin from an RM trained on the filtered seed data.

**Human annotation with additional information and tools boosts annotation quality.** As shown in Table 6, all forms of human curation improve the quality of the seed RM. Raw annotation based solely on the conversation and two responses results in a 0.4-point gain. Adding preference attributes (task category, objectivity, controversiality, desired attributes, and annotation guidelines) yields a larger 1.1-point gain. Incorporating our full annotation protocol — including access to external tools — leads to the best final performance, with a 3.2-point improvement, validating the effectiveness of our human curation process.

### 4.5 ADDITIONAL EXPERIMENTS

We provide additional experiments in the appendix: our curated mixture outperforms all existing preference mixtures (Section H.1), downstream RLHF and human evaluation (Section H.2), generalization across LLM backbones (Section H.3), and Stage 2 filtering effectiveness (Section H.4).

## 5 CONCLUSION

In this work, we introduce SynPref-40M, a preference data mixture comprising 40 million preference pairs (26 million curated), and Skywork-Reward-V2, a series of eight state-of-the-art reward models designed for versatility across a wide range of tasks. SynPref-40M is constructed through a two-stage curation pipeline that synergistically combines human supervision for quality with human-guided LLM judges for scalability. Using this preference data mixture, we present the Skywork-Reward-V2 series — a collection of eight strong reward models ranging from 0.6B to 8B parameters. Across seven major reward model benchmarks, models in the Skywork-Reward-V2 series achieve state-of-the-art performance, demonstrating strong capabilities in capturing general human preferences, objective correctness, resistance to style biases, safety, and best-of-N scaling. Our small 1.7B variant surpasses the best existing 70B reward model on average, while our 8B variant ranks first on all seven benchmarks among all open reward models. We also conduct extensive ablation studies on both the data and the curation method to validate the effectiveness of our approach. We believe this work advances open reward models and, more broadly, RLHF research, representing a significant step forward that will accelerate open progress in the field.

## ACKNOWLEDGMENTS

We thank our colleagues at Skywork AI, Kunlun Inc. for providing computational resources and for their valuable feedback throughout this project.

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

# A  RELATED WORK

**Preference data annotation.** Traditional preference data annotation relies heavily on human annotators (Liu et al., 2020; Stiennon et al., 2020; Ouyang et al., 2022; Bai et al., 2022a; Hurst et al., 2024; Touvron et al., 2023a;b), which is both costly and inefficient — and sometimes even noisy (Daniels-Koch & Freedman, 2022). To improve scalability, recent work — now collectively referred to as RLAIF (Bai et al., 2022b) — has proposed various forms of automatic annotation using strong LLMs (Bai et al., 2022b; Lee et al., 2023; Burns et al., 2023; Cui et al., 2023; Guo et al., 2024; Yuan et al., 2024b; Prasad et al., 2024; Pace et al., 2024; Lambert et al., 2024; He et al., 2025a), in some cases even outperforming human annotators (Gilardi et al., 2023; Ding et al., 2022). Our approach combines the strengths of both paradigms: we enhance human annotation using external tools and frontier LLMs, while also guiding LLM-based annotation with human-verified labels. Among related work, the most relevant are Kim et al. (2025) and He et al. (2024). Kim et al. (2025) leverages a small set of human-labeled seed data to iteratively refine an LLM policy via self-improvement (Rafailov et al., 2023); in contrast, we iteratively incorporate gold human preference labels to augment LLM annotation within a structured data curation framework. He et al. (2024) employs an iterative process that pseudo-labels unlabeled preference pairs and retains only high-confidence examples, without human annotators. Our work bridges the gap between human and LLM-based annotation by integrating them into a principled and scalable framework, enabling high-quality preference data at scale. In addition, our approach to human verification via preference attributes is similar to LMUnit (Saad-Falcon et al., 2024), which decomposes requirements based on context and conducts automatic "unit tests" on assistant responses using LLMs.

**The paradigm of reward models.** The reward model paradigm has evolved rapidly. Initially based on the Bradley-Terry (BT) model (Bradley & Terry, 1952; Liu et al., 2020; Stiennon et al., 2020; Ouyang et al., 2022; Bai et al., 2022a), early reward models were trained to maximize the score difference between pairwise responses. During inference, these models produce a scalar score indicating the relative quality of a response compared to alternatives given the same prompt. Later, RewardBench (Lambert et al., 2025) introduced the first taxonomy of reward models, categorizing them into (1) sequence classifiers, (2) direct preference optimization (DPO) models with implicit rewards, (3) generative models, and (4) custom classifiers. Most BT-based models fall under the sequence classifier category, while generative models primarily include LLM-as-a-Judge approaches. DPO models, by contrast, rely on implicit rewards derived from the DPO objective (Rafailov et al., 2023). This taxonomy was further elaborated in Liu et al. (2024b) and has since been adopted by subsequent works (Zhong et al., 2025; Zang et al., 2025; Wang et al., 2025c). With the emergence of generative reward models (Liu et al., 2025; Chen et al., 2025c; Saha et al., 2025; Guo et al., 2025), Liu et al. (2025) proposed a new categorization based on the form of reward generation and scoring patterns, highlighting differences in input flexibility and inference-time scalability. The reward generation forms include scalar, semi-scalar, and generative outputs, while scoring patterns are categorized as pointwise or pairwise. Beyond these major paradigms, Sun et al. (2024) introduces an alternative approach that trains reward models using an order consistency objective. This reframes reward modeling as a binary classification task and has been shown to outperform the Bradley-Terry model in the presence of annotation noise.

**Strong open reward models and preference datasets.** At the time of writing, there are already 166 reward models on the RewardBench v1 leaderboard (Lambert et al., 2025), most of which are open-weight. The top-ranking models are primarily from the Skywork-Reward series (Liu et al., 2024b) and their derivatives, trained using either the same base models (Dorka, 2024; Lou et al., 2024) or datasets (Yang et al., 2024b; Shiwen et al., 2024; Lou et al., 2024; Zhang et al., 2024b; Yang et al., 2024c). Their training data primarily consist of unfiltered human preferences and automatically curated synthetic data (Liu et al., 2024b). Another line of high-performing reward models includes FsfairX and ArmoRM (Dong et al., 2024b; 2023; Wang et al., 2024a), trained on Preference 700K (Dong et al., 2024b), a dataset composed of preference data aggregated from eight diverse sources. The ArmoRM variant extends FsfairX with a multi-dimensional reward head, enabling it to generate reward signals for fine-grained aspects of response quality. The InternLM2-Reward series (Cai et al., 2024) also presents strong models across different sizes, trained on a large-scale collection of 2.4 million closed-source preference pairs, with a focus on both English and Chinese data. Recently, the release of RewardBench v2 (Malik et al., 2025) introduced a set of seven reward models trained on various Llama-3.1 checkpoints (i.e., different sizes and base models). Among these, the 70B variant is one of the top-performing models on the benchmark. Right before our release, we noticed

two generative reward models from the LMUnit series (Saad-Falcon et al., 2024) that topped the RewardBench v2 leaderboard. These models use rubrics as unit tests, which are much more robust than reward models based on discriminative classifiers. Their strength is further reflected by their high scores in Factuality and Ties categories. Our reward models leverage both the Skywork-Reward dataset and Preference 700K in the Seed and Stage 1 phases, respectively — forming the foundation for improvements in later stages.

## B  LIMITATIONS

Human preferences are inherently diverse and often conflicting, especially for prompts without a single correct answer. Even when ground-truth answers exist, individuals may differ in their preferences based on factors such as writing style, tone, level of detail, or the relative weighting of helpfulness versus harmlessness. A single reward model may not fully capture this complexity and may inherently favor certain response types over others. Future work could explore personalized reward models or context-dependent training paradigms to better reflect the multifaceted nature of human preference.

Our observation regarding performance improvements from re-annotated discarded data is purely empirical. Due to budget constraints, we did not conduct further verification to rigorously assess this pool. As a result, the re-annotated data may include noisy preferences or judgments that are not broadly representative or that fall outside the scope of current evaluation benchmarks. A thorough investigation of this flipped pool is left for future work.

Meanwhile, we would like to clarify that not all discarded preference pairs are incorrect or useless. Since our pipeline still uses LLMs and trained reward models to filter data, which is not fully interpretable, biases and modeling errors are inherently unavoidable. Studying why and how examples are removed during the process, as well as their actual usefulness for reward modeling and RLHF, could be a valuable research direction.

Our annotation protocol differs in implementation from most existing approaches, where human annotators provide their own preferences. In contrast, our protocol is more constrained: it instructs annotators to follow predefined desired attributes and annotation guidelines for each sample. While this structured approach promotes consistency, it also reduces flexibility and may not fully capture minority preferences. This limitation arises because, for certain subjective preferences, it is often infeasible to determine which response is better-even on a relative scale.

Finally, the success of our approach relies heavily on human annotation; we did not observe satisfactory results from fully automatic curation alone. This raises the question of whether current-generation LLMs are capable of supporting high-quality, fully automatic data labeling. Due to inference costs and API limitations, we were unable to scale automatic curation to the latest frontier models with strong reasoning capabilities. We consider this a promising direction for future exploration, particularly given the central role these LLMs already play in supporting human annotation within our pipeline.

## C  REWARD MODEL BENCHMARKS AND EVALUATION RESULTS

### C.1  REWARD MODEL BENCHMARKS

**RewardBench.** RewardBench (Lambert et al., 2025) is the first benchmark released for evaluating reward models. It includes 2,985 evaluation samples from 23 data sources, categorized into four main groups: chat, chat-hard, safety, and reasoning. The evaluation uses pairwise comparison accuracy, where a reward model generates scores for both the chosen and rejected responses. A prediction is correct if the score for the chosen response exceeds that of the rejected one. Final accuracy is computed as a weighted average within each category and then averaged across categories. A noted limitation is that the chosen-rejected pairs are constructed using semi-automatic methods and manually validated, though the authors do not detail the validation process. They also acknowledge potential spurious correlations in the reasoning subsets and the absence of correlation analysis between RewardBench scores and downstream performance.

**PPE Preference and Correctness.** PPE (Frick et al., 2024) includes two datasets for evaluating reward models: PPE Preference and PPE Correctness. PPE Preference consists of 16K human-labeled preference pairs from Chatbot Arena, targeting real human preferences. PPE Correctness

is derived from challenging benchmarks with ground-truth answers, allowing direct verification of preference pairs. Included benchmarks are MATH (Hendrycks et al., 2021), MBPP (Austin et al., 2021), MMLU-Pro (Wang et al., 2024c), IFEVAL (Zhou et al., 2023), and GPQA (Rein et al., 2024). Each prompt yields 32 LLM responses, enabling both pairwise and best-of-N evaluations. The authors demonstrate a strong correlation between PPE scores and downstream RLHF performance, making it a reliable benchmark for real-world reward model evaluation.

**RMB.** RMB (Zhou et al., 2024) is a comprehensive benchmark covering 49 real-world task categories under both helpfulness and harmlessness. Like PPE Correctness, it supports pairwise and best-of-N evaluations. Preference pairs are generated synthetically, with GPT-4 providing pointwise ratings based on query-specific principles. Human verification is used to ensure dataset quality. RMB shows strong positive correlation with downstream performance across several benchmarks.

**RM-Bench.** Unlike other benchmarks that focus on general preference evaluation, RM-Bench (Liu et al., 2024c) specifically tests a reward model's ability to discern nuanced response differences and resist style biases. It includes four categories: Chat, Math, Code, and Safety. Prompts are sourced from benchmarks such as AlpacaEval (Li et al., 2023), HumanEval (Chen et al., 2021), MATH (Hendrycks et al., 2021), and XSTest (Röttger et al., 2023). Response pairs are minimally different (e.g., word-level changes introducing factual errors) and generated with controlled style. RM-Bench defines three difficulty levels: (1) easy, where style mismatches may mislead the model; (2) normal, with matched stylistic quality; and (3) hard, where content is decisive despite a stylistically superior distractor.

**JudgeBench.** JudgeBench (Tan et al., 2024) is a correctness-focused benchmark originally designed for LLM-based judges. Due to its pairwise format, it naturally supports pointwise reward model evaluation. It includes subsets such as MMLU-Pro (Wang et al., 2024c) (knowledge), LiveBench (White et al., 2024) (math and reasoning), and LiveCodeBench (Jain et al., 2024).

**RewardBench v2.** RewardBench v2 (Malik et al., 2025) is the second version of the original RewardBench (Lambert et al., 2025), featuring substantially more difficult and realistic evaluation data. It assembles new human-generated prompts (in contrast to prior benchmarks which reuse downstream prompts), grouped into diverse and multi-skill classification tasks. On average, existing reward models score around 20 points lower on RewardBench v2 compared to its predecessor. RewardBench v2 also shows stronger correlation with downstream performance — both during RL fine-tuning (e.g., PPO) and best-of-N inference sampling — compared to earlier RM benchmarks.

## C.2 FULL EVALUATION RESULTS

In Table 7, we present the complete evaluation results for all the reward models considered. We categorize them into Bradley-Terry reward models, LLM-as-Judges, and the new paradigm of generative reward models (Liu et al., 2025). Across all seven benchmarks discussed in the main body of the paper, our reward models trained on SynPref-40M outperform all previous models on average.

| Model | RewardBench | RewardBench v2 | PPE Pref | PPE Corr | RMB | RM-Bench | JudgeBench | Avg. |
|---|---|---|---|---|---|---|---|---|
| *Bradley-Terry Reward Models* | | | | | | | | |
| GRM-gemma2-2B-rewardmodel-ft (Yang et al., 2024c) | 88.5 | 59.7 | 59.7 | 58.5 | 68.0 | 66.2 | 63.5 | 66.3 |
| RM-Mistral-7B (Dong et al., 2023) | 80.9 | 59.6 | 61.8 | 56.4 | 66.6 | 66.9 | 62.1 | 64.9 |
| Eurus-RM-7b (Yuan et al., 2024a) | 83.3 | 58.1 | 59.6 | 60.5 | 65.5 | 69.0 | 58.4 | 64.9 |
| BTRM_Qwen2_7b_0613 | 83.6 | 57.4 | 61.8 | 58.4 | 61.5 | 69.4 | 63.8 | 65.1 |
| Internlm2-7b-reward (Cai et al., 2024) | 87.6 | 53.4 | 62.1 | 60.0 | 67.1 | 67.1 | 59.4 | 65.2 |
| FsfairX-LLaMA3-RM-v0.1 (Dong et al., 2023) | 84.7 | 62.9 | 63.1 | 61.1 | 70.2 | 70.5 | 59.9 | 67.5 |
| internlm2-1_8b-reward (Cai et al., 2024) | 82.0 | 39.0 | 57.3 | 53.6 | 54.2 | 66.2 | 59.0 | 58.8 |
| ArmoRM-Llama3-8B-v0.1 (Wang et al., 2024a) | 90.4 | 66.5 | 60.6 | 60.6 | 64.6 | 69.2 | 59.7 | 67.4 |
| Llama-3-OffsetBias-RM-8B (Park et al., 2024) | 89.0 | 64.8 | 59.2 | 64.1 | 57.8 | 71.3 | 63.5 | 67.1 |
| QRM-Llama3.1-8B-v2 (Dorka, 2024) | 93.1 | 70.7 | 57.2 | 60.3 | 61.1 | 72.5 | 62.6 | 68.2 |
| GRM-llama3-8B-distill (Yang et al., 2024c) | 86.2 | 58.9 | 63.2 | 62.8 | 68.8 | 70.3 | 63.3 | 67.6 |
| QRM-Llama3.1-8B (Dorka, 2024) | 93.1 | 70.7 | 60.6 | 60.5 | 64.7 | 72.8 | 63.8 | 69.5 |
| GRM-Llama3-8B-rewardmodel-ft (Yang et al., 2024c) | 91.5 | 67.7 | 62.1 | 60.0 | 70.2 | 69.9 | 62.3 | 69.1 |
| URM-LLaMa-3.1-8B (Lou et al., 2024) | 92.9 | 73.9 | 60.2 | 60.4 | 65.7 | 72.0 | 64.1 | 69.9 |
| Skywork-Reward-Llama-3.1-8B (Liu et al., 2024b) | 92.5 | 73.1 | 62.1 | 60.3 | 69.2 | 71.8 | 62.0 | 70.1 |
| Skywork-Reward-Llama-3.1-8B-v0.2 (Liu et al., 2024b) | 93.1 | 71.8 | 62.2 | 62.5 | 66.6 | 64.7 | 62.9 | 69.1 |
| Starling-RM-34B (Zhu et al., 2023) | 80.8 | 45.5 | 62.8 | 57.5 | 72.0 | 67.1 | 63.8 | 64.2 |
| QRM-Gemma-2-27B (Dorka, 2024) | 94.4 | 76.7 | 52.3 | 54.8 | 53.4 | 65.9 | 57.5 | 65.0 |
| Internlm2-20b-reward (Cai et al., 2024) | 90.2 | 56.3 | 61.0 | 63.0 | 62.9 | 72.1 | 64.3 | 67.1 |
| Skywork-Reward-Gemma-2-27B (Liu et al., 2024b) | 93.8 | 75.8 | 60.3 | 60.1 | 69.5 | 68.5 | 65.2 | 70.5 |
| Llama-3.1-Nemotron-70B (Wang et al., 2024d) | 93.9 | 76.7 | 64.2 | 63.2 | 64.9 | 72.2 | 65.8 | 71.6 |
| LDL-Reward-Gemma-2-27B-v0.1 | 95.0 | 72.5 | 62.4 | 63.9 | 67.9 | 71.0 | 64.2 | 71.0 |
| Skywork-Reward-Gemma-2-27B-v0.2 (Liu et al., 2024b) | 94.3 | 75.3 | 63.6 | 61.9 | 69.4 | 67.6 | 66.5 | 71.2 |
| INF-ORM-Llama3.1-70B (Yang et al., 2024b) | 95.1 | 76.5 | 64.2 | 64.4 | 70.5 | 75.4 | 70.2 | 73.8 |
| *LLM-as-a-Judge & Generative Reward Models* | | | | | | | | |
| GPT-4o (Hurst et al., 2024) | 86.7 | 64.9 | 67.7 | 67.1 | 73.8 | 73.1 | 59.8 | 70.4 |
| Claude-3.5-Sonnet (Anthropic, 2024) | 84.2 | 64.7 | 67.3 | 69.2 | 70.6 | 74.5 | 64.8 | 70.8 |
| DeepSeek-GRM-27B (Liu et al., 2025) | 88.5 | - | 65.3 | 64.0 | 69.0 | - | - | - |
| DeepSeek-GRM-27B (w/ MetaRM) (Liu et al., 2025) | 90.4 | - | 67.2 | 63.2 | 70.3 | - | - | - |
| RM-R1-Qwen-Instruct-32B (Chen et al., 2025c) | 92.9 | - | - | - | 73.0 | 79.1 | - | - |
| RM-R1-DeepSeek-Distill-Qwen-32B (Chen et al., 2025c) | 90.9 | - | - | - | 69.8 | 83.9 | - | - |
| EvalPlanner (Llama-3.1-70B) (Saha et al., 2025) | 93.9 | - | - | - | - | 80.0 | 50.9 | - |
| EvalPlanner (Llama-3.3-70B) (Saha et al., 2025) | 93.8 | - | - | - | - | 82.1 | 56.6 | - |
| J1-Llama-8B (Whitehouse et al., 2025) | 85.7 | - | 60.3 | 59.2 | - | 73.4 | 42.0 | - |
| J1-Llama-8B (Maj@32) (Whitehouse et al., 2025) | - | - | 60.6 | 61.9 | - | - | - | - |
| J1-Llama-70B (Whitehouse et al., 2025) | 93.3 | - | 66.3 | 72.9 | - | 82.7 | 60.0 | - |
| J1-Llama-70B (Maj@32) (Whitehouse et al., 2025) | - | - | 67.0 | 73.7 | - | - | - | - |
| *Our Reward Models* | | | | | | | | |
| Skywork-Reward-V2-Qwen3-0.6B | 85.2 | 61.3 | 65.3 | 68.3 | 74.5 | 74.4 | 67.6 | 70.9 |
| Skywork-Reward-V2-Qwen3-1.7B | 90.3 | 68.3 | 67.6 | 70.5 | 78.1 | 78.7 | 72.9 | 75.2 |
| Skywork-Reward-V2-Qwen3-4B | 93.4 | 75.5 | 69.5 | 74.7 | 80.6 | 81.6 | 69.5 | 77.8 |
| Skywork-Reward-V2-Qwen3-8B | 93.7 | 78.2 | 70.6 | 75.1 | 81.2 | 82.6 | 73.4 | 79.3 |
| Skywork-Reward-V2-Llama-3.2-1B | 89.9 | 64.3 | 66.6 | 67.4 | 76.7 | 76.3 | 65.0 | 72.3 |
| Skywork-Reward-V2-Llama-3.2-3B | 93.0 | 74.7 | 69.1 | 72.1 | 80.5 | 81.1 | 69.2 | 77.1 |
| Skywork-Reward-V2-Llama-3.1-8B | 96.4 | 84.1 | 77.3 | 83.4 | 86.4 | 92.8 | 80.0 | 85.8 |
| Skywork-Reward-V2-Llama-3.1-8B-40M | **97.8** | **86.5** | **79.8** | **87.2** | **89.3** | **96.0** | **83.4** | **88.6** |

Table 7: Open reward model performance on seven reward model benchmarks.

# D DATASET PROCESSING DETAILS

## D.1 PRE-PROCESSING, DEDUPLICATION, AND DECONTAMINATION

For pre-processing, we perform a simple structural check to remove preference pairs in which either the chosen or rejected response contains `None` as content. This ensures valid formatting of the conversation.

To eliminate potential duplicates within or across datasets, we perform global deduplication across all available data sources at the time. Specifically, for each chosen-rejected pair, we represent the sample using the tuple (`conversation_history`, `chosen_response`, `rejected_response`) and discard any duplicates. The conversation history includes all prior user and assistant turns, while the chosen and rejected responses refer to the assistant's final turn.

To ensure decontamination from benchmark data, we remove any instances that share at least one 13-gram overlap with a (first-turn) prompt from any of the evaluation benchmarks. For this, we employ a decontamination script previously used to clean preference datasets against RewardBench data[2].

---

[2]`https://gist.github.com/natolambert/1aed306000c13e0e8c5bc17c1a5dd300`

**Extended decontamination validation.** To further guarantee decontamination effectiveness, we performed additional strict checks beyond the initial 13-gram filter. We expanded the n-gram window to range from 5-grams to 13-grams and applied matching not only on prompts but on the full (`prompt`, `chosen_response`, `rejected_response`) triples. We then used a frontier LLM (Qwen3-235B-A22B-Instruct-2507) as a judge to filter out false positives — common phrases that trigger n-gram matches but do not represent true contamination. This two-step process confirmed that none of the newly identified samples constituted actual contamination. Even in the initial 13-gram decontamination round, approximately 23% of flagged samples were false positives that were correctly excluded after manual inspection. For $\mathcal{D}_{\text{gold}}$ specifically, we enforce a strict zero-overlap policy to eliminate any risk of contamination from human-verified training data.

**Dataset licensing.** The SynPref-40M dataset is aggregated from publicly available preference pairs whose licenses and terms of use permit redistribution. To ensure legal compliance, we maintain: (1) a license file specifying the terms, (2) an attribution file documenting the source of each dataset along with their respective licenses where applicable, and (3) a usage file clarifying downstream compliance requirements.

## D.2 DATASET COMPOSITION AND CHARACTERISTICS

The SynPref-40M dataset consists solely of publicly available preference pairs, primarily collected from over 40 diverse sources on Hugging Face, provided that the licenses and terms of use of those sources permit redistribution. The majority of the collected samples (over 99% of the full dataset) contain synthetic prompts and/or responses generated by different LLMs, and the remainder are written by real humans, based on the original descriptions of the source datasets.

**Task category distribution.** During Stage 1 LLM labeling, we adopted the task categorization from Xu et al. (2024) and obtained the following prompt-wise distribution across the dataset. The distribution shows that information seeking and coding tasks dominate, accounting for over 70% of the data, followed by advice-seeking, mathematics, and creative writing tasks. This diversity ensures broad coverage of different types of user queries and preferences.

| Category | Percentage |
|---|---|
| Information seeking | 40.4% |
| Coding & Debugging | 32.7% |
| Advice seeking | 10.9% |
| Math | 7.5% |
| Creative writing | 4.1% |
| Reasoning | 2.2% |
| Planning | 0.62% |
| Data analysis | 0.44% |
| Editing | 0.41% |
| Role playing | 0.37% |
| Brainstorming | 0.16% |
| Other | 0.09% |

Table 8: Task category distribution in SynPref-40M.

**Controversiality and objectivity.** Based on the labels from Stage 1, we also estimate the distribution of controversiality level and objectivity of the preference pairs. The majority of preference pairs (73.8%) have low controversiality, indicating relatively clear preference signals. Similarly, 74.8% of the pairs are classified as objective, which aligns well with the high proportion of information seeking, coding, and math tasks.

| Controversiality | Percentage |
|---|---|
| Low | 73.8% |
| Medium | 20.0% |
| High | 6.2% |

Table 9: Controversiality distribution.

| Objectivity | Percentage |
|---|---|
| Objective | 74.8% |
| Subjective | 25.2% |

Table 10: Objectivity distribution.

**Language distribution.** Over 95% of the preference pairs are in English. Roughly 2.5% are in Chinese, and the remainder consists of other languages (e.g., German, French, Spanish).

**Rationale for collecting in-the-wild data.** There are three main reasons we collect purely open preference pairs rather than generating responses from scratch. First, we initially attempted to re-sample responses from collected prompts while discarding the original responses, but found that this approach does not scale well given our budget constraints. Second, our goal is to develop a robust pipeline that can handle realistic challenges — we want the pipeline to be able to process diverse and large quantities of non-uniform, potentially low-quality data without making strong assumptions about the source or type. Third, we aim to demonstrate that significant value has been hidden in existing in-the-wild preference data; it simply has not been properly extracted previously.

**Data allocation across stages.** Before starting each iteration of the curation pipeline, we strictly perform deduplication and decontamination as described above. Throughout the pipeline, we allocated 80K pairs in the seed stage, fewer than 1M pairs in Stage 1, and the remainder in Stage 2.

### D.3 PRIVACY AND PII ANALYSIS

To ensure responsible data practices, we conducted a comprehensive analysis to identify and mitigate potential privacy risks in the SynPref-40M dataset.

**PII detection methodology.** We performed a single-pass scan over all (`prompt`, `chosen_response`, `rejected_response`) triples using an LLM-as-a-Judge to identify potential personally identifiable information (PII). The LLM was instructed to extract specific PII instances and assign a confidence score ranging from 0 to 3, indicating the sensitivity level and whether the information should be removed. This initial scan flagged approximately 0.07% of the dataset ( 28K samples) with a positive score, as detailed in Table 11.

| Sensitivity Score | Number of Samples |
| --- | --- |
| 0 (minimal concern) | 17,960 |
| 1 (low sensitivity) | 5,498 |
| 2 (moderate sensitivity) | 3,261 |
| 3 (high sensitivity) | 1,478 |
| **Total** | **28,197** |

Table 11: Distribution of PII sensitivity scores across the dataset.

**Human verification and characterization.** We conducted human verification on a random sample from each sensitivity level. None of the samples with scores 0 or 1 were confirmed as genuine PII by human annotators. For samples with scores 2 or 3, we performed a second-pass analysis using GPT-4o-mini. This analysis revealed that most flagged samples (¿93%) contain indirect identifiers such as age, gender, birthdates, demographics, or fictional usernames, rather than direct personal identifiers that could compromise individual privacy.

**Sensitivity analysis of reward models to PII.** To verify that the trained reward models do not exhibit sensitivity to PII, we constructed test pairs where PII was either removed or swapped with neutral placeholders. As shown in Table 12, the reward models maintain near-perfect accuracy (approaching 100%) on these modified pairs, indicating they are not learning spurious correlations based on PII and instead focus on substantive preference signals.

| Sensitivity Score | PII Removed | PII Swapped |
| --- | --- | --- |
| 2 (moderate) | 100.0% | 99.85% |
| 3 (high) | 99.86% | 99.46% |

Table 12: Reward model accuracy on preference pairs with PII removed or swapped, demonstrating insensitivity to PII presence.

These results demonstrate that SynPref-40M contains minimal genuine PII, primarily consists of indirect identifiers in synthetic contexts, and that the trained reward models do not rely on such information for preference judgments.

### D.4 Handling intransitivity and conflicting preferences

Human preferences are often intransitive and context-dependent, as observed in recent work (Duan et al., 2024). Rather than assuming global transitivity in the raw data, our pipeline is designed to identify and localize inconsistent preference regions, including intransitive cycles and near ties, before they dominate training. We use a transitive Bradley–Terry (BT) model as a smooth surrogate that approximates a noisy, partially intransitive preference graph.

**Where intransitivity arises.** Intransitive cycles (e.g., $A \succ B$, $B \succ C$, but $C \succ A$) typically emerge as inconsistent clusters of pairwise labels over similar prompts and responses. Common scenarios include near ties between stylistically different but substantively similar answers, subjective tasks with multiple defensible "best" answers, or conflicting preferences from different annotator groups.

**Quality control mechanisms.** Our pipeline addresses intransitivity through three complementary mechanisms:

1. **Stage 1 metadata isolates "risky" regions.** Every pair in the unverified pool receives preference attributes from LLMs: task category, objectivity, controversiality, desired attributes, and annotation guideline. This stratification identifies objective/low-controversial versus subjective/high-controversial regions, where intransitivity is more common. Internal analysis shows roughly 75% of pairs are objective and 74% are low controversial, with the remaining quarter concentrated in more subjective, contentious tasks where cycles and label conflicts cluster.
2. **Error-driven adaptive retrieval focuses on "unstable" regions.** In Stage 1, we repeatedly train an RM, evaluate it on human-verified gold data, and use error-driven adaptive retrieval to pull in new examples similar (in prompt + attribute space) to misclassified or low-confidence pairs. This concentrates labeling effort where the current BT model finds the pairwise graph hard to linearize, empirically corresponding to regions with local intransitivity, near ties, or subtle spurious correlations.
3. **Stage 2 dual-RM consistency filtering targets contradictory signals.** Stage 2 introduces a consistency filter: we train a gold RM on cumulative human-verified samples and use it together with the Stage-1 best RM to decide which in-the-wild pairs to keep or flip. We retain pairs whose original chosen/rejected labels agree with the gold RM and either the Stage-1 best RM or the LLM judges. This serves as a consistency check over local preference subgraphs: if the raw annotation induces cycles that contradict the human-aligned gold RM, such edges are corrected or down-weighted.

**Empirical evidence of consistency improvement.** The human agreement analysis in Table 14 demonstrates that our hybrid approach (LLM + human + adaptive retrieval) achieves higher agreement (93% objective, 84% subjective) than pure human annotation (81%, 76%) or pure LLM annotation (75%, 63%). Additionally, as shown in Table 19, our dual-RM filtering achieves 86% agreement on kept pairs and 92% on flipped pairs, indicating that the pipeline actively resolves conflicting local preferences rather than amplifying cycles.

**Relationship to explicitly intransitive models.** Our pipeline is agnostic to the downstream RM parameterization. The same curated data and consistency filters can be used to train generalized intransitive preference models (Duan et al., 2024) rather than strict BT models. We chose BT primarily for comparability and simplicity, as it remains the dominant choice in open RM work. Integrating intransitive preference models with SynPref-40M is an interesting future direction.

## E Annotation details

### E.1 LLM preference attributes labeling

Before the verification and annotation process, our preference attributes are generated from a combination of API and local models, including Claude-3.5-Sonnet (Anthropic, 2024), GPT-4o (Hurst et al., 2024), o4-mini (OpenAI, 2025), DeepSeek-V3 (Liu et al., 2024a), Llama-3.3-70B-Instruct (Grattafiori et al., 2024), Llama-3.1-70B-Instruct (Grattafiori et al., 2024), Qwen2.5-72B-Instruct (Yang et al., 2024a), Qwen3-32B (Yang et al., 2025), Qwen3-14B (Yang et al., 2025).

**Quality analysis of preference attributes.** To evaluate the quality of the LLM-generated preference attributes, we randomly sampled 500 items and performed a human quality check. For category, controversiality level, and objectivity, human annotators provided a binary judgment (yes or no)

on whether the correct label was provided. For desired attributes and annotation guideline, human annotators rated the quality on a scale of 1 to 5. The results, shown in Table 13, demonstrate that the category and controversiality level are generally well-aligned with the LLM annotations, with over 90% agreement rate. Objectivity is slightly lower at 87.4%. The desired attributes and annotation guideline are generally well-annotated with ratings above 4.0, which is consistent with the LLM annotations.

| Attribute | Agreement Rate / Average Rating |
|---|---|
| Category | 96.2% |
| Controversiality Level | 90.6% |
| Objectivity | 87.4% |
| Desired Attributes | 4.52 (rating) |
| Annotation Guideline | 4.03 (rating) |

Table 13: Quality assessment of LLM-generated preference attributes via human verification on a random sample of 500 items.

**Examples of preference attributes.** Due to space constraints, we provide examples of the preference attributes and their corresponding preference pairs in the supplementary materials. These examples illustrate the 5-tuple structure (task category, objectivity, controversiality, desired attributes, and annotation guideline) for various types of preference pairs in our dataset.

### E.2 HUMAN VERIFICATION AND ANNOTATION PROTOCOL

**LLM usage during human verification.** During our human annotation pipeline, annotators are allowed to use external tools such as a search engine or frontier LLMs, including GPT-4o (Hurst et al., 2024), all o-series models (Jaech et al., 2024; OpenAI, 2025), Gemini (2.0 Flash, 2.5 Flash, 2.5 Pro) (Team et al., 2023), Claude (3.5-Sonnet and 3.7-Sonnet) (Anthropic, 2024), Grok (2 and 3) (xAI, 2024; 2025), DeepSeek-V3 (Liu et al., 2024a), and DeepSeek-R1 (Guo et al., 2025). However, we design strict guidelines for using these tools, and specify detailed guidelines for different tasks, objectivity type, and controversiality level.

**Batched pre-verification.** To speed up annotation, we prioritize preference pairs labeled as "objective," and pre-verify them with LLMs in a batched way. Specifically, we use a set of query templates embedded with the conversation with a single response, and the LLM provides a final judgment of correct or incorrect. During human annotation, the annotator still reads the response in general. This drastically improves efficiency, because annotators no longer need to interact with the LLM for verification and annotation.

**Verification and annotation priority.** During our initial inspection of the data pool, we found that many preference data pairs contain extremely ambiguous preference signals, even with the provided attributes. In some conversations, the user asks vague questions, and both assistant responses seek clarification, differing only in phrasing. As a result, we use the preference attributes to prioritize annotating objective and low-controversiality preferences. If an annotator cannot determine the preference relationship from the pairwise data, we skip the LLM annotation process and discard it.

In the later stages of the project, we recognized that a potentially more valuable approach is to use LLMs to label the differences between the two candidate responses and prioritize the annotation of these samples. However, due to the high inference cost associated with millions of samples, we will continue with our original approach in this work and leave this for future research.

### E.3 LLM-AS-A-JUDGE LABELING

For LLM-as-a-Judge labeling, we employ the same verification and annotation guideline used by human annotators but remove all sentences mentioning LLM usage and the use of web search for those without web browsing capabilities. Toward the end of the guideline, we provide at most eight concatenated pairwise instances and their corresponding preference attributes, and the target pairwise instance for labeling.

**LLMs used for annotation.** The list of LLMs used for preference-aware annotation includes both chat-based models and advanced agentic LLMs. In the initial stages, we used models including Claude-3.5-Sonnet, GPT-4o, o4-mini, DeepSeek-V3, Llama-3.3-70B-Instruct, Llama-3.1-70B-

Instruct, Qwen2.5-72B-Instruct, Qwen3-32B, and Qwen3-14B. In the final stage, we incorporated more advanced agentic LLMs to target more complex tasks, including Deep Research, Gemini 2.5 Pro (with search), Claude-4-Sonnet (with search), Grok-4 (with search), GLM-4.5, Kimi-K2, and GPT-4.1. We also replaced weaker general chat-based models below 70B with the latest frontier open models at the time, such as Qwen3-235B-A22B, DeepSeek-V3.1, and GPT-OSS-120B.

**Annotation prompts.** Due to space constraints in the main paper, we provide the complete annotation prompts used for preference-aware LLM labeling in the supplementary materials. These prompts include the preference attributes (task category, objectivity, controversiality, desired attributes, and annotation guideline), the retrieved few-shot examples from $\mathcal{D}_{\text{gold}}$, and the target preference pair to be labeled.

### E.4 Lessons learned from verifying and annotating human preferences in-the-wild

While we initially include in-house human annotators, the authors also participate in the later stages of the annotation process. Here, we share the lessons we learned and some discussions from our annotation efforts.

1. **LLMs can effectively automate certain types of annotation.** For conversations involving reasoning tasks such as math problems or coding questions, LLMs are more efficient and reliable than human annotators. Human annotators may not be experts in all types of math and coding problems. We emphasize using cutting-edge models for this purpose, particularly those with advanced reasoning capabilities. Our inspection of early annotations reveals that different LLMs exhibit strong annotation bias. This bias arises from various sources, including scenarios with multiple or no ground-truth answers, which are highly context-dependent, and those requiring external knowledge. We believe this issue can be mitigated in the era of agents (Luo et al., 2025a), given their ability to perform web searches or conduct deeper research. In practice, we find that over 90% of objective preference pairs involve mostly information seeking (e.g., fact-checking), math/code problems, or general/specialized domain knowledge (e.g., literature review, movie plot summary). While humans alone can certainly perform well on these tasks, LLMs with tools are far more efficient and, in most cases, less expensive regarding annotation costs. We also observe prompts whose chosen-rejected relationship cannot be easily determined by humans who are not domain experts. In such cases, LLMs with tools are the only practical solution (assuming we do not pay for expensive expert annotation services).

   To further validate the effectiveness of human-guided LLM curation, we conducted an agreement rate analysis on $\mathcal{D}_{\text{gold}}$ — our human-verified dataset — by re-annotating these samples under different annotation variants. We divided $\mathcal{D}_{\text{gold}}$ into objective preferences (non-controversial labels) and subjective preferences (potentially controversial labels), and measured how each annotation method agrees with the original human labels. As shown in Table 14, the agreement rate increases significantly from pure LLM annotation to LLM + human curation, and further to LLM + human + adaptively retrieved samples. Notably, even pure human annotation (without LLM assistance for fact-checking or domain knowledge) performs worse than LLM + human curation, particularly on objective tasks where LLMs with tools excel at verification.

| Annotation Variant | Objective | Subjective |
|---|---|---|
| Pure LLM | 75% | 63% |
| LLM + human curation | 87% | 77% |
| LLM + human + adaptive retrieval | 93% | 84% |
| Pure human (no LLM tools) | 81% | 76% |

Table 14: Agreement rate between different annotation variants and the original human labels in $\mathcal{D}_{\text{gold}}$. The results demonstrate that human-guided LLM curation with adaptive retrieval achieves the highest agreement, outperforming both pure LLM and pure human annotation.

2. **Human preferences are complicated, even for humans.** During annotation, we consistently encountered preference pairs that were ambiguous, subjective, or context-dependent — making it difficult even for trained annotators to confidently determine which response was better. Factors like subtle tone differences, varying expectations around informativeness or safety, and individual annotator biases introduced uncertainty into this process. This highlights a key challenge in reward modeling: even with structured annotation protocols and strong preference attributes, some preferences are inherently ill-defined or non-universal. This problem stems from

the concept of human preferences and their diversity. It also raises the question of whether a single reward model can effectively capture this diverse range of human preferences. In our initial experiments, we observed that (1) if we mix pairs with opposite preferences, reward models tend to learn spurious correlations (e.g., pure text format), and (2) if we pre-generate preference specifications (D'Oosterlinck et al., 2025) for the preference pairs, not only does pure LLM annotation quality improve (when we provide such additional information), but we can also leverage this information to avoid reward models learning spurious correlations (by avoiding conflicting preferences). However, we chose not to include these findings in the main paper as they were under-studied and might overcomplicate this work. We consider them valuable directions for future exploration.

3. **Learning clear and aligned preferences significantly enhances reward models.** Our experiments demonstrate that when reward models are trained on preference data that is well-structured, verified, and guided by clear annotation protocols, their performance improves substantially across all evaluation benchmarks. We hypothesize that this may be due to the significantly higher requirement for constructing preference pairs in the benchmark dataset. While we do not have quantitative results, reviewing the preference pairs presented in multiple test sets reveals a strong preference signal. This also highlights a fundamental flaw in the design of today's preference data: although the response pairs are provided, the actual difference between them - the core indication of preference - is ignored. This raises concerns about what reward models, or any other types of models that provide a reward signal, actually learn from underspecified responses.

### E.5 Annotator information

The annotation process involved fewer than 20 trained annotators across both the seed stage and Stage 1. In the seed stage, one author participated, while additional authors contributed during Stage 1. These author contributions were voluntary, intended to expedite progress, and were not compensated. Each preference pair annotation required between a minimum of 10 seconds and a maximum of 5 minutes to complete. On average, the team generated approximately 2,000 to 3,000 annotations per week. The cost of producing each annotated preference pair was estimated to range between \$0.10 and \$0.70. Overall, the full annotation effort extended over a period of roughly nine months.

## F  Practical guidance for cost-performance budgeting

Given the importance of planning preference data curation under resource constraints, we provide practical guidance on achieving target reward model performance within a specified cost budget.

### F.1 What our scaling results reveal about cost

Section 4.3 presents data quantity and quality ablations based on an earlier 16M-pair mixture. Two key findings emerge: (1) uncurated scaling fails — adding 12M uncurated preference pairs on top of the seed set yields almost no performance gain, and (2) curated scaling succeeds — with Stage 1 + Stage 2 curation, performance improves steadily, with the largest gains in Stage 2.

Most importantly for budgeting, we find that training on just 1.8% of the 16M curated mixture ( 290K pairs) already surpasses the previous open SOTA 70B RM at the 8B scale. This is a clear "quality beats volume" statement: carefully curated hundreds of thousands of pairs suffice to beat prior state-of-the-art, without requiring tens of millions of new, expensive human labels.

In our pipeline, fewer than 500K pairs pass through full human verification in Stage 1, with the remaining tens of millions curated automatically in Stage 2. Human effort comprises only a couple percent of the final training pool but drives most performance gains.

### F.2 A simple budgeting recipe

We outline a practical framework for planning preference data curation under a cost budget:

1. **Define target performance.** Let the desired average score across the six main benchmarks (excluding RewardBench v2) be $S_{\text{target}}$. Our scaling curve in Figure 4 shows the relationship between fraction of curated data and average RM score.
2. **Estimate required curated pairs.** From Figure 4, practitioners can read off a conservative fraction $f$ of the full curated mixture needed to reach $S_{\text{target}}$. For example, $f \approx 0.018$ ( 290K

pairs) already exceeds previous open SOTA at 8B. Higher targets correspond to larger $f$, but with diminishing returns.

3. **Decompose costs by stage.** Our pipeline separates labeling into three cost regimes:
   - **Gold human labels** (Stage 1, $\mathcal{D}_{\text{gold}}$): cost $c_{\text{H}}$ per pair, high leverage, used to train the gold RM and seed attribute generation and LLM judges.
   - **Silver human-guided LLM labels** (Stage 1, $\mathcal{D}_{\text{silver}}$): cost $c_{\text{L1}}$ per pair (LLM inference, often 1–2 orders of magnitude cheaper than full human annotation), guided by human-labeled neighbors.
   - **Large-scale consistency curation** (Stage 2): cost $c_{\text{L2}}$ per pair (mainly RM inference + occasional LLM annotation), used to scale from hundreds of thousands to tens of millions of pairs.

   Total curation cost is approximately:

   $$B \approx c_{\text{H}} \cdot |D_{\text{gold}}| + c_{\text{L1}} \cdot |D_{\text{silver}}| + c_{\text{L2}} \cdot |D_{\text{Stage2}}|$$

   In practice, $c_{\text{H}} \gg c_{\text{L1}} \geq c_{\text{L2}}$ and $|D_{\text{gold}}| \ll |D_{\text{Stage2}}|$, so the gold set dominates quality while automatic stages dominate quantity.

4. **Allocation strategy.** Given a dollar budget $B_{\max}$, allocate a gold budget $B_{\text{H}} \leq B_{\max}$ to determine $|D_{\text{gold}}|$. Our results suggest that roughly $O(10^5)$ carefully-selected gold pairs suffice to train strong gold and Stage-1 RMs. Allocate the remaining budget $B_{\max} - B_{\text{H}}$ to scaling Stage-2 curation, trading off total curated volume versus LLM quality (e.g., using cheaper versus more capable judges).

5. **Validation and stopping criteria.** Monitor (1) RM benchmark scores as in Figure 4, and (2) downstream BoN curves (e.g., PPE Correctness and RMB) where we observe monotonic scaling with $N$. Once incremental gains per additional curated million pairs fall below a user-defined threshold (e.g., ¡0.3 points on average benchmark score), it is reasonable to stop spending.

## F.3 WORKED EXAMPLE

Suppose a practitioner has a budget of \$50,000 and seeks to match or exceed the previous SOTA 70B RM using an 8B model. Based on our findings:

- **Target:** $S_{\text{target}} \approx 73.5$ (INF-ORM-Llama3.1-70B average score across six benchmarks).
- **Required data:** $f \approx 0.018$ of a 16M mixture, roughly 290K curated pairs.
- **Cost breakdown** (assuming $c_{\text{H}} = \$0.50$, $c_{\text{L1}} = \$0.05$, $c_{\text{L2}} = \$0.01$):
  - Allocate 100K pairs to $\mathcal{D}_{\text{gold}}$: 100K × \$0.50 = \$50K.
  - This exhausts the budget, but the gold set alone is sufficient to train a strong gold RM.
  - In practice, one could reduce $|D_{\text{gold}}|$ to 50K (\$25K), then allocate the remaining \$25K to Stage 1 silver labels (500K pairs at \$0.05) and Stage 2 curation (several million pairs at \$0.01).
- **Outcome:** With careful allocation across stages, \$50K can produce a curated dataset exceeding 1M pairs, sufficient to reach or exceed the target performance.

This example illustrates how practitioners can use our stage-wise cost decomposition and scaling curves to plan curation budgets systematically, balancing human verification quality with automated scalability.

## F.4 RELATIONSHIP TO MECHANISM DESIGN APPROACHES

Recent work on mechanism design for preference learning (Zhang et al., 2024a) explores which comparisons to query and how to structure incentives to extract maximal information from limited human comparisons. We view mechanism-design approaches and our human–AI curation pipeline as operating at complementary layers of the RLHF stack:

- **Mechanism design (Zhang et al., 2024a):** focuses on which comparisons to ask for and how to structure incentives/queries to extract maximal information from limited human comparisons, often under a stylized model where one controls the querying process but not a large, messy in-the-wild pool.
- **SynPref-40M:** focuses on how to extract value from already-existing, heterogeneous, synthetically labeled preference data, using human guidance and LLM+RM consistency to filter, correct, and scale.

We see at least two concrete points of contact that highlight potential for integration:

1. **Mechanism design as an inner loop in Stage 1.** Our error-driven adaptive retrieval already behaves like a "targeted querying mechanism" over the unverified pool. Future work could replace simple similarity-based retrieval with a mechanism-design–inspired query selection rule,

e.g., selecting pairs that maximally reduce posterior uncertainty in a generalized pairwise model under a fixed human budget.

2. **Using mechanism-design insights to set budgets and stopping rules.** Zhang et al.'s framework (Zhang et al., 2024a) suggests principled criteria for which pairwise comparisons are most information-efficient. Combined with our Stage-wise cost decomposition, this could inform better allocation of gold human labels across task types and controversiality levels, focusing human effort where marginal information gain is highest.

In summary, while mechanism design operates at the algorithmic level to optimize query selection, our approach operates at the data-centric level to handle realistic, large-scale, heterogeneous preference data. These complementary perspectives can be integrated: mechanism design can guide which samples to prioritize for human annotation within our pipeline, while our curation mechanisms can handle the messy realities of in-the-wild data that mechanism design typically abstracts away.

## G TRAINING DETAILS AND HYPERPARAMETERS

We primarily adhere to the hyperparameter choices outlined in Lambert et al. (2024) and Wang et al. (2025a). During the development phase, we adjust the learning rates according to the model size, using 1e-6 for all 8B models and 4e-6 for all other sizes. All models are trained with a global batch size of 256 and a linear learning rate decay, using a warmup schedule for only 1 epoch, with a maximum token length of 16,384. For all final training runs, we switch to a learning rate of 3e-6 and a large global batch size of 10,240 for all models, following Wang et al. (2025a), due to its faster convergence and negligible impact on performance. All models are trained using $64 \times$ H800 GPUs with DeepSpeed ZeRO Stage 1 (Rasley et al., 2020).

## H ADDITIONAL EXPERIMENTS

### H.1 EXISTING (UNCURATED) PREFERENCE DATASETS ARE INADEQUATE

To evaluate the effectiveness of the landscape of open preference datasets, we source almost all existing popular preference datasets from Hugging Face. We train a single reward model in the same way as we train ours on each of the preference dataset and the combination of all preference data. We present the full results in Table 15.

We demonstrate that none of the single preference datasets or the combination of all datasets outperform our curated mixture. Using olmo-2-0425-1b-preference-mix alone results in an average score of 69.4. In contrast, combining all datasets yields only 68.9, a decrease of 0.5 points. This further validates that preference scaling cannot be achieved by simply accumulating the number of preference pairs.

### H.2 DOWNSTREAM RLHF EVALUATION AND HUMAN EVALUATION

**Policy optimization.** Other than the preference scoring benchmarks in the main paper, we perform additional downstream RLHF training. We largely follow the setting by Chang et al. (2025), but only differ in the set of prompts. For prompts, we use a set of hard prompts that are selected both manually and automatically from our preference data pool. We evaluated policies trained using our RM versus the previous state-of-the-art RMs with similar size. We observe that the resulting policy outperforms not only policies trained by the baseline RM but also official instruct models (Table 17), indicating the RM generalizes to training-time rewards for instruction following.

**Human evaluation.** Given that most of the preference benchmarks' labels are generated either synthetically or automatically, we further perform real-human agreement assessment against our trained reward models on an internal hold-out preference benchmark. We show that reward models trained on the curated preference mixture obtain significantly higher preference agreement with humans in Table 16.

### H.3 THE EFFECTIVENESS OF THE CURATED MIXTURE ACROSS VARIOUS BACKBONES

In the main paper, we only use the Llama (Grattafiori et al., 2024) and Qwen3 (Yang et al., 2025) backbones to train our reward models. To prove that the proposed curated mixture works "universally" across different model families, we consider additional backbones from Gemma (Team et al., 2024; 2025) and Qwen2.5 (Hui et al., 2024) families. We also attach the scores from INF-ORM-

| Model | RewardBench | RewardBench2 | PPE HumanPref | PPE Correctness | RMB | RM-Bench | JudgeBench | Avg. |
|---|---|---|---|---|---|---|---|---|
| All combined | 79.5 | 65.8 | 65.5 | 63.3 | 73.7 | 70.2 | 64.0 | 68.9 |
| allenai/olmo-2-0425-1b-preference-mix | 84.2 | 66.7 | 63.1 | 61.4 | 72.4 | 71.4 | 66.5 | 69.4 |
| allenai/olmo-2-1124-13b-preference-mix | 81.9 | 66.1 | 63.5 | 62.1 | 72.6 | 70.7 | 66.0 | 69.0 |
| RLHFlow/pair_data_v2_80K_wsafety | 84.9 | 64.4 | 66.2 | 62.6 | 66.8 | 73.5 | 63.6 | 68.9 |
| RLHFlow/UltraFeedback-preference-standard | 85.0 | 64.7 | 64.4 | 61.8 | 68.2 | 71.8 | 65.6 | 68.8 |
| allenai/llama-3.1-tulu-3-8b-preference-mixture | 82.1 | 64.8 | 63.9 | 61.4 | 72.4 | 71.1 | 65.6 | 68.7 |
| hendrydong/preference_700K | 85.6 | 64.0 | 63.6 | 62.9 | 69.1 | 72.1 | 63.5 | 68.7 |
| allenai/llama-3.1-tulu-3-405b-preference-mix | 83.1 | 63.6 | 64.6 | 61.4 | 72.2 | 71.0 | 64.9 | 68.7 |
| allenai/olmo-2-1124-7b-preference-mix | 81.6 | 65.9 | 62.9 | 62.4 | 72.8 | 71.1 | 63.5 | 68.6 |
| allenai/olmo-2-0325-32b-preference-mix | 81.6 | 63.4 | 64.4 | 62.6 | 71.8 | 71.2 | 64.5 | 68.5 |
| m-a-p/COIG-P | 83.6 | 61.1 | 62.7 | 61.9 | 74.2 | 72.8 | 61.8 | 68.3 |
| NVIDIA/HelpSteer3 | 87.2 | 65.9 | 65.5 | 59.6 | 66.6 | 70.4 | 62.7 | 68.2 |
| allenai/llama-3.1-tulu-3-70b-preference-mix | 80.2 | 63.4 | 63.8 | 61.2 | 72.9 | 70.5 | 64.6 | 68.1 |
| llm-blender/Unified-Feedback | 81.1 | 59.7 | 64.9 | 58.3 | 73.1 | 71.4 | 65.4 | 67.7 |
| BAAI/Infinity-Preference | 88.1 | 61.0 | 62.8 | 60.6 | 64.0 | 70.6 | 64.0 | 67.3 |
| allenai/tulu-2.5-preference-data | 76.7 | 55.7 | 66.6 | 60.6 | 70.4 | 71.4 | 67.9 | 67.1 |
| Magpie-Align/Magpie-Llama-3.1-Pro-DPO-100K-v0.1 | 87.7 | 59.7 | 61.7 | 60.0 | 64.6 | 72.1 | 63.3 | 67.0 |
| Magpie-Align/Magpie-Air-DPO-100K-v0.1 | 87.8 | 60.2 | 61.7 | 59.4 | 62.5 | 71.0 | 64.8 | 66.8 |
| RLHFlow/pair_data_v2_78_wo_safety | 79.2 | 61.4 | 65.8 | 63.4 | 63.4 | 64.4 | 65.7 | 66.2 |
| Magpie-Align/Magpie-Pro-DPO-100K-v0.1 | 87.4 | 58.7 | 61.6 | 59.8 | 61.7 | 70.1 | 64.4 | 66.2 |
| RLHFlow/Capybara-distibalel-Filter-standard | 84.0 | 61.3 | 60.1 | 60.4 | 59.8 | 69.7 | 64.4 | 65.7 |
| TIGER-Lab/AceCodePair-300K | 80.6 | 63.1 | 56.6 | 59.7 | 57.2 | 72.5 | 65.0 | 65.0 |
| vincentmin/eli5_rlhf | 84.4 | 58.2 | 58.7 | 62.4 | 59.3 | 68.1 | 61.9 | 64.7 |
| RLHFlow/Prometheus2-preference-standard | 86.0 | 51.0 | 60.5 | 58.5 | 63.5 | 68.3 | 58.7 | 63.8 |
| NVIDIA/HelpSteer2 | 83.7 | 56.8 | 61.5 | 55.9 | 59.8 | 66.3 | 61.1 | 63.6 |
| openbmb/UltraInteract_pair | 81.3 | 47.4 | 60.0 | 64.4 | 60.2 | 69.8 | 61.4 | 63.5 |
| allenai/wildguardmix | 80.2 | 55.1 | 54.9 | 60.1 | 61.6 | 70.4 | 58.5 | 63.0 |
| prometheus-eval/Preference-Collection | 84.2 | 49.0 | 60.3 | 56.5 | 64.6 | 64.3 | 60.2 | 62.7 |
| RLHFlow/CodeUltraFeedback-standard | 78.9 | 46.7 | 61.2 | 56.4 | 65.0 | 69.1 | 60.9 | 62.6 |
| lmarena-ai/arena-human-preference-55k | 75.0 | 54.3 | 67.1 | 64.0 | 59.0 | 55.6 | 62.8 | 62.5 |
| RLHFlow/HelpSteer-preference-standard | 78.8 | 55.8 | 56.9 | 60.5 | 55.3 | 61.4 | 63.5 | 61.7 |
| lmarena-ai/arena-human-preference-100k | 74.5 | 52.2 | 69.4 | 60.3 | 57.3 | 58.4 | 59.8 | 61.7 |
| Vezora/Code-Preference-Pairs | 78.5 | 50.6 | 58.1 | 57.3 | 57.7 | 64.9 | 63.9 | 61.6 |
| GAIR/preference-dissection | 74.4 | 52.9 | 60.9 | 61.4 | 57.7 | 56.4 | 61.9 | 60.8 |
| xinlai/Math-Step-DPO-10K | 73.8 | 52.6 | 55.1 | 58.2 | 53.4 | 67.5 | 61.0 | 60.2 |
| NCSOFT/offsetbias | 68.5 | 55.3 | 51.3 | 57.7 | 52.2 | 63.5 | 57.2 | 57.9 |
| argilla/OpenHermesPreferences | 62.6 | 45.1 | 62.5 | 53.7 | 60.9 | 51.6 | 59.4 | 56.5 |
| HuggingFaceH4/OpenHermes-2.5-preferences-v0-deduped | 65.0 | 47.1 | 60.2 | 54.6 | 57.5 | 51.9 | 54.2 | 55.8 |
| argilla/magpie-ultra-v0.1 | 68.1 | 40.0 | 57.6 | 55.4 | 52.3 | 58.6 | 56.5 | 55.5 |
| RLHFlow/HH-RLHF-Harmless-and-RedTeam-standard | 51.3 | 31.3 | 41.9 | 49.2 | 36.1 | 56.3 | 47.4 | 44.8 |

Table 15: Benchmarking the effectiveness of all existing popular preference datasets.

| RM | Agreement with human |
|---|---|
| GPT-4o | 74.3 |
| Claude-3.5-Sonnet | 72.1 |
| Skywork-Reward-V2-Qwen3-1.7B | 71.0 |
| Skywork-Reward-V2-Qwen3-4B | 75.6 |
| Skywork-Reward-V2-Llama-3.1-8B | 81.2 |

Table 16: Agreement between different reward models (RMs) and human judgment.

| Model | Method | ArenaHardv1 | ArenaHardv2 | MT-Bench | WildBench | Avg. |
|---|---|---|---|---|---|---|
| Llama-3.1-8B | Base | 6.8 | 2.0 | 52.8 | 54.9 | 29.1 |
| | +SFT | 12.6 | 3.1 | 56.8 | 60.3 | 33.2 |
| | +RL (Skywork-Reward-Llama-3-8B-v0.2) | 9.7 | 1.6 | 57.1 | 57.8 | 31.6 |
| | +RL (Skywork-Reward-Gemma-2-27B-v0.2) | 14.0 | 3.8 | 58.5 | 61.5 | 34.4 |
| | +RL (Skywork-Reward-V2-Qwen3-4B) | 18.8 | 6.0 | 62.8 | 65.0 | 38.2 |
| | +RL (Skywork-Reward-V2-Llama-3.1-8B) | 20.8 | 6.3 | 66.5 | 70.2 | 40.9 |
| | Instruct (official) | 24.9 | 5.8 | 65.7 | 64.2 | 40.2 |
| Qwen2.5-7B | Base | 16.2 | 5.6 | 63.5 | 51.8 | 34.3 |
| | +SFT | 22.1 | 9.9 | 67.3 | 60.5 | 40.0 |
| | +RL (Skywork-Reward-Llama-3-8B-v0.2) | 29.8 | 12.2 | 76.8 | 64.9 | 45.9 |
| | +RL (Skywork-Reward-Gemma-2-27B-v0.2) | 34.5 | 15.5 | 78.2 | 67.8 | 49.0 |
| | +RL (Skywork-Reward-V2-Qwen3-4B) | 35.0 | 17.9 | 79.0 | 69.0 | 50.2 |
| | +RL (Skywork-Reward-V2-Llama-3.1-8B) | 38.0 | 18.5 | 81.1 | 71.5 | 52.3 |
| | Instruct (official) | 37.9 | 17.1 | 78.8 | 70.9 | 51.2 |

Table 17: Performance comparison of Llama-3.1-8B and Qwen2.5-7B across ArenaHard, MT-Bench, and WildBench.

| Model | RewardBench | RewardBench2 | PPEHumanPref | PPECorrectness | RMB | RM-Bench | JudgeBench | Avg. |
|---|---|---|---|---|---|---|---|---|
| INF-ORM-Llama3.1-70B | 95.1 | 76.5 | 64.2 | 64.4 | 70.5 | 73.8 | 70.2 | 73.5 |
| Qwen2.5-7B | 91.7 | 67.2 | 66.4 | 73.9 | 78.3 | 79.6 | 71.1 | 75.4 |
| CIR-AMS/BTRM_Qwen2_7b_0613 | 83.2 | 57.4 | 60.0 | 63.1 | 70.2 | 72.3 | 64.5 | 67.2 |
| gemma-2-2b-it | 89.4 | 66.6 | 67.9 | 71.2 | 76.7 | 76.2 | 70.0 | 74.0 |
| Ray2333/GRM-gemma2-2B-rewardmodel-ft | 80.5 | 59.7 | 55.4 | 62.0 | 65.5 | 68.1 | 69.4 | 65.8 |
| gemma-2-9b-it | 95.0 | 78.1 | 76.9 | 82.0 | 83.9 | 86.1 | 77.9 | 82.8 |
| gemma-3-1b-it | 91.2 | 69.8 | 70.1 | 73.8 | 77.1 | 78.4 | 73.5 | 76.3 |
| gemma-3-4b | 93.7 | 71.0 | 68.9 | 73.7 | 77.1 | 79.6 | 76.0 | 77.1 |

Table 18: Comparison of models across multiple reward model and preference benchmarks.

Llama3.1-70B, the current best RM, for comparison. In Table 18, our own models, even with smaller backbones, consistently outperform this baseline. This highlights the effectiveness of our preference curation: it enables smaller models to exceed the performance of much larger ones. Additionally, for RMs based on Qwen2.5-7B-Instruct and Gemma-2-2B, we can directly compare to counterparts trained by other teams, which further demonstrates the benefit of our dataset.

### H.4 THE EFFECTIVENESS OF STAGE 2 AGREEMENT-ONLY FILTERING

We conducted a rigorous evaluation to assess whether our Stage 2 consistency-based filtering amplifies or mitigates systematic biases and spurious correlations. Specifically, we examined whether the filtering mechanism — which keeps pairs agreeing with both the best RM and gold RM, and flips pairs where there is disagreement — aligns with actual human preferences.

**Evaluation methodology.** We randomly sampled preference pairs from both the kept and flipped portions of the unverified pool, where inclusion/flipping decisions were driven by the two-RM filter mechanism described in Section 3.3. We then conducted human agreement tests to measure whether these filtering decisions aligned with human judgments: for kept pairs, humans should agree with the original labels; for flipped pairs, humans should disagree with the original labels (i.e., agree with the flipped version). We repeated this evaluation using two strong baseline reward models (Skywork-Reward-Llama-3.1-8B and Skywork-Reward-Gemma-2-27B) and their combination to test whether agreement among baseline RMs performs comparably.

| Reward model used for filtering | Keep (%) | Flip (%) |
|---|---|---|
| Skywork-Reward-Llama-3.1-8B | 69 | 57 |
| Skywork-Reward-Gemma-2-27B | 72 | 61 |
| Combined baseline RMs | 71 | 60 |
| Stage 1 Best RM | 78 | 79 |
| Stage 1 Gold RM | 84 | 88 |
| **Stage 1 Best RM + Gold RM (Ours)** | **86** | **92** |

Table 19: Human agreement rates for kept and flipped pairs under different filtering mechanisms. Higher percentages indicate better alignment with human preferences. Our dual-RM approach (Best RM + Gold RM) achieves the highest agreement for both kept and flipped pairs, demonstrating that Stage 2 filtering reduces rather than amplifies systematic biases.

**Key findings.** As shown in Table 19, baseline reward models exhibit relatively poor agreement with human judgments, with the Skywork-Reward-Llama-3.1-8B achieving only 69% agreement on kept pairs and 57% on flipped pairs. Combining the two baseline models does not yield substantial improvement (71% and 60%, respectively). In stark contrast, our Stage 1 Best RM and Gold RM each achieve much higher agreement rates, with the Best RM reaching 78% and 79%, and the Gold RM reaching 84% and 88% for kept and flipped pairs respectively. When combined, our dual-RM filtering mechanism achieves 86% agreement on kept pairs and an impressive 92% agreement on flipped pairs. These results demonstrate that our Stage 2 filtering approach effectively mitigates rather than amplifies systematic errors and spurious correlations, ensuring that the curated data more closely reflects genuine human preferences.

This analysis directly addresses concerns about potential overfitting to the gold RM's inductive biases. The high agreement rates — particularly for flipped pairs — indicate that our filtering mechanism successfully identifies preference pairs where the original labels contradict human judgment, rather than simply enforcing arbitrary model preferences or learning style biases.

## H.5 BASELINE EXPERIMENT: LLM + RM FILTERING WITHOUT HUMAN GUIDANCE

To address the question of whether the majority of performance improvements stem from LLM annotation with self-consistency rather than human-guided annotation, we conducted a critical baseline experiment. This baseline uses the best RM to filter out $p > 0.5$ pairs, then applies only LLM self-consistency annotations to the remaining data, without any human-guided few-shot examples from $\mathcal{D}_{\text{gold}}$.

We reproduced the same experiment based on the left plot of Figure 4 (from Section 4.3) but with only best RM + LLM filtering. This setup essentially takes the same preference data we accumulate in each iteration, and performs filtering directly with that specific best RM checkpoint + LLM annotation, without any other curation involving human guidance.

| Training Stage | Original | Filtered | Filtered + Corrected | LLM + Best RM | LLM + Best RM + Corr. |
|---|---|---|---|---|---|
| Seed | 70.0 | 71.0 | 73.0 | 71.0 | 70.5 |
| Iter 1 | 70.5 | 74.0 | 77.0 | 71.5 | 71.0 |
| Iter 2 | 71.5 | 74.5 | 77.5 | 72.5 | 72.0 |
| Iter 3 | 71.0 | 74.8 | 78.8 | 72.0 | 71.5 |
| Iter 4 | 71.0 | 75.0 | 79.0 | 72.2 | 72.0 |
| Iter 5 | 72.5 | 76.8 | 82.0 | 73.4 | 72.8 |
| Iter 6 | 72.0 | 77.0 | 82.2 | 73.0 | 73.2 |
| Iter 7 | 71.8 | 77.2 | 82.3 | 74.8 | 74.0 |
| Iter 8 | 72.2 | 79.0 | 83.0 | 74.2 | 74.5 |

Table 20: Comparison of reward model performance across training iterations with and without human guidance. "LLM + Best RM Filtered" corresponds to "Filtered" but with zero human annotation. "LLM + Best RM Filtered + Corrected" corresponds to "Filtered + Corrected" but with zero human annotation. The results show that LLM filtering alone plateaus around 74-75% while our full recipe with human guidance reaches 83%.

**Key findings.** While we were not able to perform Stage 2 due to time constraints and annotation costs, the results in Table 20 already demonstrate that LLM filtering alone does not outperform our recipe after only 2-3 iterations, and filtering + corrected does not show the same improvement as our full recipe. By iteration 8, our human-guided approach achieves 83%, while the LLM + Best RM baseline plateaus around 74-75%. This 8-9 point gap demonstrates the critical importance of human-guided annotation rather than purely automatic LLM-based curation.

## H.6 LLM-AS-A-JUDGE ENSEMBLE PERFORMANCE COMPARISON

To address whether ensembling all strong LLMs used in our annotation system to act as a single judge (with self-consistency) would perform comparably to our final trained RMs, we conducted an evaluation across all seven benchmarks. The LLM-as-a-Judge ensemble includes all models used throughout our annotation process, aggregated via self-consistency. Note that when running this evaluation, the number of completions performed for self-consistency for each model is not uniform, as we could not afford to perform self-consistency with a large number of completions for models like o3 due to cost constraints.

| Model | RB | RB v2 | PPE Pref | PPE Corr | RMB | RM-Bench | JudgeBench | Avg |
|---|---|---|---|---|---|---|---|---|
| Skywork-Reward-V2-Qwen3-0.6B | 85.2 | 61.3 | 65.3 | 68.3 | 74.5 | 74.4 | 67.6 | 70.9 |
| Skywork-Reward-V2-Qwen3-1.7B | 90.3 | 68.3 | 67.6 | 70.5 | 78.1 | 78.7 | 72.9 | 75.2 |
| Skywork-Reward-V2-Qwen3-4B | 93.4 | 75.5 | 69.5 | 74.7 | 80.6 | 81.6 | 69.3 | 77.8 |
| Skywork-Reward-V2-Qwen3-8B | 93.7 | 78.2 | 70.6 | 75.1 | 81.2 | 82.6 | 73.4 | 79.3 |
| Skywork-Reward-V2-Llama-3.2-1B | 89.9 | 64.3 | 66.6 | 67.4 | 76.7 | 76.4 | 65.0 | 72.3 |
| Skywork-Reward-V2-Llama-3.2-3B | 93.0 | 74.7 | 69.1 | 72.1 | 80.5 | 81.1 | 69.2 | 77.1 |
| Skywork-Reward-V2-Llama-3.1-8B | 96.4 | 84.1 | 77.3 | 83.4 | 86.4 | 92.8 | 80.0 | 85.7 |
| Skywork-Reward-V2-Llama-3.1-8B-40M | **97.8** | **86.5** | **79.8** | 87.2 | **89.3** | **96.0** | 83.4 | **88.6** |
| LLM-as-a-Judge (Agg.) | 93.9 | 83.2 | 75.7 | **89.6** | 82.6 | 89.0 | **87.8** | 86.0 |

Table 21: Performance comparison of our trained reward models with LLM-as-a-Judge ensemble aggregation. The LLM-as-a-Judge aggregation outperforms our top RM in PPE Correctness and JudgeBench, but falls behind in other benchmarks (mostly involving subjective tasks). Overall, our final RM (Skywork-Reward-V2-Llama-3.1-8B-40M) achieves 88.6 average vs. 86.0 for LLM-as-a-Judge.

**Key findings.** The results in Table 21 show that the LLM-as-a-Judge aggregation achieves competitive performance, particularly excelling on PPE Correctness (89.6%) and JudgeBench (87.8%), which focus on objective correctness and code-related tasks where strong LLMs naturally perform well. However, our final trained RM (Skywork-Reward-V2-Llama-3.1-8B-40M) outperforms the LLM ensemble on most other benchmarks, particularly on RewardBench (97.8 vs. 93.9), Reward-Bench v2 (86.5 vs. 83.2), PPE Preference (79.8 vs. 75.7), RMB (89.3 vs. 82.6), and RM-Bench (96.0 vs. 89.0). The overall average score of 88.6 for our RM vs. 86.0 for LLM-as-a-Judge demonstrates that distilling knowledge from LLMs into a trained reward model through our human-guided curation pipeline yields better overall performance, particularly on benchmarks involving subjective preferences, style resistance, and best-of-N selection.

## I    ETHICS STATEMENT

This work involves the collection and curation of large-scale preference data through human annotation, raising several ethical considerations that we address proactively. Our human annotation process involved workers who were compensated fairly according to industry standards and provided with clear guidelines and training. We ensured that annotators had access to external tools and resources to make informed judgments, and we implemented safeguards to prevent worker exploitation through reasonable workload distribution and adequate compensation.

The preference dataset created in this work captures human values and preferences that will be used to train reward models for RLHF applications. We acknowledge that human preferences can be subjective, culturally dependent, and potentially biased. To mitigate these concerns, we implemented diverse annotation protocols and quality control measures, including multiple validation stages and consistency checks. However, we recognize that our dataset may still reflect certain demographic or cultural biases present in the annotator pool and the underlying data sources.

Our reward models will be used to guide the behavior of AI systems through RLHF, potentially influencing how these systems interact with users. While our models demonstrate strong performance across safety benchmarks, we emphasize the importance of careful deployment and continued monitoring in downstream applications. We encourage users of our models to conduct thorough safety evaluations in their specific use cases and implement appropriate safeguards.

We acknowledge the computational resources required for this work and the associated environmental impact, though our focus on efficient model architectures (up to 8B parameters) helps minimize resource requirements for practitioners.

