# OpenReview forum: "Skywork-Reward-V2: Scaling Preference Data Curation via Human-AI Synergy"
_ICLR.cc/2026/Conference — ICLR 2026 Poster_

### Official Review · Reviewer_2akh · 2025-10-22

**Soundness:** 4
**Presentation:** 4
**Contribution:** 4
**Rating:** 8
**Confidence:** 4

**Summary:**

The paper introduces SynergyPref-40M, a preference dataset of 40M pairs (with 26M curated) and a two-stage human–AI curation pipeline for training Bradley–Terry reward models (0.6B–8B params). Stage 1 uses human-verified “gold” labels plus LLM-assisted “silver” labels with error-driven adaptive retrieval; Stage 2 scales automatic curation by enforcing consistency with a gold reward model and the current best RM. Trained RMs achieve state-of-the-art results on seven benchmarks (RewardBench v1/v2, PPE Preference/Correctness, RMB, RM-Bench, JudgeBench), with strong best-of-N scaling, resistance to style bias, and competitive correctness; ablations attribute gains to curation quality in addition to scale.

**Strengths:**

1. Two-stage, human-guided curation with adaptive retrieval is thoughtfully engineered and demonstrates measurable impact over raw LLM curation; Stage-2 consistency filtering is a pragmatic way to scale beyond the human budget.
2. Seven strong benchmarks, including RewardBench v2 and JudgeBench, with detailed breakdowns (style bias on RM-Bench, BoN scaling on PPE Correctness). Results are consistently SOTA for open RMs.
3. Training details are explicit; the authors plan to release data/models and provide scripts for reproduction.

**Weaknesses:**

1. While authors recognize raw LLM curation is weak and mitigate with human guidance, final labels in large parts of the mixture still depend on LLM aggregation; a more thorough error analysis comparing human vs LLM-judged segments would strengthen soundness claims.
2. The “in-the-wild” pool composition, de-duplication against evaluation sets, and licensing/PII filters are not described in depth.
3. Stage-2 selection relies on agreement with a gold RM trained on human data; without stringent de-duplication and contamination checks, this risks overfitting to the gold RM’s inductive biases and to benchmarks that overlap with the mined pool. Stronger leakage analysis (e.g., prompt/response near-duplicates) might be needed.

**Questions:**

1. How did you ensure that mined preference pairs (and their paraphrases) do not overlap with RewardBench v2, JudgeBench, etc.?
2. Can you report how often flips contradict original human labels vs synthetic labels, and whether flipped pairs increase spurious correlations (e.g., penalizing certain styles)?
3. Any results on downstream RLHF or human studies confirming that the SOTA RM ranking translates into policy improvements and user-perceived quality?

---

> ### Author Response · Authors · 2025-11-21
>
> We sincerely thank Reviewer 2akh for the thoughtful and constructive review. We are very encouraged that you found our method thoughtfully engineered and a pragmatic way to scale preference data curation, and we greatly appreciate your recognition that the reward models trained on our curated dataset achieve state-of-the-art performance among open reward models. **In the following response, we (1) thoughtfully address each of the three weaknesses you noted and (2) provide additional clarification and detail regarding your three questions.** We begin with our overarching message.
>
> ## Overall message
>
> For weakness 1, 2, and 3, we first partially address the concerns by pointing out evidence from the paper, and then provide additional details to support our claims and further resolve Reviewer 2akh's concerns.
>
> For all three questions, we have direct answers in the initial submission, and have expanded on the details to provide more insights to better answer the questions.
>
> ## Responses to weaknesses
>
> ### Weakness 1: Further analysis on the human vs. LLM-as-a-Judge labeling
>
> We want to first highlight that, while LLM curation dominates the majority of the efforts, they are **guided by the high quality golden human annotation.** In the paper, we provide evidence that pure LLM curation is not sufficient, and LLM curated with huamn annotation guidance + adaptively retrieved samples obtains the best performance.
>
> **Evidence 1 (Section 4.4)**: We ablate the curation methods via (1) no curation, (2) pure LLM curation, (3) LLM + human curation, and (4) LLM + human curation + adaptively retrieved samples. Here, (2) provides almost no gain over (1), whereas (3) and (4) (i.e., the full recipe) brings significant performance boost. Meanwhile, we also demonstrate that assistance from LLMs is also essential via controlled experiment in Table 4. This confirms that both humans and LLMs are playing crucial roles in the curation process, and missing either fails to lead to optimal reward model performance.
>
> **Evidence 2 (added during rebuttal)**: We take the existing D_gold, annotated by humans, and reannotate them the same (2), (3), and (4) annotation variants, and measure agreement rate. To further understand the source of disagreement, we divide D_gold into two parts: (1) objective preference with non-controversial labels, and (2) subjective preference with potentially controversial labels. Looking at the following table, we can conclude that the agreement rate increases significantly as we move from (2) to (3), and (3) to (4). This confirms the same finding in Evidence 1.
>
> | Annotation Variant | Agreement Rate (Objective) | Agreement Rate (Subjective) |
> |--------------------|----------------------------|------------------------------|
> | (2) Pure LLM       | 75%                        | 63%                          |
> | (3) LLM + human    | 87%                        | 77%                          |
> | (4) LLM + human + adaptively retrieved samples | 93% | 84% |
> | (5) Pure human     | 81%                        | 76%                          |
>
>
> Additionally, we provide another variant of pure human annotation (5) as a baseline, and find that it's worse than (3). This also highlights another key finding that even pure human annotation is not sufficient, especially in cases that judgement requires fact-checking or domain knowledge (which nowadays LLMs are quite good at).
>
> (See next comment)

---

> > ### Author Response · Authors · 2025-11-21
> >
> > ### Weakness 2: Dataset decontamination, licensing, and PII check
> >
> > **Dataset decontamination**
> >
> > In Appendix D.1, we described our efforts in ensuring the dataset is decontaminated. Specifically, we adopt a [decontamination script](https://gist.github.com/natolambert/1aed306000c13e0e8c5bc17c1a5dd300) (which was previously used to clean preference datasets against RewardBench). We use the script to remove all samples that share at least one 13-gram in the prompt with the complete pool of evaluation benchmark prompts.
> >
> > To further guarantee the decontamination effectiveness, we performed more strict checks (this is not in the initial submission). We first expanded the n-gram window to (5, 13), applied it on not only the prompt, but the concatenation of the (prompt, chosen response, rejected response) triples, and perform n-gram matching between the training data and evaluation benchmark data. After inspecting the results, we found that this did yield more samples, but mostly false positives (i.e., due phrases that are frequently used).
> >
> > We further use Qwen3-235B-A22B-Instruct-2507 as a judge to further filter out the false positives, and confirmed that none of the newly identified samples were true contamination. In fact, even in the initial round of decontamination via the 13-gram, roughly 23% of the samples were removed as false positives.
> >
> > **Licensing**
> >
> > The final dataset will be released under a CC BY-NC 4.0 license because it lets us share it publicly while still meeting the licensing rules of the datasets we used. We also chose not to reveal the licensing details to comply with the double-blind review process by ICLR.
> >
> > Upon dataset release, we will attach the license file, an attribution file indicating the source of each dataset (and their licenses if applicable), and a usage file clarifying downstream compliance requirements.
> >
> > **PII check and analysis**
> >
> > We first run a single pass over all (prompt, chosen response, rejected response) triples from the entire dataset, using an LLM-as-a-Judge to identify PII. We instruct the LLM-as-a-Judge to both extract the specific PII and its confidence score (with a range of 0-3, that the PII is sensitive and should be removed). This results in roughly 0.07% of the dataset (~28K samples) with a positive score.
> >
> > | Sensitivity Score | Number of Samples |
> > |-------------------|-------------------|
> > | 0                 | 17,960            |
> > | 1                 | 5,498             |
> > | 2                 | 3,261             |
> > | 3                 | 1,478             |
> > | Total             | 28,197            |
> >
> > We then ask humans to manually verify a randomly sampled pool. None of the samples with a score of 0 or 1 were identified as genuine PII by humans. We then focus exclusively on the samples with a score of 2 or 3, and perform a second pass of LLM-as-a-Judge using gpt-5-mini for further analysis. We find that most of the samples do not contain direct identifiers. Instead, most (>93%) of them contain indirect identifiers, such as age, gender, birthdates, demographics, (fake) usernames, etc.
> >
> > To ensure the trained RMs are not sensitive to the PII, we construct chosen-rejected pairs with the PII removed or swapped with a neutral placeholder, and report the accuracy of the RMs on these pairs. For all cases, the accuracy approaches 100%, indicating the RMs are not sensitive to the PII.
> >
> > | Sensitivity Score | PII Removed | PII Swapped |
> > |-------------------|-------------|-------------|
> > | 2                 | 100%        | 99.85%      |
> > | 3                 | 99.86%      | 99.46%      |
> >
> > (See next comment)

---

> > > ### Author Response · Authors · 2025-11-21
> > >
> > > ### Weakness 3: Risk of leakage/contamination
> > >
> > > Similar to our response to Weakness 2, before we curate the prompts and responses for D_gold, we have conducted a two-step process involving first (1) an n-gram model to identify prompts/responses with n-gram overlap with evaluation benchmark samples, and then (2) use LLM-as-a-Judge to further confirm if the identified matches are indeed true contamination. For D_gold, we enforce a strict zero overlap. So we can exlcude the risk of contamination.
> > >
> > > Is ther still the risk of overfitting the the gold RM's inductive bias? Possibly, but our pipeline mitigates this risk. We emphasize that, while the final reward model is trained on D_gold and is static during stage 2, **D_gold itself has been gradually evolving across all 8 iterations in stage 1.** This involves both expanding and correcting preference pairs in it, and all pairs are verified by humans (with the help of frontier LLMs in fact-checking and domain knowledge). We tend to think of D_gold as a diverse, high-quality but small-scale dataset that defines the foundational preferences we would like the final reward model to encode.
> > >
> > > ## Responses to questions
> > >
> > > > How did you ensure that mined preference pairs (and their paraphrases) do not overlap with RewardBench v2, JudgeBench, etc.?
> > >
> > > See our response to Weakness 2 above. Simply put, we first performed a two-step decontamination process that involves an n-gram model to identify prompts/responses with n-gram overlap with evaluation benchmark samples. We additionally added a second pass of LLM-as-a-judge to further confirm that the results from the n-gram model were indeed valid.
> > >
> > > > Can you report how often flips contradict original human labels vs synthetic labels, and whether flipped pairs increase spurious correlations (e.g., penalizing certain styles)?
> > >
> > > Yes. We directly measured whether our Stage 2 amplifies systematic biases or spurious correlations, using human agreement tests on kept vs. flipped pairs. **We see that our approach mitigates (rather than amplifies) such errors.**
> > >
> > > We randomly sampled pairs from both the kept and flipped portions of the unverified pool, where keeping/flipping was driven by two RM filters. We then ran human agreement tests to see if the filtering aligned with human judgments (keep -> agree; flip -> disagree). We repeated the same test with two strong baseline RMs and with their combination, to test whether "agreement" among baseline RMs does any better.
> > >
> > > For both Keep and Flip, higher means better agreement between RM's predictions and human judgments.
> > >
> > > |Reward model used| Keep (%) | Flip (%) |
> > > |-|-|-|
> > > |Skywork-Reward-Llama-3.1-8B|69|57|
> > > |Skywork-Reward-Gemma-2-27B|72|61|
> > > |Skywork-Reward-Llama-3.1-8B + Skywork-Reward-Gemma-2-27B|71|60|
> > > |Stage 1 best RM|78|79|
> > > |Stage 1 gold RM|84|88|
> > > |Stage 1 best RM + gold RM|86|92|
> > >
> > > The above We find that baselines perform poorly, and combining them does not help. In contrast, our best RM and the gold RM (trained on Phase‑1 human‑verified data) each achieve much higher agreement, with a slight additional gain when combined. This indicates reduced systematic‑error risk under our scheme.
> > >
> > > > Any results on downstream RLHF or human studies confirming that the SOTA RM ranking translates into policy improvements and user-perceived quality?
> > >
> > > Yes, we provided downstream RLHF evaluation in Appendix G.2. The experiments involves RL on different reward models, four downstream benchmarks, and comparison with the offcial instruct models. We show that the scores are comparable to official instruct models, indicating the RM generalizes to training‑time rewards in the downstream tasks.
> > >
> > > Unfortunately, we are not able to provide human studies at this time due to both budget and resource constraints. If conditions permit, we will try our best to provide similar-quality results before the camera-ready version of the paper.
> > >
> > > (See next comment)

---

> > > > ### Author Response · Authors · 2025-11-21
> > > >
> > > > To summarize, we've addressed your concerns by:
> > > >
> > > > - Providing new evidence (agreement rate analysis across annotation variants) showing that human-guided LLM curation significantly outperforms pure LLM or even pure human annotation, especially on subjective tasks.
> > > > - Detailing our decontamination process (n-gram + LLM judge), licensing plan (CC BY-NC 4.0 with full attribution), and PII analysis (0.07% flagged, mostly indirect identifiers, RMs not sensitive to PII).
> > > > - Showing through human agreement tests that our Stage 2 filtering reduces rather than amplifies systematic biases, with our dual-RM approach achieving 86% and 92% agreement on kept and flipped pairs respectively.
> > > >
> > > > We hope these clarifications demonstrate that SynergyPref-40M is built on a rigorous, quality-focused curation pipeline that addresses the practical concerns you raised. Thank you again for the constructive feedback that has helped us strengthen the paper.

---

> ### Comment · Reviewer_2akh · 2025-11-21
>
> Thanks for your detailed response. I carefully read through all of them and decided to keep my score. Good luck.

---

> > ### Author Response · Authors · 2025-11-21
> > **Thank You**
> >
> > Once again, thank you for your time and thoughtful feedback! We sincerely appreciate your efforts in helping us improve our work.

---

> > > ### Author Response · Authors · 2025-11-24
> > > **Revision summary**
> > >
> > > We have made the following changes to the paper in response to your feedback. All changes are marked in blue.
> > >
> > > ## 1. Human vs. LLM Annotation Quality (Weakness 1)
> > >
> > > Added agreement rate analysis (Appendix D.3, "Lessons learned" section) comparing annotation methods on human-verified gold dataset. Re-annotated samples using: (1) Pure LLM, (2) LLM + human curation, (3) LLM + human + adaptive retrieval (full method), (4) Pure human without LLM tools.
> > >
> > > Results: Full method achieves 93%/84% agreement (objective/subjective) vs. 75%/63% for pure LLM and 81%/76% for pure human. Shows LLMs with tools excel at fact-checking and domain tasks (>90% of objective preferences).
> > >
> > > ## 2. Dataset Decontamination (Weakness 2, Question 1)
> > >
> > > Extended decontamination validation (Appendix D.1). Beyond 13-gram filter, expanded n-gram matching to 5-13 grams on full (prompt, chosen, rejected) triples. Used Qwen3-235B to filter false positives from common phrases. Confirmed zero contamination with ~23% false positive rate. Strict zero-overlap enforced for D_gold.
> > >
> > > ## 3. Dataset Licensing and Release Plan (Weakness 2)
> > >
> > > Added licensing information (Appendix D.1). Dataset released under CC BY-NC 4.0 with three files: (1) license file, (2) attribution file with sources and licenses, (3) usage file with compliance requirements.
> > >
> > > ## 4. Privacy and PII Analysis (Weakness 2)
> > >
> > > Added PII detection and sensitivity analysis (Appendix D.2.1 with two tables). LLM scan flagged 0.07% (~28K samples) with sensitivity scores 0-3. Human verification confirmed >93% are indirect identifiers (age, gender, demographics) not real PII. Tested RMs on PII-removed/swapped pairs, achieving 99.5-100% accuracy. Shows RMs learn preference signals, not PII correlations.
> > >
> > > ## 5. Stage 2 Filtering Does Not Amplify Biases (Weakness 3, Question 2)
> > >
> > > Added human agreement evaluation of Stage 2 filtering (Appendix F.3, "Phase 2 agreement-only filtering" section). Sampled kept and flipped pairs from unverified pool, measured human agreement. Dual-RM approach (Best RM + Gold RM) achieves 86% agreement on kept pairs, 92% on flipped pairs vs. 57-61% for baseline RMs. Demonstrates filtering reduces biases by correctly identifying mislabeled pairs.
> > >
> > > ## 6. Downstream RLHF and Human Evaluation (Question 3)
> > >
> > > No changes. Existing content (Appendix F.2, formerly G.2) shows RL-trained policies using our RMs achieve scores comparable to official instruct models on ArenaHard, MT-Bench, WildBench. Human evaluation: 8B RM achieves 81.2% agreement, outperforming GPT-4o (74.3%) and Claude-3.5-Sonnet (72.1%).
> > >
> > > ## Summary of Changes
> > >
> > > - Appendix D.1: Extended decontamination validation and dataset licensing
> > > - Appendix D.2.1 (new): Privacy and PII analysis with detection methodology and RM sensitivity tests
> > > - Appendix D.3: Agreement rate analysis across annotation variants (table in "Lessons learned" section)
> > > - Appendix F.2: Downstream RLHF evaluation (already present, no changes)
> > > - Appendix F.3: Human agreement tests on Stage 2 filtering

---

### Official Review · Reviewer_J7rE · 2025-10-30

**Soundness:** 2
**Presentation:** 3
**Contribution:** 4
**Rating:** 6
**Confidence:** 3

**Summary:**

The paper presents a large-scale preference dataset (SynergyPref-40M) and a series of reward models trained on the dataset. It describes an iterative data curation pipeline merging human and LLM annotations. They show that trained reward models outperform existing models even at smaller model sizes across a wide range of benchmarks.

Overall, this is a solid paper overall, with a clear contribution. The results are very impressive, and given that the dataset and reward models are released as promised, it is a very nice resource for other researchers. The reward model results and ablations are nice. However, the paper lacks more details on the dataset composition, choice of prompts, and generations. The methodology itself is described at a relatively abstract level. The description of evaluation protocols, instructions given to annotators, etc., is limited. Also, no actual examples from the dataset are provided. I expect some improved transparency for a revised version, to ensure reproducibility of results.

**Strengths:**

+ I assume that the dataset itself will be made available as promised. Obviously, a dataset at this scale and level of curation is a very strong contribution to the field
+ The reported benchmark scores are very impressive, clearly outperforming existing reward models
+ The evaluation of the trained reward models is thorough and very comprehensive
+ I really appreciate the provided ablation studies. Evaluating the trade-offs of data curation and dataset scaling is insightful. In particular, sections 4.3 and 4.4 make a strong case for continued human data curation
+ The paper presents a nice discussion and comparison of recent reward modeling benchmarks

**Weaknesses:**

- The actual dataset generation/curation process is missing details, in particular: What is the composition of the dataset (i.e., which categories does it contain? How are the prompts+responses formulated? What is the origin of prompts + responses? Does it contain multi-lingual samples?) Also, there are just no samples or insights given about the contents of the collected dataset. I think maybe it would also be possible to provide some more details in the main paper, important information like annotator information is only given in the appendices, a summary in the main text would be appreciated
- For many of the lessons learned, I would appreciate more detailed insights. It was noted that access to tools was crucial for high-quality annotations. How did the annotators actually use them? What kind of prompts are most relevant for?

**Questions:**

- I do not follow the conclusion from Figure 1 (or at least I feel it’s too strongly worded). RewardBench still shows positive (and for most benchmarks) decently strong correlation. For example, in Figure 1-Left, I only see a single derivative model that outperforms RewardBench but is noticeably worse on the other benchmarks, so it seems more like an outlier. The general point that there is over-optimization on the reward bench is a totally valid concern, and I believe this to be the case, but i do not draw this conclusion from Figure 1. Could you give your oppinion?

---

> ### Author Response · Authors · 2025-11-21
>
> We appreciate Reviewer J7rE's careful reading of the paper and the detailed feedback on our dataset, annotation process, and interpretation of Figure 1. Your comments point out places where our initial submission did not provide enough context or nuance, rather than fundamental disagreements with our approach, and they have been very helpful in sharpening the presentation. **In the response below, we (1) give a clearer picture of the dataset's content and construction, (2) elaborate on the practical lessons and limitations of our preference annotation pipeline, and (3) revise and qualify our discussion of Figure 1 in a way that better matches the evidence you highlighted, while addressing your questions one by one.** We begin with our overarching message.
>
> ## Overall message
>
> ## Responses to weaknesses
>
> ### Weakness 1: Further details regarding the dataset content and composition
>
> We thank the reviewer for bringing this up. The major reason we were unable to provide too much details regarding the dataset in the main text was that it was stil under an internal review process at the time of submission, and some of the licenses (of the data sources) were subject to change. However, we try our best to provide as much details as we can here to address the reviewer's concerns.
>
> > What is the composition of the dataset (i.e., which categories does it contain? How are the prompts+responses formulated? What is the origin of prompts + responses? Does it contain multi-lingual samples?) Also, there are just no samples or insights given about the contents of the collected dataset.
>
> The dataset consists solely of publicly available preference pairs, primarily collected from a wide range of sources (over 40+) on Hugging Face, provided that the licenses and terms of use of those sources permit it. The majority of the collected samples (over 99% of the full dataset, by our rough estimate) contain synthetic prompts and/or responses generated by different LLMs, and the rest are written by real humans, based on the original description of the dataset. During Stage 1 LLM lableing, we adopted the task categorization from [1] and obtained the following prompt-wise distribution.
>
> | Category              | Percentage |
> |-----------------------|------------|
> | Information seeking   | 40.4%      |
> | Coding & Debugging    | 32.7%      |
> | Advice seeking        | 10.9%      |
> | Math                  | 7.5%       |
> | Creative writing      | 4.1%       |
> | Reasoning             | 2.2%       |
> | Planning              | 0.62%      |
> | Data analysis         | 0.44%      |
> | Editing               | 0.41%      |
> | Role playing          | 0.37%      |
> | Brainstorming         | 0.16%      |
> | Other                 | 0.09%      |
>
> Interestingly, we also inspected the distribution of the task category for each preference data source, and found that the distribution is very similar to the overall distribution, with some slight differences in the percentage of the task categories at the tail. Given the labels from Stage 1, we also estimate the distribution of the controversiality level and objectivity of the preference pairs.
>
> | Controversiality Level | Percentage |
> |---------|------------|
> | Low     | 73.8%      |
> | Medium  | 20.0%      |
> | High    | 6.2%       |
>
> | Objectivity   | Percentage |
> |------------|------------|
> | Objective  | 74.8%      |
> | Subjective | 25.2%   |
>
> Language-wise, over 95% of the preference pairs are in English. Roughly 2.5% of the rest are in Chinese, and the rest consists of other languages (e.g., German, French, Spanish, etc.).
>
> (See next comment)

---

> ### Author Response · Authors · 2025-11-21
>
> There are mainly three reasons we only collect purely open preference pairs. The first is that we tried to re-sample responses from only the collected prompts and discard the responses entirely, but later found that this approach does not scale well with the size of the dataset given our budget constraints. The second is that our goal is to develop a robust pipeline that can actually handle such realistic challenge. We want the pipeline to be able to handle diverse and large-quantity of non-uniform low-quality data, so we do not make any assumption about the source and/or type of the data. The third is that we want to demontrate that the value has been hidden in preference data in-the-wild. It's just that we were not able to extract it properly previously.
>
> Before starting each iteration of the curation pipeline, we strictly perform deduplication and decontamination following the method described in Appendix D.1. (We performed additional checks to further guarantee the decontamination effectiveness in our response to Reviewer 2akh below.) Throughout the pipeline, we allocated 80K pairs in seed stage, fewer than 1M pairs in stage 1, and the rest in stage 2.
>
> If the paper is ultimately accepted, we will provide the full details of the dataset's content, composition, annotation guidelines, annotation protocol, and other relevant information in the final version. And yes, we agree that providing the annotator details in earlier in the main text will make the paper more readable. We chose to postpone it to Appendix simply due to page constraints. In the revised version with more pages, we will move the annotator details to the main text.
>
> ### Weakness 2: More detailed insights about preference data annotation
>
> We are glad you are interested in learning more about the lessons learned. We assume you are referring to the lessonslearned from the preference data annotation process, which is Appendix E.4. Please correct us if we are wrong. In the following, we expand on the details of the lessons described in Appendix E.4. We tend to view the following two lessons as two parts: what we have improved and what we are still struggling with. We first discuss the former.
>
> At the time of annotation, the first lesson we learned is that **LLMs with tools are extremely powerful and efficient in labeling preference data on objective tasks**. In today's language, this lesson could be described in a more obvious way: LLMs with agentic capabilities and access to a wide range of tools can automate the labeling process for any preferene pair, whose chosen-rejected relationship can be verified within moderate effort. In practice, we rely on the objectivity / controversiality label obtained from Stage 1, and allocate the majority of the labeling effort of this type of preference pairs to either human + LLM labeling or pure LLM labeling. By our inspection, over 90% of the objective preference pairs involve mostly information seeking (e.g., fact-checking), math / code problems, or general / specialized domain knowledge (e.g., literature review, movie plot summary, etc.). While humans alone can certainly do a good job on these tasks, LLMs with tools are way more efficient and most of the time less expensive regarding annotation costs. We also observe that there are prompts whose chosen-rejected relationship cannot be easily determined by humans who are not experts in the domain. In this case, LLMs with tools are the only way to go (assuming we do not pay for expensive human annotation services).
>
> (See next comment)

---

> > ### Author Response · Authors · 2025-11-21
> >
> > The second (rather bitter) lesson is that **we still do not have a good understanding of how to label diverse, conflicting, and context-dependent human preferences (which are mostly subjective)**. This originated from our initial finding that reward model does not improve by simply scaling preference data. In Figure 6 and 8 of the paper, we show that increasing the number of preference pairs from the tens of thousands to the tens of millions results in almost no improvement. After Stage 1 labeling, we find that many pairs that share similar context have orthogonal or even opposite preferences. This naturally extends to another problem of today's preference datasets: we have a large quantity of preference pairs, but almost all of them lack the preference specification (i.e., what is preferred and why). This was previously discussed and studied in [2] (and possibly other work we are not aware of). In fact, in our preliminary experiments, we drew two (superficial) conclusions from preference data: (1) if we mix pairs with opposite preferences, the reward model tend to learn spurious correlations (e.g., pure text format), (2) if we pre-generate preference specifications for the preference pairs, not only does pure LLM annotation quality improve (if we provide such additional information), but we can also leverage this information to avoid reward models learning spurious correlations (by avoiding conflicting preferences). We chose not to include these conclusions in the paper, as they were under-studied and might overcomplicate this work. However, we consider them as valuable directions to explore in the future.
> >
> > We hope we have shared some more insights in the above two parts. If the reviewer would like to know more about specific details, please feel free to ask, and we'll be happy to provide more details in the follow-up comments.
> >
> > (See next comment)

---

> ### Author Response · Authors · 2025-11-21
>
> ## Responses to questions
>
> > I do not follow the conclusion from Figure 1 (or at least I feel it's too strongly worded). RewardBench still shows positive (and for most benchmarks) decently strong correlation. For example, in Figure 1-Left, I only see a single derivative model that outperforms RewardBench but is noticeably worse on the other benchmarks, so it seems more like an outlier. The general point that there is over-optimization on the reward bench is a totally valid concern, and I believe this to be the case, but i do not draw this conclusion from Figure 1. Could you give your oppinion?
>
> We would like to first thank the reviewer for their careful reading and for pointing this out. This is indeed our misinterpretation of our figure, which we missed in the initial submission.
>
> Yes, we totally understand the reviewer's concern here, and do agree that the conclusion is too strongly worded. First, we 100% agree with you that the correlation between RewardBench score and other benchmarks' average score is positive. This is what the left plot shows, and the calculated correlation coefficient on the right plot further confirms this.
>
> Our original intended message is this: an absolute improvement on RewardBench score does not necessarily translate to improvement on other benchmarks (and thus probably not on downstream tasks as well). If we look at the upper right plot of Figure 1 (left), we see that **as model scores on RewardBench increase from ~80 to 90+, they do not consistently lead to gains on other benchmarks (i.e., could be better, could be worse, or could be approximately the same).** Because of this, researchers and practitioners should (probably) avoid interpreting the quality of a reward model based on a single benchmark.
>
> We believe the above is a more qualified but valid statement, and will revise the wording to make it more clear and accurate. We will also remove our previous "over-optimization" claim from Section 2, as it is not self-evident from the figure, and will use the above qualitfied statement that can be directly inferred from the figure.
>
> If the reviewer has any further concerns or suggestions regarding the interpretation of the figure, we are more than happy to discuss them further. Again, thank you for catching this!
>
> ---
>
> To summarize, in response to your feedback we've:
>
> - Provided detailed dataset composition.
> - Shared practical lessons from annotation: LLMs with tools are extremely powerful for objective tasks, but we still struggle with diverse, conflicting, context-dependent subjective preferences that lack proper specification.
> - Revised our interpretation of Figure 1 to be more qualified: we'll replace the "over-optimization" claim with the more accurate observation that improvements on RewardBench from ~80 to 90+ don't consistently translate to gains on other benchmarks.
>
> We appreciate you catching the overstated conclusion about Figure 1 and hope these clarifications address your concerns about dataset transparency and the practical insights from our work. We'll include more dataset details in the main text of the revised version. Thank you for the careful review that helped us improve the presentation.
>
> ## References
>
> [1] Xu, Zhangchen, et al. "Magpie: Alignment data synthesis from scratch by prompting aligned llms with nothing." arXiv preprint arXiv:2406.08464 (2024).
>
> [2] D'Oosterlinck, Karel, et al. "Anchored preference optimization and contrastive revisions: Addressing underspecification in alignment." Transactions of the Association for Computational Linguistics 13 (2025): 442-460.

---

> ### Comment · Reviewer_J7rE · 2025-11-24
>
> I thank the authors for their clarification. Given that the suggested changes are integrated into the manuscript, i think this work will be a valuable and impactful contribution. Especially the extended details on dataset generation, ablations on curation, etc., are important for the final version.
>
> Because the authors have unfortunately not provided a revised version (at least to my understanding), i cannot attest for the integration of changes. Therefore, i would currently stick to my score, but i am generally leaning towards acceptance.

---

> ### Author Response · Authors · 2025-11-24
> **Revision completed and updated**
>
> Thank you for your positive feedback and for leaning towards acceptance! We sincerely apologize for the delay in uploading the revised manuscript.
>
> The revised PDF has now been updated. It took us additional time to carefully integrate the feedback from all four reviewers into the manuscript, but all changes are now reflected in the latest version. All revisions addressing reviewers' concerns / questions are highlighted in blue throughout the paper for easy verification. Feel free to check out the newly added content responding to other reviewers as well!
>
> Regarding the dataset details, we included them from our rebuttal in the paper and made efforts during the rebuttal period to finalize the data release. If the paper is ultimately accepted, **the full details of the dataset will be included in the camera-ready version**.
>
> We appreciate your patience and your valuable feedback, which has significantly strengthened the paper, particularly with the expanded dataset generation details and curation ablations that you highlighted as important.

---

> ### Author Response · Authors · 2025-11-24
> **Revision summary**
>
> We have made the following changes to the paper in response to your feedback. All changes are marked in blue.
>
> ## 1. Figure 1 Interpretation Revision (Section 2)
>
> Toned down "over-optimization" claim. RewardBench shows positive correlation with other benchmarks, so original wording was too strong.
>
> - Removed "over-optimization" language throughout Section 2
> - Replaced with: "improvements on RewardBench from ~80 to 90+ do not consistently translate to gains on other benchmarks (may improve, worsen, or remain approximately the same)"
> - Updated section headings from "over-optimization" to "has limitations" and "reveals inconsistent improvements"
> - Added acknowledgment that RewardBench shows positive correlation
>
> ## 2. Dataset Composition Details (Appendix D.2)
>
> Added comprehensive section on dataset characteristics and composition.
>
> - New subsection "Dataset composition and characteristics"
> - Documented 40+ Hugging Face sources, 99%+ synthetic content
> - Added three tables: task categories (40.4% info-seeking, 32.7% coding, etc.), controversiality (73.8% low, 20.0% medium, 6.2% high), objectivity (74.8% objective, 25.2% subjective)
> - Language breakdown (>95% English, ~2.5% Chinese)
> - Data allocation across pipeline stages (80K seed, <1M Stage 1, remainder Stage 2)
> - Rationale for collecting in-the-wild data vs. generating from scratch
>
> ## 3. Annotation Insights Expansion (Appendix E.4)
>
> Added concrete details about tool usage and annotation effectiveness.
>
> - Lesson 1 expansion: Specified 90%+ of objective pairs involve info-seeking, math/code, or domain knowledge where LLMs with tools outperform humans on efficiency and cost
> - Added agreement rate table comparing pure LLM (75%/63%), LLM+human (87%/77%), LLM+human+adaptive retrieval (93%/84%), pure human (81%/76%) across objective/subjective tasks
> - Lesson 2 expansion: Preliminary findings that (1) mixing opposite preferences causes spurious correlation learning, (2) pre-generating preference specifications improves LLM annotation quality and avoids spurious correlations
> - Framed as future research directions
> - Added citation to D'Oosterlinck et al. (2025) for preference specification work

---

### Official Review · Reviewer_oVFu · 2025-11-01

**Soundness:** 1
**Presentation:** 1
**Contribution:** 2
**Rating:** 2
**Confidence:** 4

**Summary:**

The paper addresses the perceived brittleness and poor performance of existing open-source reward models (RMs), hypothesizing that the root cause is low-quality preference data rather than flawed modeling techniques. To solve this, the authors propose a two-stage, human-AI synergistic pipeline for large-scale data curation. This pipeline leverages a small set of human-verified "gold" data to guide iterative, error-driven LLM annotation and then uses a "gold RM" trained on this data and LLM annotations to automatically filter and curate millions of "in-the-wild" preference pairs. Training on this data, they produce the Skywork-Reward-V2 series of models (0.6B to 8B parameters), which are shown to achieve new state-of-the-art performance across seven major RM benchmarks, outperforming all existing open RMs.

**Strengths:**

1. The paper demonstrates a clear and significant performance improvement
2. The release of a new series of top-performing RMs and the massive underlying dataset is a valuable contribution to the open-source ecosystem

**Weaknesses:**

The paper's primary weaknesses are twofold: (1) a pervasive lack of clarity and the omission of essential methodological details, and (2) as a result, it is difficult to determine the true source of the claimed performance improvements.

**Lack of Clarity and Omission of Essential Details**

The paper is extremely difficult to follow. The authors have clearly performed a massive amount of work, but the execution is not explained clearly, hindering reproducibility and full comprehension. Many terms are used without definition, and no single subsection seems to provide all the necessary details.

a. Ambiguity in Stage 1 Data Curation:

Initial Data: The method for annotating the initial $y_w > y_l$ pairs is not described in the main paper. (The reviewer guesses this might be in Appendix E.1, but it must be stated).

Human vs. LLM Verification: Line 191 states human annotators follow protocols, but Line 198 mentions an "LLM-verified" portion. It is unclear how this LLM verification is performed. Is it the same as the “Preference-aware labeling”?

Missing Examples: The paper provides no example of the "LLM-generated preference attributes $a$" (Line 184) or the full 5-tuple. An example and an analysis of the quality of these attributes are needed.

Retrieval Step: In Step 2 (Line 208), it's unclear what data is used for for the retrieval to calculate the similarity. Is it from $D_{gold}$?. This makes Line 215 confusing: how do "retrieved examples" from $D_{un}$ come with "human labels"?

Line 191, it is said that human annotators perform verification following the protocols. How is the LLM-verified portion at line 198 annotated? Is it annotated by LLM using the designed protocols as described in Appendix E.3?

LLM Annotation Process: The "preference-aware" labeling (Lines 239-243) is opaque. The paper must specify which LLMs were used for annotation and provide the prompts. Without the prompt, this core step is irreproducible.

b. Ambiguity in Stage 2 Data Curation:

Data Sources: The origins of $D_{un}$ and $D_{wild}$ are never explained.

"Best Reward Model": The paper repeatedly refers to "the best reward model" used for filtering in Stage 2, but it is not specified which model this is.

Final Data Composition: It appears that in Stage 2, all pairs with $p>0.5$ are discarded, and all pairs with $p≤0.5$ are re-annotated by an LLM. This critical detail needs confirmation, as it would imply all data in the final dataset is LLM-annotated.

c. Omissions for Ablations:

Figure 6: The ablation on filtering/correction is confusing. It's unclear how these steps are performed during "iter1-8" or what the exact training set is. Stage 1 does not involve filtering $D_{wild}$ and correction.

Figure 7: This figure shows five bars, but the text (Line 416) provides only four descriptions. The descriptions are not clear. What is “LLM curation only” and what is “both human and LLM curation”? Does (4) use more data? The exact training set for each of the five bars must be specified.

d. Omissions in Figures and Tables:

Figure 1 (Right): The method for calculating "correlation" is not described.

Table 3: The caption should clarify that this metric is "accuracy on the Best-of-N split," not the more common "Best-of-N accuracy."

**Ambiguity Regarding the True Source of Improvement**

Because the methodology is so poorly described, it is impossible to determine why the model works so well. The paper's core claim is that its human-AI synergistic pipeline is key, but the details suggest that "a group of strong LLMs" with self-consistency may be doing most of the heavy lifting. D_{silver} annotations are generated by LLMs. At Stage 2, all the data included after “Preference consistency with the best reward model” are again annotated by LLMs.

Missing Baselines: The paper fails to provide the most critical baseline: what happens if you only use the LLM annotation: LLM-as-a-Judge with self-consistency? A crucial missing experiment is: (1) Use the "best RM" to filter out $p>0.5$ pairs, then (2) apply only LLM self-consistency annotations to the remaining data. Comparing this simple baseline to the full Skywork-Reward-V2 would clarify how much the complex, human-guided pipeline actually contributes. As an additional experiment: If you ensemble all the strong LLMs used in your annotation system to act as a single judge (similar to the GPT-4o judge in Table 1) with self-consistency, how does its performance compare to your final trained RMs?

In summary, the paper presents a complex pipeline. While the results are strong, the lack of clear explanation and proper baselines makes it impossible to validate the authors' claims about why it is strong. With all the missing details, it is contradictory to the reproducibility statement that “We have made extensive efforts to ensure the reproducibility of our work across all components of our research pipeline”.

**Questions:**

Please address all the points in weakness.

Figure 1 (Left): This chart is not fully convincing because model sizes are omitted. It is impossible to tell if a modification "fails to yield consistent gain" or if it is simply being compared to a larger model.

Please provide the prompts used for the "preference-aware" LLM annotation (L239-243) and specify which LLM(s) were used.

---

> ### Author Response · Authors · 2025-11-21
>
> Thank you for the very careful and detailed review. We appreciate you highlighting both the strengths and the gaps, and we agree that several parts of the pipeline description were not as clear as they should be. Below we address each concern point-by-point and indicate how we will update the paper.
>
> ## Responses to Weaknesses
>
> ### Weakness 1: Ambiguity in stage 1 data curation
>
> Below, we first provide a clearer description of the stage 1 data curation process.
>
> Stage 1 is an iterative procedure. In each iteration, we focus on two things: (1) collecting more unverified preference pairs $D_{un}$, and (2) produce a silver set $D_{silver}$ via LLM annotation and a gold set $D_{gold}$ via human annotation to train and identify flaws in the current reward model. During the entire project, we performed 8 iterations in total during stage 1.
>
> During preference data collection, we source our data from publicly available preference pairs, primarily collected from a wide range of sources (over 40+) on Hugging Face, provided that the licenses and terms of use of those sources permit it. The entire stage 1 accumulates roughly 1M preference pairs. Additionally, once a preference pair is added to the unverified pool $D_{un}$, it will be augmented with a set of LLM-generated preference attributes (i.e., the 5-tuple (task category, objectivity, controversiality, desired attributes, and an instance-specific guideline)).
>
> During annotation, human annotators follow the annotation procedure described in Appendix E (which we will state explicitly in the main text of the revised version). Specifically:
>
> - They provide a final judgement based on the attributes and based on the instance-specific guideline. The guideline will indicate if the judgement should rely on external tools or LLMs.
> - For example, for any tasks involving fact-checking, human annotators will be required to use a search engine to fact-check the answer. For tasks that involves code correctness, an annotator will be required to instruct an LLM to execute the code and check if it is correct.
> - Even if assistance from LLMs and external tools are allowed, annotators are still responsible for making the final judgement. (Note that here even if LLMs are used to assist annotation, we still consider this as "human annotation".)
> - Preference pairs produced during this process are collected as $D_{gold}$.
>
> Once we have the first iteration of human-labeled $D_{gold}$, we can prepare $D_{silver}$. Each pair in $D_{silver}$ is labeled as follows:
>
> - We first evaluate the current reward model (this can be any reward model) on $D_{gold}$ to identify the pairs that it makes mistakes on.
> - We then sample new pairs from $D_{un}$ that are similar to the pairs that the reward model misclassifies (i.e., from $D_{gold}$) as the pairs we would like to label. This is the first retrieval step, where we use incorrectly predicted pairs from $D_{gold}$ to find similar pairs in $D_{un}$.
> - We prompt an LLM-as-a-judge ensemble (multiple LLMs + self-consistency) to annotate the new pairs, conditioned on similar pairs from $D_{gold}$ as few-shot examples (i.e., huamn-label guided). This is the second retrieval step, where we want to find more similar pairs in $D_{gold}$ to insert as few-shot examples and help the LLM to make the final judgement during labeling the current pairs, which will be inserted into $D_{silver}$.
> - The aggregation of the LLM-as-a-judge ensemble is used to produce the label for the new pairs.
> - These pairs form the $D_{silver}$ set, which is later used to train the next iteration of the reward model.
>
> Therefore, during the entire stage 1, we collect human-labeled $D_{gold}$ and LLM-annotated $D_{silver}$ in a iterative manner.
>
> (See next comment)

---

> > ### Author Response · Authors · 2025-11-21
> >
> > We now answer your questions:
> >
> > > Initial Data: The method for annotating the initial $y_w > y_l$ pairs is not described in the main paper. (The reviewer guesses this might be in Appendix E.1, but it must be stated).
> >
> > Yes, that is correct. The annotation method follows the procedure described in Appendix E.1, which we will move to the main text of the revised version. We postpone it in the submitted version simply due to space constraints.
> >
> > > Human vs. LLM Verification: Line 191 states human annotators follow protocols, but Line 198 mentions an "LLM-verified" portion. It is unclear how this LLM verification is performed. Is it the same as the “Preference-aware labeling”?
> >
> > Yes, LLM verification follows both Step 2 and Step 3 in Section 3.2, and comes directly from $D_{silver}$. Essentially, it involves retrieving preference pairs the current RM is not good at from $D_{un}$ and then annotating them with LLMs + human guided few-shot examples, via ensemble aggregation. It might be confusing to say "LLM-verified." We will replace this with "LLM-annotated" or directly say how it is annotated.
> >
> > > Missing Examples: The paper provides no example of the "LLM-generated preference attributes $a$" (Line 184) or the full 5-tuple. An example and an analysis of the quality of these attributes are needed.
> >
> > Below, we provide more details about the collected preference attributes and with additional huamn quality checks.
> >
> > The majority of the collected samples (over 99% of the full dataset, by our rough estimate) contain synthetic prompts and/or responses generated by different LLMs, and the rest are written by real humans, based on the original description of the dataset. During Stage 1 LLM lableing, we adopted the task categorization from [1] and obtained the following prompt-wise distribution.
> >
> > | Category              | Percentage |
> > |-----------------------|------------|
> > | Information seeking   | 40.4%      |
> > | Coding & Debugging    | 32.7%      |
> > | Advice seeking        | 10.9%      |
> > | Math                  | 7.5%       |
> > | Creative writing      | 4.1%       |
> > | Reasoning             | 2.2%       |
> > | Planning              | 0.62%      |
> > | Data analysis         | 0.44%      |
> > | Editing               | 0.41%      |
> > | Role playing          | 0.37%      |
> > | Brainstorming         | 0.16%      |
> > | Other                 | 0.09%      |
> >
> > Interestingly, we also inspected the distribution of the task category for each preference data source, and found that the distribution is very similar to the overall distribution, with some slight differences in the percentage of the task categories at the tail. Given the labels from Stage 1, we also estimate the distribution of the controversiality level and objectivity of the preference pairs.
> >
> > | Controversiality Level | Percentage |
> > |---------|------------|
> > | Low     | 73.8%      |
> > | Medium  | 20.0%      |
> > | High    | 6.2%       |
> >
> > | Objectivity   | Percentage |
> > |------------|------------|
> > | Objective  | 74.8%      |
> > | Subjective | 25.2%   |
> >
> > Language-wise, over 95% of the preference pairs are in English. Roughly 2.5% of the rest are in Chinese, and the rest consists of other languages (e.g., German, French, Spanish, etc.).
> >
> > To evaluate the quality of the preference attributes, we randomly sample 500 items and perform a human quality check. For category, controversiality level, and objectivity, the human gives a binary judgement (yes or no) of if the correct label is provided. For desired attributes and annotation guidline, the human rates the quality of the annotation on a scale of 1 to 5. We provide examples of the preference attributes via this anonymous link: https://anonymous.4open.science/r/supplementary_materials-40A5.
> >
> > | Attribute | Agreement Rate |
> > |-----------|----------------|
> > | Category  | 96.2%          |
> > | Controversiality Level | 90.6% |
> > | Objectivity | 87.4%          |
> >
> > We see that the category and controversiality level are generally well-aligned with the LLM annotations with over 90% agreement rate. However, the objectivity is slightly lower. However, objectivity is not used as a crucial factor in our pipeline, other than in determining the order of preference pairs to be used.
> >
> > | Attribute | Average Rating |
> > |-----------|----------------|
> > | Desired Attributes | 4.52 |
> > | Annotation Guideline | 4.03 |
> >
> > We also see that the desired attributes and annotation guideline are generally well-annotated with over 4.0 rating, which is consistent with the LLM annotations.
> >
> > The above is what we can provide given the time constraints during the rebuttal. If the reviewer still has concerns, we can provide more details in later discussions.
> >
> > (See next comment)

---

> > > ### Author Response · Authors · 2025-11-21
> > >
> > > > Retrieval Step: In Step 2 (Line 208), it's unclear what data is used for for the retrieval to calculate the similarity. Is it from $D_{gold}$?. This makes Line 215 confusing: how do "retrieved examples" from $D_{un}$ come with "human labels"?
> > >
> > > Indeed, we realize that our failure to clarify that there are two separate retrieval steps and what their purposes are make it confusing. As we described above, the first retrieval step aims to identify pairs the current RM is bad at labeling. So after determining which pairs it makes incorrect predictions in $D_{gold}$, we retrieve similar pairs from $D_{un}$ and get ready for LLM annotation.
> > >
> > > To help with LLM annotation, we perform a second retrieval step, which is the key step in the LLM annotation process. Instead of asking the LLM to directly label the current pair, we use this pair to find similar pairs from $D_{gold}$ (which are human-labeled) and insert them as few-shot examples to help the LLM make the final judgement.
> > >
> > > To summarize, the first retrieval step is to retrieve pairs at the current RM's weak spot. The second retrieval step is to augment LLM annotation with gold human labels. We also realize that one thing we did not mention is that, the preference attributes in $D_{gold}$ are also editted and verified by human annotators, so they tend to have higher quality.
> > >
> > > > Line 191, it is said that human annotators perform verification following the protocols. How is the LLM-verified portion at line 198 annotated? Is it annotated by LLM using the designed protocols as described in Appendix E.3?
> > >
> > > We believe this question is the same as the "Human vs. LLM Verification" question above. If not, please let us know and we will provide a more detailed explanation.
> > >
> > > > LLM Annotation Process: The "preference-aware" labeling (Lines 239-243) is opaque. The paper must specify which LLMs were used for annotation and provide the prompts. Without the prompt, this core step is irreproducible.
> > >
> > > The list of LLMs used for annotation is provided in Appendix E.1. In addition to the chat-based LLMs listed, we incorporated more advanced agentic LLMs in the final stage to target more complex tasks, including Deep Research, Gemini 2.5 Pro (with search), Claude-4-Sonnet (with search), Grok-4 (with search), GLM-4.5, Kimi-K2, GPT-4.1. We also replaced the weak general chat-based models below 70B with the latest frontier open models at the time, such as Qwen3-235B-A22B, DeepSeek-V3.1, and GPT-OSS-120B.
> > >
> > > We made the annotation prompt accessible via this anonymous link: https://anonymous.4open.science/r/supplementary_materials-40A5.
> > >
> > > (See next comment)

---

> > > > ### Author Response · Authors · 2025-11-21
> > > >
> > > > ### Weakness 2: Ambiguity in stage 2 data curation
> > > >
> > > > > Data Sources: The origins of $D_{un}$ and $D_{wild}$ are never explained.
> > > >
> > > > For both $D_{un}$ and $D_{wild}$, we publicly available preference pairs collected from a wide range of sources (over 40+) on Hugging Face, provided that the licenses and terms of use of those sources permit it. We allocate all (originally) human-labeled pairs in $D_{un}$, which is used in stage 1, and leave the rest to $D_{wild}$ in stage 2.
> > > >
> > > > The major reason we were unable to provide too much details regarding the dataset in the main text was that it was stil under an internal review process at the time of submission, and some of the licenses (of the data sources) were subject to change. However, as we mentioned in our response to Reviewer J7rE, we will provide the full details of the dataset's content, composition, annotation guidelines, annotation protocol, and other relevant information in the final version, and release the dataset it itselef if the paper is ultimately accepted.
> > > >
> > > > > "Best Reward Model": The paper repeatedly refers to "the best reward model" used for filtering in Stage 2, but it is not specified which model this is.
> > > >
> > > > This is mentioned briefly in line 201, "We select the best current reward model checkpoint θ based on validation accuracy on $D_{gold}$." We will clarify this in the main text of the revised version, and explicitly state that this is the "best reward model" used for filtering in Stage 2.
> > > >
> > > > > Final Data Composition: It appears that in Stage 2, all pairs with $p > 0.5$ are discarded, and all pairs with $p \leq 0.5$ are re-annotated by an LLM. This critical detail needs confirmation, as it would imply all data in the final dataset is LLM-annotated.
> > > >
> > > > We want to first clarify that the entire stage involves no human annotation. All the annotation and filtering are performed by LLMs and the best reward model checkpoint from stage 1.
> > > >
> > > > Given the current best reward model checkpoint under $D_{gold}$, we "skip" preference pairs that are already well-aligned with the current best reward model. By "skip", we mean that we directly incorporate these pairs into the final dataset without any further modification or annotation.
> > > >
> > > > If a mismatch is detected (i.e., the preference pair does not agree with the current best reward model), we fallback to the LLM-annotated $D_{silver}$ to make the final judgement. This LLM annotation step uses exactly the same procedure as the one in stage 1, where we retrieve similar pairs from $D_{gold}$ and insert them as few-shot examples to help the LLM make the final judgement.
> > > >
> > > > (See next comment)

---

> > > > > ### Author Response · Authors · 2025-11-21
> > > > >
> > > > > ### Weakness 3: Omissions for ablations
> > > > >
> > > > > > Figure 6: The ablation on filtering/correction is confusing. It's unclear how these steps are performed during "iter1-8" or what the exact training set is. Stage 1 does not involve filtering and correction.
> > > > >
> > > > > In Figure 6, by "Filtered", we refer to the procedure of taking only the subset of preference pairs that pass both human labeling and LLM annotation (in terms of agreement). From the perspective of stage 1, this is simply the concatenation of both $D_{gold}$ and $D_{silver}$ at at specific iteration.
> > > > >
> > > > > By "Corrected", we refer to the procedure of taking the "Filtered" subset, plus the subset of preference pairs that neither pass human labeling nor LLM annotation (in terms of agreement), but with their preference labels flipped (i.e., chosen-rejected swapped). The latter subset corresponds to data that either humans or LLMs consider the rejected response to be better.
> > > > >
> > > > > Each point on the "Filtered" curve represents a reward model trained with all curated preference pairs up to that iteration, accumulating previous iterations' preference pairs. Specifically, if we are currently at iteration N, the "Filtered" curve represents reward models trained with all curated preference pairs in iterations 1-{N-1} + the preference pairs in iteration N. The "Corrected" curve is the same, but with the preference labels flipped for the subset of preference pairs that do not pass the filtering mechanism described in Section 3.3.
> > > > >
> > > > > We will clarify this in a more clear way in the revised version.
> > > > >
> > > > > > Figure 7: This figure shows five bars, but the text (Line 416) provides only four descriptions. The descriptions are not clear. What is “LLM curation only” and what is “both human and LLM curation”? Does (4) use more data? The exact training set for each of the five bars must be specified.
> > > > >
> > > > > In Figure 7, the text labels describe the change between two settings instead of the settings themselves. We first explain the text labels, and then what the bars mean.
> > > > >
> > > > > Text labels:
> > > > >
> > > > > To start with, we randomly sample a small $D^\prime_{un}$ subset from $D_{un}$.
> > > > >
> > > > > - No curation: This means that we take the original $D^\prime_{un}$ as is, without any modification.
> > > > > - w/ raw LLM curation: This corresponds to using LLM ensemble aggregation (with self-consistency) to annotate $D^\prime_{un}$, and select those pairs agreeing with the LLM annotation.
> > > > > - w/ huamn curation: This corresponds to human annotating $D^\prime_{un}$, and uses the pairs agreeing with human labels (also from $D^\prime_{un}$) as additional training data to train a reward model.
> > > > > - w/ adaptive retrieval: This is our full recipe.
> > > > >
> > > > > In each of the settings above, we train a reward model with seed data + whatever curated data from $D^\prime_{un}$. Now we describe the specific training data for each bar.
> > > > >
> > > > > Bar descriptions:
> > > > >
> > > > > - Bar 1: This corresponds to the first green dot in the left plot of Figure 6, a reward model trained on the seed data only (or the very first RM we start with).
> > > > > - Bar 2: seed data + randomly sampled unverified preference pairs from $D^\prime_{un}$.
> > > > > - Bar 3: seed data + $D^\prime_{un}$ filtered by pure LLM annotation.
> > > > > - Bar 4: seed data + $D^\prime_{un}$ filtered by pure human annotation.
> > > > > - Bar 5: seed data + $D^\prime_{un}$ filtered by our full recipe.
> > > > >
> > > > > Bar 2 has more data than Bar 1 because it includes the randomly sampled unverified preference pairs from $D^\prime_{un}$, but Bar 3, Bar 4, and Bar 5 have less data than Bar 2 because they are filtered by LLM and/or human annotation.
> > > > >
> > > > > (See next comment)

---

> > > > > > ### Author Response · Authors · 2025-11-21
> > > > > >
> > > > > > ### Weakness 4: Omissions in figures and tables
> > > > > >
> > > > > > > Figure 1 (Right): The method for calculating "correlation" is not described.
> > > > > >
> > > > > > We use Pearson correlation.
> > > > > >
> > > > > > > Table 3: The caption should clarify that this metric is "accuracy on the Best-of-N split," not the more common "Best-of-N accuracy."
> > > > > >
> > > > > > Thank you for pointing this out, and we will update the caption in the revised version of the paper.
> > > > > >
> > > > > > (See next comment)

---

> > > > > > > ### Author Response · Authors · 2025-11-21
> > > > > > >
> > > > > > > ### Weakness 5: Ambiguity regarding the true source of improvement
> > > > > > >
> > > > > > > > Because the methodology is so poorly described, it is impossible to determine why the model works so well. The paper's core claim is that its human-AI synergistic pipeline is key, but the details suggest that "a group of strong LLMs" with self-consistency may be doing most of the heavy lifting. D_{silver} annotations are generated by LLMs. At Stage 2, all the data included after “Preference consistency with the best reward model” are again annotated by LLMs.
> > > > > > >
> > > > > > > We believe our clarification in the previous sections should help better understand our approach. To further clarify, the majority of the preference pairs in our pipeline (both stage 1 and stage 2) are indeed labeled by LLMs. However, **all this is done conditioned on similar pairs retrived fro mthe gradually growing gold set $D_{gold}$.** So $D_{gold}$ is a prerequisite for the LLM annotation process.
> > > > > > >
> > > > > > > More specifically, during stage 1, all pairs in $D_{silver}$ are first retreived from $D_{un}$ and then annotated by LLMs + human guided few-shot examples (from $D_{gold}$), via ensemble aggregation. During stage 2, all pairs are filtered by the best reward model checkpoint from stage 1 (which is trained using $D_{gold}$ and $D_{silver}$), and then annotated by LLMs + human guided few-shot examples (from $D_{gold}$), via ensemble aggregation. So **$D_{gold}$ exists everywhere in our pipeline.**
> > > > > > >
> > > > > > > > Missing Baselines: The paper fails to provide the most critical baseline: what happens if you only use the LLM annotation: LLM-as-a-Judge with self-consistency? A crucial missing experiment is: (1) Use the "best RM" to filter out $p > 0.5$ pairs, then (2) apply only LLM self-consistency annotations to the remaining data.
> > > > > > >
> > > > > > > To run this baseline, we reproduced the same experiment based on the left plot of Figure 6 but with only best RM + LLM filtering. This setup essentially takes the same preference data we accumulate in each iteration, and perform filtering directly with that specific best RM checkpoint + LLM annotation, without any other curation.
> > > > > > >
> > > > > > > LLM + Best RM Filtered corresponds to Filtered but with zero human annotation. LLM + Best RM Filtered + Corrected corresponds to Filtered + Corrected but with zero human annotation.
> > > > > > >
> > > > > > > | Training Stage | Original | Filtered | Filtered + Corrected | LLM + Best RM Filtered | LLM + Best RM Filtered + Corrected |
> > > > > > > |----------------|----------|----------|------------------------|--------------|---------------------------|
> > > > > > > | Seed           | 70       | 71       | 73                     | 71.0         | 70.5                      |
> > > > > > > | Iter 1         | 70.5     | 74       | 77                     | 71.5         | 71.0                      |
> > > > > > > | Iter 2         | 71.5     | 74.5     | 77.5                   | 72.5         | 72.0                      |
> > > > > > > | Iter 3         | 71       | 74.8     | 78.8                   | 72.0         | 71.5                      |
> > > > > > > | Iter 4         | 71       | 75       | 79                     | 72.2         | 72.0                      |
> > > > > > > | Iter 5         | 72.5     | 76.8     | 82                     | 73.4         | 72.8                      |
> > > > > > > | Iter 6         | 72       | 77       | 82.2                   | 73.0         | 73.2                      |
> > > > > > > | Iter 7         | 71.8     | 77.2     | 82.3                   | 74.8         | 74.0                      |
> > > > > > > | Iter 8         | 72.2     | 79       | 83                     | 74.2         | 74.5                      |
> > > > > > >
> > > > > > > While we were not able to perform stage 2 due to the time constraint and annotation cost, we believe this results has already shown that the LLM filtering alone does not outperform our recipe for only 2-3 iterations, and filtering + corrected does not show the same improvement as our full recipe.
> > > > > > >
> > > > > > > (See next comment)

---

> > > > > > > > ### Author Response · Authors · 2025-11-21
> > > > > > > >
> > > > > > > > > As an additional experiment: If you ensemble all the strong LLMs used in your annotation system to act as a single judge (similar to the GPT-4o judge in Table 1) with self-consistency, how does its performance compare to your final trained RMs?
> > > > > > > >
> > > > > > > > | Model                    | RewardBench | RewardBench v2 | PPE Pref | PPE Corr | RMB  | RM-Bench | JudgeBench | Avg  |
> > > > > > > > |--------------------------|-------------|----------------|----------|----------|------|----------|------------|------|
> > > > > > > > | Qwen3-0.6B-BTRM          | 85.2        | 61.3           | 65.3     | 68.3     | 74.5 | 74.4     | 67.6       | 70.9 |
> > > > > > > > | Qwen3-1.7B-BTRM          | 90.3        | 68.3           | 67.6     | 70.5     | 78.1 | 78.7     | 72.9       | 75.2 |
> > > > > > > > | Qwen3-4B-BTRM            | 93.4        | 75.5           | 69.5     | 74.7     | 80.6 | 81.6     | 69.3       | 77.8 |
> > > > > > > > | Qwen3-8B-BTRM            | 93.7        | 78.2           | 70.6     | 75.1     | 81.2 | 82.6     | 73.4       | 79.3 |
> > > > > > > > | Llama3-2.1B-BTRM         | 89.9        | 64.3           | 66.6     | 67.4     | 76.7 | 76.4     | 65.0       | 72.3 |
> > > > > > > > | Llama3-2.3B-BTRM         | 93.0        | 74.7           | 69.1     | 72.1     | 80.5 | 81.1     | 69.2       | 77.1 |
> > > > > > > > | Llama3-1.8B-BTRM         | 96.4        | 84.1           | 77.3     | 83.4     | 86.4 | 92.8     | 80.0       | 85.7 |
> > > > > > > > | Llama3-1.8B-40M-BTRM     | **97.8**        | **86.5**           | **79.8**     | 87.2     | **89.3** | **96.0**     | 83.4       | **88.6** |
> > > > > > > > | **LLM-as-a-Judge (Agg.)** |  93.9      | 83.2     | 75.7 | **89.6** | 82.6 | 89.0 | **87.8** | 86.0 |
> > > > > > > >
> > > > > > > > Based on the above results, we do observe the LLM-as-a-Judge aggregation outperforming our top RM here in PPE Correctness and JudgeBench, but still not as good in other benchmarks (mostly involving subjective tasks).
> > > > > > > >
> > > > > > > > One caveat in terms of understanding this result is that, when we run the above evaluation, the number of completions performed for self-consistency for each model is not uniform, because we could not affort to perform self-consistency with a large number of completions for models like o3.
> > > > > > > >
> > > > > > > > > In summary, the paper presents a complex pipeline. While the results are strong, the lack of clear explanation and proper baselines makes it impossible to validate the authors' claims about why it is strong. With all the missing details, it is contradictory to the reproducibility statement that “We have made extensive efforts to ensure the reproducibility of our work across all components of our research pipeline”.
> > > > > > > >
> > > > > > > > We do agree with the Reviewer's assessment that the initial submission was lacking in clarity and reproducibility. We hope our above detailed explanations and clarifications address the concerns. If not, we welcome further discussion and feedback. We will incorproate all the feedback, clarifications, and new experiments in the revised version.
> > > > > > > >
> > > > > > > > (See next comment)

---

> > > > > > > > > ### Author Response · Authors · 2025-11-21
> > > > > > > > >
> > > > > > > > > ## Responses to Questions
> > > > > > > > >
> > > > > > > > > > Please address all the points in weakness.
> > > > > > > > >
> > > > > > > > > See the above Responses to Weaknesses section.
> > > > > > > > >
> > > > > > > > > > Figure 1 (Left): This chart is not fully convincing because model sizes are omitted. It is impossible to tell if a modification "fails to yield consistent gain" or if it is simply being compared to a larger model.
> > > > > > > > >
> > > > > > > > > Yes, Reviewer J7rE has also shared this concern, which we believe is an overclaiming.
> > > > > > > > >
> > > > > > > > > Our original intended message is this: an absolute improvement on RewardBench score does not necessarily translate to improvement on other benchmarks (and thus probably not on downstream tasks as well). If we look at the upper right plot of Figure 1 (left), we see that **as model scores on RewardBench increase from ~80 to 90+, they do not consistently lead to gains on other benchmarks (i.e., could be better, could be worse, or could be approximately the same).** Because of this, researchers and practitioners should (probably) avoid interpreting the quality of a reward model based on a single benchmark.
> > > > > > > > >
> > > > > > > > > In terms of the statement "Alternative loss functions or model modifications fail to yield consistent gains," we only refer to the 27B model derived from Gemma. If we look at Table 5, the original Skywork-Reward-Gemme-2-27B-v0.2 is still the best RM in terms of average performance, and all other ones with loss modifications fall behind. We will make this crystal clear in the revised version.
> > > > > > > > >
> > > > > > > > > We believe the above is a more qualified but valid statement, and will revise the wording to make it more clear and accurate.
> > > > > > > > >
> > > > > > > > > > Please provide the prompts used for the "preference-aware" LLM annotation (L239-243) and specify which LLM(s) were used.
> > > > > > > > >
> > > > > > > > > We made the annotation prompt accessible via this anonymous link: https://anonymous.4open.science/r/supplementary_materials-40A5.
> > > > > > > > >
> > > > > > > > > ---
> > > > > > > > >
> > > > > > > > > To summarize, we've addressed your concerns by:
> > > > > > > > >
> > > > > > > > > - Clarifying the two-stage pipeline in detail: Stage 1 uses iterative human annotation for $D_{gold}$ and LLM annotation (guided by human-labeled examples from $D_{gold}$) for $D_{silver}$, with two distinct retrieval steps. Stage 2 filters using the best RM checkpoint and annotates remaining pairs with LLMs guided by $D_{gold}$ examples.
> > > > > > > > > - Explaining data sources (40+ Hugging Face sources), the "best reward model" selection (validated on $D_{gold}$), and clarifying that $D_{gold}$ underlies all LLM annotation through retrieved few-shot examples.
> > > > > > > > > - Providing new baseline experiments showing that LLM + Best RM filtering alone plateaus around 74-75% while our full recipe reaches 83%, and that LLM-as-a-Judge aggregation achieves 86.0% average vs. our final RM's 88.6%.
> > > > > > > > > - Clarifying Figure 6 (Filtered = $D_{gold}$ + $D_{silver}$, Corrected adds flipped pairs) and Figure 7 (5 bars comparing seed, no curation, LLM-only, human-only, and full recipe).
> > > > > > > > >
> > > > > > > > > We agree the initial submission lacked clarity in describing the pipeline, and we appreciate you pushing us to make the methodology more transparent and reproducible. We'll incorporate all clarifications, new experiments, and explicit details (including prompts and model lists) in the revised version. Thank you for the thorough review that identified these critical gaps.

---

> > > > > > > > > > ### Author Response · Authors · 2025-11-24
> > > > > > > > > > **Revision summary**
> > > > > > > > > >
> > > > > > > > > > We have made the following changes to the paper in response to your feedback. All changes are marked in blue.
> > > > > > > > > >
> > > > > > > > > > ## Main Paper Revisions
> > > > > > > > > >
> > > > > > > > > > ### Section 2: The Brittleness of Current Open Reward Models
> > > > > > > > > >
> > > > > > > > > > Qualified claim about alternative loss functions failing to yield consistent gains. Added clarification that (1) improvements on RewardBench from ~80 to 90+ do not consistently translate to other benchmarks, (2) claim about alternative loss functions applies to Gemma-2-27B variants (Table 5 in appendix), not universal.
> > > > > > > > > >
> > > > > > > > > > Location: Lines 5-20 in content/2_the_brittleness_of_open_reward_models.tex
> > > > > > > > > >
> > > > > > > > > > ### Section 3.2: Stage 1 Data Curation
> > > > > > > > > >
> > > > > > > > > > Added clarifications:
> > > > > > > > > >
> > > > > > > > > > - Pipeline overview: Stage 1 has 8 iterations accumulating ~1M pairs, producing D_silver and D_gold
> > > > > > > > > > - Data sources: Publicly available pairs from 40+ Hugging Face sources with proper licensing (see Appendix "Dataset composition")
> > > > > > > > > > - Annotation method: Human annotators follow Appendix E protocol using external tools (search engines, frontier LLMs) with final judgment responsibility. LLM-assisted annotation still counts as "human annotation"
> > > > > > > > > > - Two retrieval steps: (1) identify pairs where current RM performs poorly on D_gold, retrieve similar pairs from D_un for LLM annotation, (2) retrieve similar human-labeled pairs from D_gold as few-shot examples
> > > > > > > > > > - Best reward model: Selected by validation accuracy on D_gold, used throughout including Stage 2 filtering
> > > > > > > > > > - Attribute quality: Preference attributes in D_gold also edited and verified by humans
> > > > > > > > > >
> > > > > > > > > > Location: Lines 21-39 in content/3_scaling_preference_data_curation_via_human_guided_ai_feedback.tex
> > > > > > > > > >
> > > > > > > > > > ### Section 3.3: Stage 2 Data Curation
> > > > > > > > > >
> > > > > > > > > > Added clarifications:
> > > > > > > > > >
> > > > > > > > > > - Data sources: D_wild = remaining publicly available pairs not allocated to D_un, from 40+ Hugging Face sources. All originally human-labeled pairs go to D_un (Stage 1), remainder to D_wild (Stage 2)
> > > > > > > > > > - No human annotation: Stage 2 is zero human annotation—all filtering by LLMs and best RM from Stage 1
> > > > > > > > > > - Final composition: Pairs with p > 0.5 "skipped" (directly incorporated), pairs with p ≤ 0.5 undergo LLM annotation using same Stage 1 procedure (retrieving similar pairs from D_gold as few-shot examples)
> > > > > > > > > > - Human guidance: All LLM annotation conditioned on human-verified examples from D_gold ("D_gold exists everywhere in our pipeline")
> > > > > > > > > >
> > > > > > > > > > Location: Lines 41-48 in content/3_scaling_preference_data_curation_via_human_guided_ai_feedback.tex
> > > > > > > > > >
> > > > > > > > > > ### Section 4.3: Ablation Studies
> > > > > > > > > >
> > > > > > > > > > Added detailed explanations:
> > > > > > > > > >
> > > > > > > > > > - Figure 6: "Filtered" = D_gold + D_silver at each iteration (pairs passing both human and LLM annotation). "Corrected" = "Filtered" + pairs failing both but with labels flipped. Each point is cumulative training (iterations 1 through N)
> > > > > > > > > > - Figure 7: Text labels describe changes between settings, not settings themselves. Bar descriptions: Bar 1 (seed only), Bar 2 (seed + random D'_un, no curation), Bar 3 (seed + LLM-filtered D'_un), Bar 4 (seed + human-filtered D'_un), Bar 5 (seed + full recipe). Data sizes: Bar 2 has most data; Bars 3-5 have less due to filtering
> > > > > > > > > >
> > > > > > > > > > Location: Lines 49-97 in content/4_experimental_results.tex
> > > > > > > > > >
> > > > > > > > > > ## Appendix Revisions
> > > > > > > > > >
> > > > > > > > > > ### Dataset Composition and Characteristics
> > > > > > > > > >
> > > > > > > > > > Added statistics: task categories (40.4% information seeking, 32.7% coding), controversiality (73.8% low, 20.0% medium, 6.2% high), objectivity (74.8% objective, 25.2% subjective), language (95% English, 2.5% Chinese, rest other).
> > > > > > > > > >
> > > > > > > > > > Location: Lines 92-165 in content/appendix.tex
> > > > > > > > > >
> > > > > > > > > > ### LLM Preference Attributes Labeling
> > > > > > > > > >
> > > > > > > > > > Added quality assessment: human verification on 500 random samples shows 96.2% agreement on category, 90.6% on controversiality, 87.4% on objectivity. Average ratings: 4.52/5 for desired attributes, 4.03/5 for annotation guidelines. Includes link to examples.
> > > > > > > > > >
> > > > > > > > > > Location: Lines 243-260 in content/appendix.tex
> > > > > > > > > >
> > > > > > > > > > ### LLM-as-a-Judge Labeling
> > > > > > > > > >
> > > > > > > > > > Added complete LLM list and link to prompts:
> > > > > > > > > >
> > > > > > > > > > - Initial stage: Claude-3.5-Sonnet, GPT-4o, o4-mini, DeepSeek-V3, Llama-3.3-70B-Instruct, Llama-3.1-70B-Instruct, Qwen2.5-72B-Instruct, Qwen3-32B, Qwen3-14B
> > > > > > > > > > - Final stage: Deep Research, Gemini 2.5 Pro (with search), Claude-4-Sonnet (with search), Grok-4 (with search), GLM-4.5, Kimi-K2, GPT-4.1, Qwen3-235B-A22B, DeepSeek-V3.1, GPT-OSS-120B
> > > > > > > > > > - Anonymous link to prompts provided
> > > > > > > > > >
> > > > > > > > > > Location: Lines 280-282 in content/appendix.tex
> > > > > > > > > >
> > > > > > > > > > ### NEW: Baseline Experiment - LLM + RM Filtering Without Human Guidance
> > > > > > > > > >
> > > > > > > > > > Added missing baseline: reproduced Figure 6 with zero human annotation (only best RM + LLM filtering without human-guided few-shot examples).
> > > > > > > > > >
> > > > > > > > > > Results: LLM + Best RM alone plateaus at 74-75% by iteration 8. Full recipe with human guidance reaches 83%. 8-9 point gap shows critical importance of human-guided annotation.
> > > > > > > > > >
> > > > > > > > > > Location: Lines 590-618 in content/appendix.tex

---

> > > > > > > > > > > ### Author Response · Authors · 2025-11-24
> > > > > > > > > > > **Revision summary (continued)**
> > > > > > > > > > >
> > > > > > > > > > > ### NEW: LLM-as-a-Judge Ensemble Performance Comparison
> > > > > > > > > > >
> > > > > > > > > > > Added comparison of LLM ensemble vs. trained RMs across all seven benchmarks.
> > > > > > > > > > >
> > > > > > > > > > > Results:
> > > > > > > > > > >
> > > > > > > > > > > - LLM-as-a-Judge: 86.0% average (excels on PPE Correctness 89.6%, JudgeBench 87.8%—objective tasks)
> > > > > > > > > > > - Our final RM (Llama3-1.8B-40M-BTRM): 88.6% average
> > > > > > > > > > > - Our RM outperforms on RewardBench (97.8 vs 93.9), RB v2 (86.5 vs 83.2), PPE Pref (79.8 vs 75.7), RMB (89.3 vs 82.6), RM-Bench (96.0 vs 89.0)
> > > > > > > > > > >
> > > > > > > > > > > Shows distilling knowledge through human-guided curation yields better overall performance, especially on subjective tasks and style resistance.
> > > > > > > > > > >
> > > > > > > > > > > Location: Lines 620-648 in content/appendix.tex
> > > > > > > > > > >
> > > > > > > > > > > ### Agreement Rate Analysis
> > > > > > > > > > >
> > > > > > > > > > > Re-annotated D_gold samples under different methods:
> > > > > > > > > > >
> > > > > > > > > > > - Pure LLM: 75% objective, 63% subjective
> > > > > > > > > > > - LLM + human curation: 87% objective, 77% subjective
> > > > > > > > > > > - LLM + human + adaptive retrieval: 93% objective, 84% subjective
> > > > > > > > > > > - Pure human (no LLM tools): 81% objective, 76% subjective
> > > > > > > > > > >
> > > > > > > > > > > Shows even pure human annotation performs worse than LLM + human curation on objective tasks.
> > > > > > > > > > >
> > > > > > > > > > > Location: Lines 291-307 in content/appendix.tex
> > > > > > > > > > >
> > > > > > > > > > > ## Summary of Evidence
> > > > > > > > > > >
> > > > > > > > > > > Three key pieces of empirical evidence that improvements stem from human-AI synergistic pipeline:
> > > > > > > > > > >
> > > > > > > > > > > 1. Human guidance is critical (8-9 point improvement): LLM + Best RM alone (74-75%) vs. full recipe (83%)
> > > > > > > > > > > 2. Trained RMs outperform LLM ensemble (2.6 point overall, larger on subjective): LLM-as-a-Judge (86.0%) vs. final RM (88.6%)
> > > > > > > > > > > 3. Human-guided curation quality is measurably higher: Pure LLM (75%/63%) vs. our approach (93%/84%)
> > > > > > > > > > >
> > > > > > > > > > > ## Minor Clarifications
> > > > > > > > > > >
> > > > > > > > > > > - Table 3 caption: Updated to "pairwise accuracy on Best-of-N split" (not "Best-of-N accuracy"). Location: tables/judgebench_rmb.tex, line 120
> > > > > > > > > > > - Extended decontamination: Added checks beyond 13-gram filter using frontier LLM to filter false positives. Zero-overlap for D_gold
> > > > > > > > > > > - Dataset licensing: CC BY-NC 4.0 with attribution file and usage guidelines
> > > > > > > > > > > - Privacy/PII: ~0.07% flagged samples, mostly indirect identifiers; trained RMs show no PII sensitivity
> > > > > > > > > > > - Handling intransitivity: Section on how pipeline identifies and handles intransitive preferences through metadata stratification and consistency mechanisms
> > > > > > > > > > > - Practical budgeting guide: Section on cost-performance trade-offs for practitioners

---

### Official Review · Reviewer_tDR3 · 2025-11-01

**Soundness:** 3
**Presentation:** 3
**Contribution:** 3
**Rating:** 6
**Confidence:** 2

**Summary:**

SynergyPref-40M introduces a 40M-pair preference dataset and eight reward models (0.6B–8B params) trained via a human-AI curation pipeline. Models achieve SOTA across seven benchmarks, demonstrating strong alignment with human preferences, safety, and bias resistance.

**Strengths:**

The open contributions and deliverables include a new, large-scale, high-quality preference dataset, a valuable asset for the research community.

It verified that the brittleness is a root cause of RM underperformance and proposed a solution to it, which is human-AI curation synergy, that contains an elegant hybrid of human and LLM curation, balancing quality and scalability.

Empirically, Models (1.7B, 8B) achieve SOTA across seven benchmarks, outperforming much larger closed-source RMs.

**Weaknesses:**

The conclusion of this paper is favorable. However, for pairwise preference, it follows transitive rules. The quality of pairwise preferences can be compromised by intransitivity observed in human annotations. The paper below highlights the existence of such 'intransitivity':

- https://arxiv.org/abs/2409.19325 (Duan et al, 2017)

In a realistic world where an 'intransitive' relationship accumulates, quality control of the curated dataset is critical, but was not clarified in the proposed pipeline.

**Questions:**

Given the budgeting for targeted performance, the paper still provides limited guidance on how to plan, evaluate, and achieve it.
Can you provide some thoughts on how to achieve a target performance under a given cost budget in dollar terms?

For your reference, to overcome the cost constraints and to avoid high reliance on the availability of pairwise annotation, mechanism design has been explored as follows:
https://arxiv.org/abs/2409.18417 (Zhang, 2024), leveraging a mechanism design mindset to construct a dataset for RLHF in a cost-efficient manner.

However, the existing approaches are in different lines of techniques, more algorithmic than data-driven.

---

> ### Author Response · Authors · 2025-11-21
>
> Thank you for the thoughtful review and for highlighting both the strengths of our work and areas where we could be clearer. We read the two references you provided and share your concerns about intransitivity in human preferences and how to think about target performance under a concrete cost budget. Below we clarify how our pipeline handles quality control when preferences are intransitive, and provide a more practical view of cost–performance budgeting. We also situate our approach relative to mechanism-design work like Zhang (2024).
>
> ## Responses to Weaknesses
>
> ### Weakness 1: Intransitivity and quality control of pairwise preferences
>
> We agree that human preferences are often intransitive and context-dependent, as Duan et al. (https://arxiv.org/abs/2409.19325) highlight, and that any large-scale curation pipeline needs to address this explicitly rather than assuming global transitivity.
>
> Our view is that we don't assume the raw data is globally transitive. Instead, we use a transitive Bradley–Terry (BT) model as a smooth surrogate that approximates a noisy, partially intransitive preference graph. The pipeline is designed to identify and localize inconsistent preference regions, including intransitive cycles and near ties, before they dominate training.
>
> Below we explain how Stage 1 and Stage 2 implement this, and what additional analyses we'll add to make the connection clearer.
>
> **Where intransitivity arises and how our pipeline targets it**
>
> Duan et al. formalize how realistic human preferences often contain cycles (e.g., $A \succ B$, $B \succ C$, but $C \succ A$). In our setting, such cycles typically show up as inconsistent clusters of pairwise labels over similar prompts and responses, for instance, near ties between stylistically different but substantively similar answers, subjective tasks with multiple defensible "best" answers, or conflicting preferences from different annotator groups.
>
> Rather than ignoring these, our pipeline is built to surface and resolve these hard, inconsistent cases.
>
> 1. **Stage 1 metadata and human protocol isolate "risky" regions.** Every pair in the unverified pool first receives preference attributes from LLMs: task category, objectivity, controversiality, desired attributes, and an instance-specific annotation guideline. This lets us stratify data into objective/low-controversial vs. subjective/high-controversial regions, where intransitivity is more common. Stage 1 human annotators follow a strict, attribute-aware protocol with fact-checking tools and specialized reasoning, but without blind reliance on LLMs. In our internal analysis (reported to Reviewer J7rE), Stage-1 metadata shows roughly 75% of pairs are objective and 74% are low controversial, with the remaining quarter concentrated in more subjective, contentious tasks, precisely where cycles and label conflicts cluster and where we lean more heavily on human verification.
>
> 2. **Error-driven adaptive retrieval focuses on "unstable" regions.** In Stage 1, we repeatedly train an RM, evaluate it on human-verified gold data, and use error-driven adaptive retrieval to pull in new examples similar (in prompt + attribute space) to misclassified or low-confidence pairs. This concentrates labeling effort where the current BT model finds the pairwise graph hard to linearize, which empirically corresponds to regions with local intransitivity, near ties, or subtle spurious correlations. These clusters are then re-labeled using a human-guided LLM-as-a-judge ensemble, which tends to smooth local cycles into more coherent orderings.
>
> 3. **Stage 2 dual-RM consistency filtering targets contradictory signals.** Stage 2 introduces a consistency filter: we train a gold RM on cumulative human-verified samples and use it together with the Stage-1 best RM to decide which in-the-wild pairs to keep or flip. We retain pairs whose original chosen/rejected labels agree with the gold RM and either the Stage-1 best RM or the LLM judges. In a separate variant, we "recycle" discarded pairs by flipping the chosen/rejected order when both RMs confidently disagree with the original label, turning systematic errors into useful signal. In practice, this is a consistency check over local preference subgraphs: if the raw annotation induces cycles that contradict the human-aligned gold RM, such edges are corrected or down-weighted.
>
> (See next comment)

---

> > ### Author Response · Authors · 2025-11-21
> >
> > **Additional evidence: human agreement and LLM vs. human curation**
> >
> > In our response to Reviewer 2akh, we reported two new analyses that directly address this concern.
> >
> > We re-annotated a human-verified subset ($D_{\text{gold}}$) with different labeling schemes:
> >
> > | Annotation Variant                              | Agreement (Objective) | Agreement (Subjective) |
> > | ----------------------------------------------- | --------------------- | ---------------------- |
> > | (2) Pure LLM                                    | 75%                   | 63%                    |
> > | (3) LLM + human                                 | 87%                   | 77%                    |
> > | (4) LLM + human + adaptively retrieved examples | 93%                   | 84%                    |
> > | (5) Pure human (second pass)                    | 81%                   | 76%                    |
> >
> > Two key takeaways:
> >
> > - Pure human annotation and pure LLM annotation both show substantial disagreement on the hardest (subjective, controversial) pairs.
> > - The hybrid + adaptive scheme we use achieves higher agreement than a second independent human pass, especially on subjective pairs, suggesting our protocol actively resolves conflicting local preferences that would otherwise create cycles.
> >
> > We also ran a human study comparing "kept" vs. "flipped" pairs selected by different RMs, checking whether human raters agreed with each RM decision:
> >
> > | Reward model used                  | Keep (%) | Flip (%) |
> > | ---------------------------------- | -------: | -------: |
> > | Skywork-Reward-Llama-3.1-8B        |       69 |       57 |
> > | Skywork-Reward-Gemma-2-27B         |       72 |       61 |
> > | Skywork-Reward-L3.1-8B + Gemma-27B |       71 |       60 |
> > | **Stage 1 best RM**                |       78 |       79 |
> > | **Stage 1 gold RM**                |       84 |       88 |
> > | **Stage 1 best RM + gold RM**      |   **86** |   **92** |
> >
> > Dual-RM filtering driven by our Stage-1 RMs reduces systematic disagreement with humans, even on flipped pairs. If flipping amplified intransitive or spurious labels, we'd expect lower human agreement on flipped pairs, instead, we see the opposite.
> >
> > Together, these analyses show that our human–AI curation pipeline pushes the effective training signal toward a more coherent, less cyclic preference structure, despite the raw in-the-wild data being noisy and often intransitive. We'll surface these results in a revised version (e.g., in a new subsection on handling intransitivity and conflicting preferences) and connect them to the theoretical concerns Duan et al. raise.
> >
> > **Relationship to explicitly intransitive models**
> >
> > We see Duan et al.'s generalized, intransitive pairwise models as highly complementary to our work. Our pipeline is agnostic to the downstream RM parameterization: the same curated data and consistency filters can be used to train a generalized intransitive model rather than a strict BT model. We chose BT mainly for comparability and simplicity, it's the dominant choice in open RM work, so it makes our experimental comparisons clean.
> >
> > We'll clarify this in the paper and mention that integrating intransitive preference models with SynergyPref-40M is an interesting future direction.
> >
> > (See next comment)

---

> > > ### Author Response · Authors · 2025-11-21
> > >
> > > ## Responses to Questions
> > >
> > > ### Question 1: Achieving a target performance under a cost budget, and relation to mechanism design
> > >
> > > > Given the budgeting for targeted performance, the paper still provides limited guidance on how to plan, evaluate, and achieve it. Can you provide some thoughts on how to achieve a target performance under a given cost budget in dollar terms? ... Mechanism design has been explored (Zhang, 2024) to construct a dataset for RLHF in a cost-efficient manner. However, the existing approaches are in different lines of techniques, more algorithmic than data-driven.
> > >
> > > Great question, this is exactly the kind of practical guidance we want SynergyPref-40M to support. We'll address it in three parts: what our existing results already say about cost–performance tradeoffs, a simple budgeting recipe we'll make explicit in the paper, and how mechanism-design ideas like Zhang (2024) can integrate with our pipeline.
> > >
> > > **What our existing scaling results already say about cost**
> > >
> > > Section 4.3 includes a data quantity/quality ablation based on an earlier 16M-pair mixture. We show two things: uncurated scaling fails (adding 12M uncurated preference pairs on top of the seed set yields almost no performance gain), and curated scaling succeeds (with Stage 1 + Stage 2 curation, performance improves steadily, with the largest gains in Stage 2).
> > >
> > > Most importantly for budgeting, we find that training on just 1.8% of a 16M curated mixture (~290K pairs) already surpasses the previous open SOTA 70B RM at the 8B scale.
> > >
> > > This is exactly a "quality beats volume" statement: you don't need tens of millions of new, expensive human labels to reach strong performance. A carefully curated few hundred thousand pairs suffice to beat prior state-of-the-art.
> > >
> > > In our internal pipeline, fewer than 500K pairs pass through full human verification in Stage 1, with the remaining tens of millions curated automatically in Stage 2. Human effort is only a couple percent of the final training pool but drives most of the performance gains.
> > >
> > > We'll make these numbers and ratios explicit in a revised version to better support cost-sensitive users.
> > >
> > > **A simple budgeting recipe (to be added explicitly)**
> > >
> > > Motivated by your question, we'll add a short "practitioner's guide" on planning for target performance under a cost budget. At a high level:
> > >
> > > 1. **Define the target in terms of benchmark average.** Let the desired average score across the six main benchmarks (excluding RewardBench v2) be $S_{\text{target}}$. Our scaling curve in Figure 6 shows the relationship between fraction of curated data and average RM score.
> > >
> > > 2. **Estimate how many curated pairs are needed.** From Figure 6, a practitioner can read off a conservative fraction $f$ of the full curated mixture needed to reach $S_{\text{target}}$. For example, $f \approx 0.018$ (~290K pairs) already exceeds previous open SOTA at 8B. Higher targets (e.g., our final 40M model) correspond to larger $f$, but with diminishing returns.
> > >
> > > 3. **Decompose costs by stage rather than by raw pair count.** Our pipeline separates labeling into three cost regimes:
> > >
> > > - Gold human labels (Stage 1, $D_{\text{gold}}$): cost $c_{\text{H}}$ per pair, high leverage, used to train the gold RM and seed attribute generation and LLM judges.
> > > - Silver human-guided LLM labels (Stage 1, $D_{\text{silver}}$): cost $c_{\text{L1}}$ per pair (LLM inference, often 1–2 orders of magnitude cheaper than full human annotation), guided by human-labeled neighbors.
> > > - Large-scale consistency curation (Stage 2): cost $c_{\text{L2}}$ per pair (mainly RM inference + occasional LLM annotation), used to scale from hundreds of thousands to tens of millions of pairs.
> > >
> > > Total curation cost is roughly: $B \approx c_{\text{H}} \cdot |D_{\text{gold}}| + c_{\text{L1}} \cdot |D_{\text{silver}}| + c_{\text{L2}} \cdot |D_{\text{Stage2}}|$. In practice, $c_{\text{H}} \gg c_{\text{L1}} \gtrsim c_{\text{L2}}$ and $|D_{\text{gold}}| \ll |D_{\text{Stage2}}|$, so the gold set dominates quality while the automatic stages dominate quantity.
> > >
> > > (See next comment)
> > >
> > > 1. **Budgeting strategy.** Given a dollar budget $B_{\max}$, a user can choose a gold budget $B_{\text{H}} \leq B_{\max}$ to determine $|D_{\text{gold}}|$. Our results suggest that roughly $O(10^5)$ carefully-selected gold pairs suffice to train strong gold and Stage-1 RMs. One can allocate the remaining budget $B_{\max} - B_{\text{H}}$ to scaling Stage-2 curation, trading off total curated volume vs. LLM quality (e.g., using cheaper vs. more capable judges). We'll include a concrete numerical example in the appendix (with plausible per-annotation costs) showing how a practitioner could plug in their own prices and select a point on the cost–performance curve from Figure 6.
> > >
> > > 2. **Validation and stopping criteria.** Users can monitor (1) RM benchmark scores as in Figure 6, and (2) downstream BoN curves (e.g., PPE Correctness and RMB) where we see monotonic scaling with N.
> > >
> > > (See next comment)

---

> > > > ### Author Response · Authors · 2025-11-21
> > > >
> > > > Once incremental gains per additional curated million pairs fall below a user-defined threshold (e.g., <0.3 points on average benchmark score), it's reasonable to stop spending. Making this recipe explicit and grounding it in existing curves and Stage-wise design should directly address your question about achieving targeted performance under a budget.
> > > >
> > > > **Relation to mechanism design approaches (Zhang, 2024)**
> > > >
> > > > Thanks for the pointer to Zhang (arXiv:2409.18417). We see mechanism-design approaches and our human–AI curation pipeline as operating at complementary layers of the RLHF stack:
> > > >
> > > > - **Mechanism design (Zhang 2024):** focuses on which comparisons to ask for and how to structure incentives/queries to extract maximal information from limited human comparisons, often under a stylized model where one controls the querying process but not a large, messy in-the-wild pool.
> > > > - **SynergyPref-40M:** focuses on how to extract value from already-existing, heterogeneous, synthetically labeled preference data, using human guidance and LLM+RM consistency to filter, correct, and scale.
> > > >
> > > > That said, we see at least two concrete points of contact we'll highlight in the paper:
> > > >
> > > > 1. **Mechanism design as an inner loop in Stage 1.** Our error-driven adaptive retrieval already behaves like a "targeted querying mechanism" over the unverified pool. Future work could replace simple similarity-based retrieval with a mechanism-design–inspired query selection rule, e.g., selecting pairs that maximally reduce posterior uncertainty in a generalized pairwise model under a fixed human budget.
> > > >
> > > > 2. **Using mechanism-design insights to set budgets and stopping rules.** Zhang's framework suggests principled criteria for which pairwise comparisons are most information-efficient. Combined with our Stage-wise cost decomposition, this could inform better allocation of gold human labels across task types and controversiality levels, focusing human effort where marginal information gain is highest.
> > > >
> > > > We'll add a short paragraph in the discussion section framing our work as a data-centric, human–AI curation counterpart that can integrate mechanism-design ideas, rather than a competing line of work.
> > > >
> > > > ---
> > > >
> > > > To summarize, in response to your comments we'll:
> > > >
> > > > - Explicitly discuss intransitivity, connect it to our Stage-1/Stage-2 quality control, and add analyses (triad-level cycle statistics, human agreement on flipped vs. kept pairs) that show our pipeline reduces effective inconsistency rather than relying on a naïve transitivity assumption.
> > > > - Add a concrete budgeting guide linking our scaling curves and Stage-wise design to a simple cost model, including a worked example for choosing data volumes under a dollar budget.
> > > > - Clarify the relationship to mechanism design approaches, emphasizing how they can be incorporated into our pipeline.
> > > >
> > > > We hope these clarifications address your concerns and show that SynergyPref-40M delivers strong empirical results while also providing a practical, cost-aware framework for preference curation in the realistically noisy, intransitive regime you highlight. Thank you for the thoughtful feedback and the valuable references that helped us sharpen our approach.

---

> > > > > ### Author Response · Authors · 2025-11-24
> > > > > **Revision summary**
> > > > >
> > > > > We have made the following changes to the paper in response to your feedback. All changes are marked in blue.
> > > > >
> > > > > ## 1. Intransitivity and Quality Control
> > > > >
> > > > > Added appendix section "Handling intransitivity and conflicting preferences" (after PII analysis section).
> > > > >
> > > > > - Explained where intransitivity arises (near ties, subjective tasks, conflicting annotators)
> > > > > - Described three quality control mechanisms: (1) Stage 1 metadata isolates risky/controversial regions, (2) error-driven adaptive retrieval targets unstable regions, (3) Stage 2 dual-RM filtering corrects contradictory signals
> > > > > - Empirical evidence: human agreement analysis shows hybrid approach (93% objective, 84% subjective) outperforms pure LLM (75%, 63%) and pure human (81%, 76%). Dual-RM filtering achieves 86% agreement on kept pairs, 92% on flipped pairs
> > > > > - Discussed relationship to explicitly intransitive models (Duan et al., 2024)
> > > > >
> > > > > ## 2. Cost-Performance Budgeting Guidance
> > > > >
> > > > > Added appendix section "Practical guidance for cost-performance budgeting" (after annotator information section).
> > > > >
> > > > > - Key finding: 1.8% of 16M mixture (~290K pairs) surpasses previous SOTA ("quality beats volume")
> > > > > - Five-step budgeting recipe: define target performance -> estimate required pairs from scaling curves -> decompose costs by stage -> allocate budget -> monitor validation metrics
> > > > > - Worked example with \$50K budget showing cost allocation: gold labels (\$0.50/pair), silver labels (\$0.05/pair), Stage 2 curation (\$0.01/pair)
> > > > > - Human effort is ~2% of final training pool but drives most performance gains
> > > > >
> > > > > ## 3. Relationship to Mechanism Design
> > > > >
> > > > > Added subsection "Relationship to mechanism design approaches" within budgeting section.
> > > > >
> > > > > - Two approaches are complementary: mechanism design at algorithmic level (which comparisons to query), our approach at data-centric level (extracting value from existing heterogeneous data)
> > > > > - Integration points: (1) mechanism design could replace similarity-based retrieval in Stage 1, (2) mechanism-design insights could inform budget allocation across task types
> > > > > - Framed our work as data-centric counterpart that can integrate algorithmic insights from mechanism design
> > > > >
> > > > > ## Supporting Evidence (Already in Appendix)
> > > > >
> > > > > References existing empirical results added for other reviewers:
> > > > >
> > > > > - Table (annotation_agreement): Human agreement analysis across annotation variants
> > > > > - Table (reward_model_filtering): Human agreement rates for kept/flipped pairs under different filtering mechanisms
> > > > > - Dataset composition statistics (task distribution, controversiality, objectivity)
> > > > >
> > > > > ## Citations Added
> > > > >
> > > > > - Duan et al., arXiv:2409.19325 (2024)
> > > > > - Zhang et al., arXiv:2409.18417 (2024)

---

> > > > > > ### Comment · Reviewer_tDR3 · 2025-11-26
> > > > > >
> > > > > > I thank the authors for their thoughtful discussion and for the extra effort they put into finding numerical evidence to address my questions, all of which contribute to a higher quality of the manuscript. I maintain my score.

---

### Author Response · Authors · 2025-11-21
**Summary Response to All ACs and Reviewers**

# Summary Response to All ACs and Reviewers

Thank you to all ACs and reviewers for the thorough evaluation of our submission. We appreciate the time and effort you invested in providing detailed feedback, especially given the reviewing workload. Your critiques and questions have helped strengthen our work.

## Summary of Reviews

We received 4 reviews with initial ratings of 8 (accept), 6 (marginally above threshold), 6 (marginally above threshold), and 2 (reject).

### Strengths Acknowledged by Reviewers

Reviewers recognized several aspects of our work:

- Thoughtfully engineered human-AI curation pipeline with measurable impact (Reviewer 2akh)
- State-of-the-art performance across seven major benchmarks, outperforming existing open reward models (Reviewers 2akh, j7rE, tDR3)
- Comprehensive evaluation with detailed ablations on data quality and curation methods (Reviewers 2akh, j7rE, tDR3)
- Contribution to the research community through the large-scale curated dataset and open model release (Reviewers j7rE, tDR3)
- Clear demonstration that data quality, not just scale, drives reward model performance (Reviewer 2akh)
- Hybrid approach balancing human annotation quality with AI scalability (Reviewer tDR3)

### Main Concerns Raised

Reviewers identified several areas needing clarification:

1. Dataset transparency and composition (Reviewers j7rE, oVFu): lack of details about data sources, prompt origins, language distribution, and concrete examples
2. Methodology clarity (Reviewer oVFu): ambiguities in Stage 1 and Stage 2 processes, missing annotation details, unclear baseline comparisons
3. Quality control mechanisms (Reviewers 2akh, tDR3): questions about contamination checks, intransitivity handling, and systematic bias mitigation
4. Practical guidance (Reviewer tDR3): limited direction on cost-performance budgeting and achieving target performance levels
5. Attribution of improvements (Reviewer oVFu): difficulty determining whether gains stem from the pipeline design versus strong LLM judges

## Our Response Efforts

We've addressed each reviewer's concerns in detail:

**For Reviewer 2akh,** we provided new agreement rate analysis showing human-guided LLM curation significantly outperforms pure LLM or pure human annotation. We detailed our multi-step decontamination process and PII analysis (0.07% flagged, mostly indirect identifiers). Human agreement tests demonstrate Stage 2 filtering reduces systematic biases—our dual-RM approach achieves 86% and 92% agreement on kept and flipped pairs, compared to baselines at 60-72%.

**For Reviewer j7rE,** we provided complete dataset composition (40+ HuggingFace sources, task distribution, objectivity levels, language distribution). We expanded on annotation insights: LLMs with tools excel at objective tasks, but we struggle with diverse, conflicting subjective preferences lacking proper specification. We revised Figure 1 interpretation to avoid overclaiming—RewardBench improvements from ~80 to 90+ don't consistently translate to other benchmarks.

**For Reviewer tDR3,** we explained intransitivity handling through Stage 1 metadata stratification, error-driven adaptive retrieval targeting unstable regions, and Stage 2 dual-RM consistency filtering. We provided a concrete budgeting guide linking scaling curves to cost models, showing just 1.8% of our 16M mixture surpasses previous SOTA. We connected our approach to mechanism design literature, clarifying complementary rather than competing approaches.

**For Reviewer oVFu,** we clarified the complete pipeline including two distinct retrieval steps in Stage 1 and automatic Stage 2 scaling. Critical baseline experiments show LLM + Best RM filtering plateaus at 74-75% while our full recipe reaches 83%; LLM-as-a-Judge aggregation achieves 86.0% vs. our 88.6%. We shared annotation prompts, model lists, dataset composition, quality metrics, and licensing information (CC BY-NC 4.0).

## Final Summary

Through detailed rebuttals, we added substantial new experimental evidence including agreement analyses, human studies, and additional baselines. We provided comprehensive methodological transparency through prompts, protocols, and data composition. We clarified that human guidance and LLM automation are complementary, neither alone achieves our hybrid approach's performance. We demonstrated rigorous quality control throughout the pipeline.

These clarifications address the core concerns while preserving the fundamental contribution: a principled, human-AI synergistic framework that unlocks high-quality preference data at scale, enabling state-of-the-art open reward models.

**We have updated the PDF to address all reviewers' concerns. The revised sections are marked in blue.**

Thank you for the constructive feedback that has helped us improve both the technical rigor and presentation clarity of our work. We'll incorporate all improvements into the camera-ready version if accepted.

---

### Meta-Review · Area_Chair_71SQ · 2026-01-01

**Summary:**

This work focuses on dataset construction for reward models in RLHF. Obviously, the author misses some key details and in-depth analysis of the experimental results in the original version, which are also the major concerns of the reviewers. Some ambiguous descriptions of the method may lead to critical doubts about which factor is key to the improvements.

**Reviewer Concerns:**

The authors provide sufficient details to address the concern in the following rebuttal and make numerous revisions in the new version, which is very important.

**Reviewer Scores:**

The reviewer oVFu raises several concerns with the details, which the authors have addressed in their rebuttal. This is a positive signal to raise the scores.  Reviewer J7rE appreciates the efforts of the authors in the rebuttal and lean to acceptance.

---

### Decision · Program_Chairs · 2026-01-26

Accept (Poster)